# Current Knowledge about the Impact of Microgravity on Gene Regulation

**DOI:** 10.3390/cells12071043

**Published:** 2023-03-29

**Authors:** Thomas J. Corydon, Herbert Schulz, Peter Richter, Sebastian M. Strauch, Maik Böhmer, Dario A. Ricciardi, Markus Wehland, Marcus Krüger, Gilmar S. Erzinger, Michael Lebert, Manfred Infanger, Petra M. Wise, Daniela Grimm

**Affiliations:** 1Department of Biomedicine, Aarhus University, Hoegh Guldbergs Gade 10, 8000 Aarhus, Denmark; 2Department of Ophthalmology, Aarhus University Hospital, Palle Juul-Jensens Blvd. 99, 8200 Aarhus, Denmark; 3Department of Microgravity and Translational Regenerative Medicine, Medical Faculty, University Hospital Magdeburg, Otto von Guericke University, Universitätsplatz 2, 39106 Magdeburg, Germany; 4Clinic for Plastic, Aesthetic and Hand Surgery, Medical Faculty, University Hospital Magdeburg, Otto von Guericke University, Leipziger Straße 44, 39120 Magdeburg, Germany; 5Research Group ‘Magdeburger Arbeitsgemeinschaft für Forschung unter Raumfahrt-und Schwerelosigkeitsbedingungen’ (MARS), Otto von Guericke University, Universitätsplatz 2, 39106 Magdeburg, Germany; 6Gravitational Biology Group, Department of Biology, Friedrich-Alexander University, 91058 Erlangen, Germany; 7Postgraduate Program in Health and Environment, University of Joinville Region, Joinville 89219-710, SC, Brazil; 8Institute for Molecular Biosciences, Johann Wolfgang Goethe Universität, 60438 Frankfurt am Main, Germany; 9The Saban Research Institute, Children’s Hospital Los Angeles, University of Southern California, 4650 Sunset Blvd, Los Angeles, CA 90027, USA

**Keywords:** microgravity, gene expression, gene regulation, space, cancer, cells, plants, microorganisms

## Abstract

Microgravity (µ*g*) has a massive impact on the health of space explorers. Microgravity changes the proliferation, differentiation, and growth of cells. As crewed spaceflights into deep space are being planned along with the commercialization of space travelling, researchers have focused on gene regulation in cells and organisms exposed to real (r-) and simulated (s-) µ*g*. In particular, cancer and metastasis research benefits from the findings obtained under µ*g* conditions. Gene regulation is a key factor in a cell or an organism’s ability to sustain life and respond to environmental changes. It is a universal process to control the amount, location, and timing in which genes are expressed. In this review, we provide an overview of µ*g*-induced changes in the numerous mechanisms involved in gene regulation, including regulatory proteins, microRNAs, and the chemical modification of DNA. In particular, we discuss the current knowledge about the impact of microgravity on gene regulation in different types of bacteria, protists, fungi, animals, humans, and cells with a focus on the brain, eye, endothelium, immune system, cartilage, muscle, bone, and various cancers as well as recent findings in plants. Importantly, the obtained data clearly imply that µ*g* experiments can support translational medicine on Earth.

## 1. Introduction

Outer space is humanity’s dream, and this adventure is becoming a reality. Recently, the Orion spacecraft returned to Earth. Splashdown was on 11 December 2022. The robotic and human Moon exploration program (ARTEMIS) of the National Aeronautics and Space Administration (NASA), European Space Agency (ESA), Japanese Aerospace Exploration Agency (JAXA), and the Canadian Space Agency (CSA) has started. The major objective of ARTEMIS is to build a base on the Moon and thus facilitate human exploration of Mars.

While conquering space, astronauts, cosmonauts, and taikonauts, together with space tourists, will face microgravity (µ*g*) and cosmic radiation, among other stressors. In addition, on their journey to the Moon and Mars, humans in space will live under the influence of the different gravity levels ranging from 1*g* to real (r-) µ*g* conditions on the Moon (0.17*g*) or Mars (0.376*g*) [1]. Normally, all Earth’s creatures are accustomed to gravity’s force, and gravity changes will significantly influence these organisms’ health and function. Therefore, detailed medical examinations have been performed on humans reporting various health problems occurring early and late during space missions [2]. For example, standing upright on Earth gathers fluid in the lower extremities. Under µ*g* conditions, the fluid becomes displaced into the chest and head (so-called ‘puffy face’), and the astronauts exhibit ‘bird legs’ (very slender legs) [3]. Furthermore, there is an increase in heart filling and diuresis, which leads to an overall reduction in blood volume and an elevation in hematocrit. In addition, space travelers excrete around 1.5 L of urine during the first few hours in space. Cardiovascular problems comprise arrhythmias, cardiac atrophy, low blood pressure, and orthostatic intolerance [4]. In addition, many astronauts suffer from space travel-sickness, the so-called space adaptation syndrome (SAS), associated with nausea and vomiting [5]. Other health problems are space flight-associated neuro-ocular syndrome (SANS) [6] and back pain [7]. Since the spine is stretched by µ*g* and becomes almost straight, the astronauts’ height increases by up to 7.5 cm on the International Space Station (ISS). After the spaceflight, many astronauts continue to experience back pain and report problems with the intervertebral discs [7].

In addition, weightlessness affects the skeletal muscles, leading to a decrease in muscle mass, strength, and endurance [2]. Furthermore, low gravity increases calcium loss from bones, inhibits bone formation, and reduces bone mass and osteoporosis [8]. The result is an increase in kidney stone formation. Furthermore, the immune system is impaired. As consequence, space travelers exhibit an increased susceptibility to infections, the reactivation of dormant viruses, and disturbed wound healing on long-term missions [2,9].

It has been known for many years that real (r-) and simulated (s-) µ*g* induce various changes in cells, microorganisms, animals, and plants [10]. Gene regulation is the key factor to the ability of a cell or an organism to sustain life and respond to environmental changes. It is a universal process used to control the amount and spatiotemporal expression of genes. Multiple publications have so far demonstrated that µ*g* changes the gene expression pattern and alters signal transduction pathways in various cell types [11,12,13,14]. Omics studies have been performed to enlarge the current knowledge about the impact of the space environment on humans, rodents, cells, and plants. NASA’s GeneLab database (https://genelab.nasa.gov/) is available and provides access to omics data from space and µ*g* simulation studies [15]. This database supports the generation of new data, facilitates the proposal of new hypotheses, provides high school student-training programs, and finally helps define the risks of human space-exploration [15].

This concise review summarizes the latest results published over the last five years (2017–2023) on µ*g*-induced alterations in the gene expression of bacteria, protists, fungi, animals, humans, cells, and plants. The review will cover changes in the numerous mechanisms involved in gene regulation, including regulatory proteins, microRNAs, and the chemical modification of DNA. All these publications point to the fact that µ*g* is an extreme stressor affecting gene expression and results in numerous problems that must be addressed before and during space missions.

## 2. Materials and Methods

To collect suitable studies, PubMed (https://pubmed.ncbi.nlm.nih.gov/), Scopus (https://www.scopus.com/home.uri), Embase (https://www.embase.com), Web of Science (https://clarivate.com), and Google scholar (https://scholar.google.com/) (last accessed on 23 March 2023) were queried using the search terms “(microgravity) AND (bone)”, “(weightlessness) AND (bone)”, “(microgravity) AND (muscle)”, “(weightlessness) AND (muscle)”, “(microgravity) AND (cartilage)”, “(weightlessness) AND (cartilage)”, “(thyroid cancer) AND (microgravity)”, “(thyroid cancer) AND (weightlessness)”, “(prostate cancer) AND (microgravity)”, “(prostate cancer) AND (weightlessness)”, “(breast cancer) AND (microgravity)”, “(breast cancer) AND (weightlessness)”, “(lung cancer) AND (microgravity)”, “(lung cancer) AND (weightlessness)”, “(immune system) AND (microgravity)”, “(immune system) AND (weightlessness)”, “(t cells) AND (microgravity)”, “(t cells) AND (weightlessness)”, “(colorectal cancer) AND (microgravity)”, “(colorectal cancer) AND (weightlessness)”, “(colorectal cancer) AND (spaceflight)”, “(colorectal cancer) AND (space travel)”, “(colorectal cancer) AND (astronaut)”, “(colorectal cancer) AND (cosmonaut)”, “(hepatocellular cancer) AND (microgravity)”, “(hepatocellular cancer) AND (weightlessness)”, “(hepatocellular cancer) AND (spaceflight)”, “(hepatocellular cancer) AND (space travel)”, “(hepatocellular cancer) AND (astronaut)”, “(hepatocellular cancer) AND (cosmonaut)”, “(gastric cancer) AND (microgravity)”, “(gastric cancer) AND (weightlessness)”, “(gastric cancer) AND (spaceflight)”, “(gastric cancer) AND (space travel)”, “(gastric cancer) AND (astronaut)”, “(gastric cancer) AND (cosmonaut)”, “(brain) AND (microgravity)”, “(brain) AND (weightlessness)”, “((eye) OR (eyes)) AND ((microgravity) OR (simulated microgravity) OR (real microgravity) OR (weightlessness) OR (weightlessness simulation) OR (space flight) OR (spaceflight) OR (clinostat microgravity))”, “(pancreatic cancer) AND (microgravity)”, “(pancreatic cancer) AND (weightlessness)”, “(pancreatic cancer) AND (spaceflight)”, “(pancreatic cancer) AND (space travel)”, “(pancreatic cancer) AND (astronaut)”, “(pancreatic cancer) AND (cosmonaut)”, “(melanoma) AND (microgravity)”, “(melanoma) AND (weightlessness)”, “(melanoma) AND (spaceflight)”, “(melanoma) AND (space travel)”, “(melanoma) AND (astronaut)”, “(melanoma) AND (cosmonaut)”, “(plant) AND (microgravity)”, “(plant) AND (microgravity)”, “(endothelial cells) AND (microgravity)”, “(endothelial cells) AND (weightlessness)” in conjunction with all their combinations with “AND (gene expression)“, “AND (gene regulation), “AND (genetics)”, and “AND (transcriptome)”. For the chapters covering bacteria, protists, fungi, and animals excluding mice, “gene expression AND microgravity” was used as a search term in Google scholar. Subsequently, hits containing “cancer”, “stem”, “thaliana”, “plant”, “human”, “mice”, and “skeletal” were eliminated. The remaining papers were checked for actual relevance; duplicates, conference posters, and conference papers were excluded, and relevant references from the remaining papers were included (assessed on 8 March 2023).

For all searches, results were confined to hits from 2017 to 2023 and given in Figure 1.

## 3. Results

### 3.1. Microgravity Platforms

Research in µ*g* requires special efforts. While accelerations above 1*g* are easily achieved in the laboratory, e.g., by centrifuges, µ*g* conditions cannot be set up on demand. However, there are certain experimental facilities allowing experiments in µ*g*. We provide a brief overview of the different possibilities. Key parameters are shown in Table 1.

If a body submits to an acceleration, no force is exerted upon it. Therefore, a freefall in an evacuated tube (no air drag) generates moments of weightlessness. The experiment must be housed in a sturdy, pressurized container that survives the deceleration upon impact at the bottom of the tube. Different designs exist: the evacuated tube can either be interred or erected inside a shielding tower. An example of the first design is NASA’s Zero Gravity Research Facility at the Glenn Research Centre [19]: a 155 m deep bore houses a steel tube in which an experimental carrier capsule is dropped. The second design is used at the Bremen Drop Tower, Germany [29]: here, a concrete tower of 146 m in height houses a drop tube of 122 m. In 2004, a catapult was installed underneath the deceleration bucket [17]. Catapulting the experiment capsule upwards before it falls back down doubles the µ*g* period, but it introduces a strong acceleration event during launch. Smaller versions only allow for short periods of µ*g*, but are simpler to construct as they do not need an evacuated tube. These facilities have become widely used in recent years [30,31]. Recently, a novel concept was developed at the Einstein-Elevator at Hannover University, Germany [21]. Here, an elevator-like cabin (the gondola) moves on rails within a 40 m tower. The experiment capsule resides within the gondola, the only volume to be evacuated. The gondola can be dropped or accelerated from the bottom to the top to fall back again. The experiment capsule floats freely inside the gondola. The same design is also implemented in the latest drop capsules at the Bremen Drop Tower for highly sensitive experiments [18]. Theoretically, one drop per every four minutes is possible at the Einstein-Elevator, adding up to 100 drops on a typical 8-h workday. However, the actual number of drops per day highly depends on the individual experiment design. A similar facility is under construction at the Chinese Academy of Sciences [32]. In general, drop tower experiments must be fully automated. Of all platforms, drop towers offer the most frequent experiment opportunities, limited only by preparation time and funding.

Parabolic flights provide more prolonged periods of µ*g* [22]. A typical parabolic flight campaign (PFC) offered by the company Novespace, Bordeaux-Merignac, France, contains 3–4 flight days with 31 parabolas on each flight [33]. During a parabola, the plane first climbs at an ever-increasing angle (“pull-up”). At 50° (“injection”), thrust is reduced, and the plane and everything in it assumes the path of a parabola: the µ*g* period begins (Figure 2A). Next, the plane climbs to the apex point and begins a downward trajectory. The pilots adjust the flight path for drag and reorient the plane accordingly. At an angle of approximately −42°, the power to the engines is restored and the plane is pulled back into the horizontal (“pull-out”), which marks the end of the µ*g* period (Figure 2A). A parabolic flight allows scientists to accompany their experiment in the plane (Figure 2B), so it must not be fully automated (Figure 2B–D). It also allows for the exchange sample containers to obtain more material for further analysis. However, samples produced later in the flight have experienced all changes in gravity that occurred before, which has to be considered during the analysis. PFCs take place multiple times per year, making them the second most available µ*g* platforms.

The Earth’s gravity must be left behind to achieve more extended periods of µ*g* exposure. Sounding rockets fly on a ballistic curve and reach space (>100 km height) [23]. Various rocket sizes are available, offering different µ*g* times. All experiments must be fully automated. The New Shepard Rocket from Blue Origin is not a traditional sounding rocket and offers opportunities to conduct experiments during its qualification phase [24]. These platforms are available only a few times per year.

Historically, µ*g* periods from days to weeks were achieved by flights with Russian recoverable satellites or, in the past, with the Space Shuttle (officially called the Space Transportation System (STS)). These opportunities are now quite rare: STS retired in 2011, and as of 2022, only one campaign with the Russian BION M-2 is being prepared, and plans for a new biosatellite were recently announced [34]. A Chinese biosatellite was launched in 2016, but it is unclear whether this approach will be pursued further [28]. Sometimes, experiments can be performed on resupply missions to the ISS [25,26]. A rather new approach is to miniaturize and automatize experiments so that they fit into a CubeSat [35]. The smallest format called 1U (for unit) measures approximately 10 × 10 × 10 cm. Units can be combined to form 2U, 3U, or 6U versions. The constraints regarding space and mass are challenging, although the cost for launching is quite affordable due to their small size. Other than the platforms described before, CubeSats are not capable of returning samples to the laboratory, which makes advanced analyses nearly impossible. Nevertheless, biological experiments were successfully conducted or launched as recently as 2022 in the context of the ARTEMIS mission [36,37].

The longest µ*g* exposure can be achieved on the ISS: the duration is basically unlimited [38]. Not only does the potential exposure time trump all other platforms, but so do costs and preparation time, making the ISS a rather exclusive research facility. In addition, 1*g* reference centrifuges are installed onboard to allow for 1*g* inflight controls, which gives a vital control to check for space-specific factors [39]. In 2021, the People’s Republic of China launched the core module Tianhe of its own Space Station, Tiangong. Tiangong is a Chinese national project, but non-Chinese scientists can obtain access via collaborations and benefit from the additional capacity for µ*g* experiments [40]. As of November 2022, two science modules, ‘*Wentian*’ and ‘*Mengtian*’, have completed the station in its first-stage configuration. In the same way as experiments on Biosatellites, taxi flights, and CubeSats, experiments on the ISS are quite rare as compared to experiments on the platforms described earlier.

Due to the limited possibilities for r-µ*g* exposure, techniques to simulate µ*g* were developed. It is a subject of discussion whether these techniques achieve their objectives or merely exert some stress on the test subjects, rendering results different from 1*g* controls. Numerous studies compared r-µ*g* results with those obtained with various simulations [41,42] and showed that their suitability depends on the test organism.

Using a two-dimensional (2D)- or three-dimensional (3D)-clinostat, a sample can be rotated on one or two axes so the resulting force averages to zero [43]. A similar device, the random positioning machine (RPM), rotates the sample in a non-continuous movement, including directional changes, achieving a similar result (Figure 3) [44]. However, the assumption that gravity is nullified is only valid for a point right in the center of the movement because outside of it, the rotational movement of the sample will result in the application of a centrifugal force. Therefore, this method is limited to small organisms or small parts of tissue samples placed in the center of the movement.

Rotating wall (perfusion) vessels (RWVs or RWPVs, also high aspect ratio vessels, HARVs or low-shear modelled microgravity (LSMMG)) are cylinders filled with liquid (medium) in which particles or organisms are suspended (Figure 3C) [45,46,47]. The rotation compensates for the sedimentation of the particles, keeping them in suspension (and therefore, the considerations in Dedolph and Dipert apply just as well).

The rotating cell culture system (RCCS) is a similar type, which uses Petri dish-like vessels. Recently, simpler designs for custom-made devices were validated to make the technology more accessible [48,49].

Diamagnetism describes that a non-para- or non-ferromagnetic substance is repelled by an external magnetic field because it induces a magnetic field in the opposite direction. Because many organic compounds and water are diamagnetic, biomatter can be suspended in a strong enough magnetic field (>15 T). This so-called magnetic levitation can be used to simulate µ*g*. It is commonly used for the exposition of cell cultures [50], tissues, and small organisms [51]; however, a preprint paper has outlined the possibility of construction in the near future of a magnetic levitation device large enough to accommodate a person over more extended periods [52].

A well-known method to simulate µ*g* in rodents is hindlimb suspension (also called hindlimb unloading or anti-orthostatic rodent suspension) [53]. Here, the animal’s hindquarters are lifted by its tail with a pulley above the cage. The animal behaves normally after some days of adaptation, using only its front legs for moving around in its cage. This simulates both the unloading of the force onto the bones and muscles and the cephalic fluid shift.

At present, the only practical methods to simulate the effects of µ*g* on humans are head-down bed rest (HDBR) studies: the test person lies on an inclined bed (−6°) with their heads lower than their feet. Experiments last from days to many weeks. Typical effects resulting from r-µ*g* exposure, such as reduced bone density, muscle mass, and muscular strength, or cephalic fluid shift, are also observed in HDBR [54].

Finally, it is important to stress that caution should always be exercised when comparing data obtained by ground-based models of µ*g* without radiation to experiments conducted in real µg which per se include a significant additional ionizing radiation. One way to solve this problem is to use an onboard 1*g* reference centrifuge during the spaceflight [38].

### 3.2. Recent Reports on Microgravity Effects on Bacteria, Protists, and Fungi

#### 3.2.1. Microgravity Affects Homeostasis between Microbiome and Host

The µ*g* environment poses many stressors on astronauts. Siddiqui et al. believe in the importance of maintaining a healthy gut biome during long-term space exposure because of the correlation between gut biome and health [55]. The gut biome interacts with the immune system and is vital for bone and muscle physiology and general metabolic and neurological health. Gut bacteria and their metabolites (e.g., short-chain fatty acids) affect tissue functions and the enteric nervous system. Spaceflight strongly dysregulates the diversity and composition of the gut biome [56]. Dysbiosis of the gut biome may lead to increased gut permeability triggering the release of inflammatory interleukins, tumor necrosis factor (TNF), and vascular endothelial growth factor a (VEGFa) [55]. It is important to note that bacteria are not the sole organisms in the gut. Viruses, archaea, fungi, and eukaryotic parasites/commensals such as helminths, which all interact with each other, must also be considered. Different groups of organisms have different effects on the host immune system. As reviewed in Vemuri et al. [57], certain bacteria, viruses, some archaea, and fungi stimulate macrophages or dendritic cells (which in turn activate macrophages), which trigger T cells and T helper cells, leading to inflammation. In contrast, other bacteria, archaea, and helminths regulate Foxp3-expressing T regulatory (Tregs) cells, which downregulate active T helper cells and, by this, have a more anti-inflammatory effect. To our best knowledge, the effects of µ*g* on these complicated interactions of the gut organisms and possible impacts on homeostasis during spaceflight leading to inflammation-inducing dysbiosis have not yet been reported.

Probiotic organisms are supposed to support a properly functioning intestinal system. One such organism, *Lactobacillus reuteri*, was exposed to s-µ*g* using an RWV and RPM [58]. Interestingly, the two different approaches to simulating µ*g* rendered rather different results, which suggests that *Lactobacillus reuteri* can sense a difference and react differently: over time, cells on the RWV tended to upregulate the investigated generic stress genes, while cells on the RPM reacted by downregulation. It is noteworthy that cell density did not significantly differ at the end of the experiment, and production of the antimicrobial substance reuterin was higher in both simulation methods than in the 1*g* control.

Wang et al. proposed that the dysregulation of the immune system affects the balanced gut microbiome, leading from a healthy gut biome with commensals to a diseased gut biome [59]. 

During a hindlimb unloaded (HU) study using mice, it was found that dysbiosis in the gut microbiome became evident after three days. The fraction of firmicutes (which includes mainly the genera *Enterococcus*, *Lactobacillus*, *Clostridium*, and *Faecalibacterium*) increased. This dysregulation is likely caused by cellular stress because 4-phenyl butyric acid (4-PBA), which mitigates ER stress, was found to be a countermeasure against dysbiosis compared to control mice [60].

The microbiome of plants is also affected by µ*g*. Simulated µ*g* affects the composition of endophytic bacteria [61]. An increase in *Enterobacteriaceae* and *Pseudomonadaceae* and decreases in *Burkholderiaceae* and *Bacillaceae* in the wheat rhizosphere were found. This is most likely due to a change in root metabolites. Decreases in carbohydrate metabolism, phenylalanine, tyrosine, and tryptophan biosynthesis, flavonoid biosynthesis, and benzoxazinoid biosynthesis functional pathways were detected as well as an increase in metabolites within the amino acid metabolic pathways, such as tyrosine metabolism, cysteine, and methionine metabolism, lysine biosynthesis, alanine, and aspartate and glutamate metabolism. In addition, specific significantly altered secreted metabolites such as D-glucuronate, D-ribose, arbutin, epicatechin, or indoleacetic acid, influenced and changed the microbial composition. Arbutin, D-glucuronate, D-ribose, and epicatechin were found to be positively correlated with *Burkholderiaceae* and negatively correlated with *Enterobacteriaceae*. Homovanillic acid excretion correlated negatively with *Burkholderiaceae* and indole negatively with *Pseudomonadacea*.

#### 3.2.2. Microgravity and Virulence

Green et al. analyzed studies concerning bacteria and the human immune system under µ*g* or low-shear force environment [62]. Astronauts face an increased risk of infections because the proliferation, biofilm formation, and expression of virulence genes in bacteria are often increased (recent reports, e.g., [63,64]), while in turn, the human immune system is impaired under space conditions [65,66,67]. Among others, changes in the cytokine expression influencing the cross-talk between the immune cells, decreased pathogen recognition, and changes in the composition of immune cells, as well as a decreased production of granzyme B and perforin, were identified [62,68]. Analyses of the proteome and transcriptome of human epithelial cells infected with *Salmonella typhimurium* during the STL-IMMUNE study onboard the Space Shuttle mission STS-131 revealed significant differences between cells infected on the ground and those infected in space [69]. Enriched Kyoto Encyclopedia of Genes and Genomes (KEGG) pathways were tumor necrosis factor (TNF) signaling, nuclear factor-kappa B (NF-*κ*B) signaling, NOD-like receptor signaling, and legionellosis, among others.

Microgravity does not always lead to increased virulence in bacteria strains. For instance, the virulence of *Yersinia pestis* exposed to s-µ*g* was reduced [70]. Many virulence-related genes, such as genes for the type-III secretion system, were downregulated. Mice infected with s-µ*g* bacteria died with some delay compared to those infected with control cells, indicating a lower degree of virulence. In addition, biofilm formation was found to be decreased in s-µ*g* [63]. Changes in the virulence of harmful microorganisms threaten crewed spaceflights because spaceflight conditions impair the human immune system, and the virulence of some bacteria increases. In a recent study, the microbiomes (of the skin and saliva, not of the gut biome) of four astronauts on the ISS were investigated before, during, and after their time on the ISS [71]. While evident biome changes were found in two astronauts, the microbiome alterations in the others were less pronounced. However, the species composition was altered during the mission, and an antimicrobial resistance gene expression was elevated, but no universal trend comparing all astronauts was found. In addition, changes in the virulence of bacteria were observed under r-µ*g* and s-µ*g* conditions [72,73].

In contrast, Gilbert et al. reported increased virulence of *Serratia marcescens* against *Drosophila melanogaster* on the ISS in µ*g* [74]. The increased virulence and antimicrobial resistance are probably due to elevated mutation rates in space. *Listeria monocytogenes* showed reduced virulence and lower tolerance towards heat and acid conditions after HARV cultivation, while the cold tolerance was increased [75]. In addition, decreased expression levels of heat stress and virulence-related genes and an upregulation of cold stress genes were found [75]. In *Bacillus subtilis*, mutation rates of an investigated resistance gene were found to be far higher compared to Earth conditions [76]. However, s-µ*g* (by HARV) also increased the antibiotic resistance in *Escherichia coli* [77], conveying additional resistance against four more antibiotics. It is speculated that either µ*g* or the space environment may lead to new epigenetic changes and an increase in mutation rate and horizontal gene transfer (HGT) [78]. It is known that bacteria use HGT for adaptation to adverse environments or rapid changes in the environment, respectively. Increased HGT was also observed under s-µ*g* conditions. Urbaniak et al. found an increased rate of HGT from co-cultured *Acinetobacter pittii* as donor to *Staphylococcus aureus* recipient strains during HARV treatment compared to 1*g* conditions [79].

#### 3.2.3. Physiological Effects of Microgravity on Bacteria and Fungi

Recently, Sharma and Curtis [80] analyzed the reported effects of µ*g* on bacterial metabolism. Although some common metabolic changes were identified, such as an increase in carbohydrate metabolism, changes in carbon substrate utilization, and alterations in amino acid metabolism as an indication for oxidative stress, no universal µ*g* response appeared. The authors concluded that future research should increase the focus on the metabolomics of bacteria as well as the corresponding changes due to stressful environments because the understanding of cause-and-effect mechanisms may lead to valuable new biotechnological applications. Effects of s-µ*g* on *Vibrio fischeri* induced an increase in the release of lipopolysaccharides and enhanced production of outer membrane vesicles. In addition, the outer membrane stability was impaired because the bacteria became more sensitive against sodium-dodecyl-sulfate (SDS) or polymyxin B. This affects the microbe-associated molecular pattern signals (MAMPs) of the bacteria and may impair bacteria–host interaction [81]. The formation of antioxidants in the mitigation of µ*g*-induced stress was found in some microorganisms.

Wild-type strains and colorless strains of *Knufia chersonesos*, a black fungus inhabiting extreme environments, were exposed to s-µ*g* in HARVs and subsequently analyzed in terms of secretome, proteome, and phenotype. Although no prominent indications for stress were found and no changes in the phenotype, differences in the secretome and proteome modulation within the two strains were detected [82].

Comparisons of melanized and non-melanized strains of the yeast *Cryptococcus neoformans* on the ISS revealed a far higher survival rate for melanized strains. As a potent antioxidant, melanin may protect the cells from oxidative stress caused by µ*g* and space radiation [83]. In addition, an increase in protecting pigments (carotenoids) after exposure to s-µ*g* was reported for *Haloarcula argentinensis* [84]. Growth, sporulation, as well as germination was found to be increased, while thickness of biofilms decreased in *Fusarium solani* exposed to random positioning [85].

A summary of all findings is given in Table 2.

### 3.3. Effects of Real or Simulated Microgravity on the Gene Expression in Animals

*Vibrio fischeri*, as mentioned earlier, lives in a symbiosis with the bobtail squid *Eprymna scolopes*, where it colonizes light organs used to camouflage the squid at night (Table 2) [81]. The formation of the light organs is regulated by symbiont-induced apoptosis. Under s-µ*g* conditions in HARVs, genes related to extrinsic/receptor-mediated and intrinsic/stress-induced apoptosis were expressed earlier and to a greater extent than 1*g*, especially those for initiator and executioner caspases. However, the increases in caspase activity could be compensated with caspase inhibitors, offering a strategy for maintaining animal–microbial homeostasis during spaceflight.

The gene expression in *Caenorhabditis elegans* was investigated in a series of experiments on the ISS: the results showed a downregulation of genes related to longevity [96] or metabolism and protein expression [89]. The most recent results indicate that the expression is partially regulated epigenetically: an overexpression of several genes was observed in histone deacetylase (had)-4 mutants as compared to wild types, indicating that in a wild-type organism, excessive expression of certain genes is epigenetically suppressed to the extent of a de facto downregulation, while in the had-4 mutant, the overexpression occurs unhampered (Table 2) [90].

*Caenorhabditis elegans* exposed to s-µ*g* by applying an RCCS showed alterations in the intestines [91]: both internal lumen and permeability were altered due to oxidative stress, even though a number of related genes were globally upregulated. Overexpression of superoxide dismutase 2 (*SOD2*), one of the genes in the intestines effectively protected the nematodes from µ*g*-induced oxidative stress damage to the intestines. In addition, specifically in the intestines, expressional adaptations of the insulin signaling pathway were found [92].

The influence of prolonged µ*g* on the heart was investigated employing the fruit fly *Drosophila melanogaster* (Table 2) [87]. Fly eggs were flown to the ISS, where the larvae hatched and developed into adult flies before returning to the ground. The hearts were smaller and weaker, which correlated to the reduced sarcomeric and extracellular matrix gene expression. Furthermore, the upregulation of proteasome subunit genes suggested an elevated proteostatic turnover. A follow-up experiment confirmed an elevated number of (likely dysfunctional) proteasomes.

With long-term spaceflight on the horizon, important questions arise regarding its impact on human fertility and the influence of the hormonal cycle on overall well-being. In the past, experiments with female mice were rather short-term or the results were ambiguous due to the fact that the animals returned to the ground alive and therefore experienced the stress of re-entry and the onset of gravity. In their experiment, Hong et al. investigated the estrous cycle of mice for the first time after prolonged exposure to µ*g* on the ISS with subsequent scarification of the animals while still staying in orbit [83]. The space-flown mice showed no signs of stress and estrous cycle discontinuation (indicating that they remained fertile) and no difference in the expression of genes of the key enzymatic steps of steroidogenesis or mitochondrial cholesterol uptake expression. The authors suggest using the estrous cycle state as a covariate, such as age and weight, due to its potential impact on the general state of experimental animals and, therefore, the interpretation of experimental data.

The immune system of mice was investigated with the Multiple Artificial-gravity Research System (MARS) aboard the ISS [94]. The thymi of µ*g* exposed mice showed significantly different gene expression patterns compared to ground controls and onboard 1*g* controls, suggesting that artificial gravity can partially mitigate the adverse effects of µ*g* but not wholly. Among the downregulated genes were those responsible for cell cycle control and chromosome organization, indicating that µ*g* leads to a smaller number of mitotic cells, which is also reflected in the smaller weight per thymus (both absolute and compared to body weight). The findings were confirmed by a second independent experiment. Analyzing the spleens and lymph nodes of the same animals employing whole transcript cDNA sequencing, Horie et al. found that not only immune-related processes are altered by µ*g*: various gene ontology (GO) terms related to the production of erythrocytes were reduced in µ*g* as compared to ground controls [95]. Again, onboard 1*g* controls showed a lower reduction, suggesting that artificial gravity can mitigate only some of the impacts of spaceflight. Further analysis showed that the expression of two transcription factors (*GATA1* and *Tal1*), which promote the expression of various genes controlling erythrocyte development, was downregulated. However, immunostaining of the spleens did not show a considerable influence of the downregulation on erythrocyte cell number and distribution. The gene expression in lymph nodes was not influenced by spaceflight.

The expression of complement component C3 was investigated in Iberian ribbed newts (*Pleurodeles waltl*) exposed to s-µ*g* via RPM [97]. C3 is a central component of the complement system, which is involved in immediate defense against microbes and regulates immunological and inflammatory processes. C3 is also highly conserved. Exposure to s-µ*g* alone did not alter C3 expression in *Pleurodeles waltl*; however, in combination with other space-related stressors, a reduction was observed. Analysis of mice subjected to HU showed also that C3 expression in the liver (the main source of C3) did not change as compared to the control [97].

### 3.4. Effects of Microgravity on the Eye and Brain

The eye, especially the retina, represents one of the most sensitive and critical human body tissues. It does, therefore, not come as a surprise that astronauts returning to Earth after a long-term stay onboard the ISS develop a complex of alterations known as SANS. Notably, NASA reported more than a decade ago that approximately 60% of 300 active astronauts were affected by such neuro-ophthalmic alterations [98]. A leading hypothesis implies that weightlessness-induced cephalad fluid shifts elevate the intracranial pressure (ICP), which may impact the observed ocular structural alterations. However, increased ambient CO_2_ levels on the ISS may also contribute to the outcome.

As a reflection on the ongoing and tireless interest in neuro-ocular changes induced by µ*g*, a growing number of studies addressing this issue, including the mechanisms underlying SANS, have therefore been initiated, and several exciting papers on gene regulation in the eye and brain after exposure to either r- and s-µ*g* conditions have been published during the last five years. These findings, which were obtained in relevant cells with eye origin and animal models (preferably rodents), either subjected to altered gravity conditions on Earth or in space, as well as in astronauts travelling onboard the ISS in space, will be summarized in the following sections and in Table 3.

#### 3.4.1. Effects of Microgravity on Cells with Eye Origin

In a recent paper, an Italian team set out to further explore the pathogenesis underlying SANS. Specifically, the molecular and cellular effects induced in human adult retinal pigment epithelium (ARPE-19) cells following incubation for three days in µ*g* onboard the ISS were investigated [99]. No changes in viability or apoptosis were observed during the µ*g* phase. However, in alignment with previous studies [14], cytoskeletal remodeling was detected following exposure to µ*g* conditions. Notably, Cialdai and co-workers found a dramatic change in the vimentin network, exemplified by the redistribution of vimentin from the surface to the perinuclear regions in the ARPE-19 cells cultured for three days onboard the ISS. This altered distribution of vimentin may thus indicate a change in the cellular shape and the inter-cellular interaction capabilities.

Further analysis revealed that spaceflight ARPE-19 cells contained structures resembling aggresomes implying that µ*g* may be directly linked to an alteration in protein processing in ARPE-19 cells. Interestingly, the observed morphological changes in the ARPE-19 cells cultivated onboard the ISS were associated with significant alterations in the transcriptome profile. Of 23.556 genes analyzed, more than 5.500 were differentially expressed after the spaceflight compared to ground controls (Table 3). Prediction analysis showed that the ISS environment significantly affected approximately 100 pathways, of which the most significantly impacted were related to the cellular response to space environment adaptation/damage [99]. Gene ontology (GO) analysis revealed that the incubation of ARPE-19 cells in r-µ*g* impacts on several critical cellular mechanisms, including the response to unfolded proteins and ion binding, consistent with cell dysfunction adaptation (Table 3) [99].

The study also tried to estimate the number of active micro (mi)RNAs and deregulated long non-coding (lnc)RNAs [99]. Of 366 screened miRNAs, 19 displayed differential downregulation of target genes. More than 250 lncRNAs were deregulated in ARPE-19 cells cultured onboard the ISS (Table 3). Finally, Cialdai and co-workers investigated the role of coenzyme Q10 (CoQ10) treatment on gene expression in ARPE-19 cells [99]. CoQ10 is a well-known antioxidant with antiapoptotic abilities. A total of 153 differentially expressed genes (DEGs) were identified in CoQ10-treated cells incubated onboard the ISS compared to the untreated control. Interestingly, 22 pathways were significantly affected, including deregulation of protein processing in the endoplasmic reticulum, mitophagy, TGF-beta signaling, Hippo signaling, p53 signaling, and the senescence pathway, clearly suggesting that CoQ10-based countermeasures against SANS should be further investigated (Table 3) [99].

#### 3.4.2. Effects of Microgravity on Eye in Animals

Going to space to conduct biological experiments on the ISS is a costly and time-consuming process, and hence an option with minimal access for the broader field of researchers involved in µ*g* research and space medicine. To advance biomedical research on the physiological effects of the space environment, NASA 2014 launched the rodent research (RR) project. Hence, with bio-banked tissues from the RR-1 spaceflight, NASA’s GeneLab platform provides DNA, RNA, and protein samples for epigenomic, transcriptomic, and proteomic analysis [100]. Disseminating these data without restriction to the scientific community may thus be an important resource for multi-omics investigations. Despite the limited resources for conducting biological experiments in space, fortunately, several animal studies are performed aboard the ISS each year. For example, concerning the impact of µ*g* on the eye and brain, four different studies based on mice, launched from the Kennedy Space Centre (KSC) on three separate SpaceX rockets (SpaceX-4, 9, and 12) to the ISS for 35- or 37-day missions, have been published during 2017–2022 (Table 3).
cells-12-01043-t003_Table 3Table 3Changes in gene regulation of (i) eye-related cells subjected to r-µ*g*, (ii) eye tissue isolated from animals exposed to microgravity conditions, or (iii) healthy human volunteers exposed to head-down tilt.SourceKindGene RegulationMicrogravityReference(i) Cell lineARPE-19Analysis of 23.556 targets: more than 5.500 differentially expressed genesApprox. 100 pathways were affected, including cellular response to space environment adaptation/damageGO analysis revealed that r-µg impacts several critical cellular mechanisms, including response to unfolded proteins and ion bindingOf 366 screened miRNAs, 19 displayed differential downregulation of target genesMore than 250 lncRNAs were deregulatedSpaceflight to the ISS,3 d[99](ii) MiceC57BL/6Male9-week-oldSeveral pathways involved in inflammation, cell repair, cell death, and metabolic stress were significantly changedAnalysis of regulated protein expression revealed changes in retinal protein expression related to immune response, metabolic function, and cellular structureSpaceflight to the ISS,35 d[101](ii) MiceC57BL/6Male10-week-oldPathways involved in, e.g., inflammation, cell death, and metabolic stress, were significantly alteredSpaceflight to the ISS,35 d[102](ii) MiceC57BL/6Male10-week-old600 DEGs were detected in the retinas of mice flownGenes related to the phototransduction pathway and visual perception were enrichedOf 75 genes associated with retinitis pigmentosa, 12 were differentially expressed.For diabetic retinopathy, only one gene was differentially expressedNo differentially regulated genes were detected for retinal-detachment disease-associated and AMDA number of transcription factors, including *Cazs1*, *Kdm4a*, *Kdm4b*, and *Kdm6b*, were differentially expressedSpaceflight to the ISS,35 d[103](ii) MiceC57BL/6Female16-week-oldDifferential methylations of many genes were observedRetinal cell homeostasis was disturbed: several genes involved in inflammation, oxidative stress, tissue remodelling, mitochondrial function, and angiogenesis were impactedReduced gravity decelerates the epigenetic clock in the mouse retinaSpaceflight to the ISS,37 d[104](ii) MiceStrain N/AFemale13 gene targets of three miRNAs were identifiedAg-treatment only altered the expression of *Zcchc9*Hindlimb unloading[105](iii) Human healthy volunteersEight male subjects*Analysis of one-carbon pathway polymorphisms in *MTRR* and *SHMT1* did not reveal any significant changes1 h settings of seated, HDT, and HDT + CO_2_[106](iii) Human healthy volunteersSix men and five women **MTRR* 66G and *SHMT1* 1420C alleles were associated with a greater increase in total retinal thickness during30 d of 6° HDT bed rest with 0.5% CO_2_ exposure[107]Abbreviations: Age-related macular degeneration (AMD); human adult retinal pigment epithelium cells (ARPE-19); days (d); differentially expressed genes (DEGs); gene ontology (GO); head-down tilt (HDT); hours (h); International Space Station (ISS). * Blood samples and non-invasive analysis of eye and/or brain.


The first paper, spearheaded by Xiao Mao [101], investigated the impact of spaceflight and artificial gravity on the mouse retina. First, changes in protein expression profiles and oxidative stress-related apoptosis were examined in nine-week-old male C57BL/6 mice following a 35-day stay on the ISS. The mice were launched to the ISS by SpaceX-9 at the KSC in 2016. One group of the spaceflight mice was housed under µ*g* conditions, whereas another group was maintained in a centrifugal habit unit facilitating a 1*g* artificial condition [101]. Quantitative analysis based on terminal deoxynucleotidyl transferase dUTP nick-end labelling (TUNEL) analysis of ocular tissue demonstrated that µ*g* significantly induced apoptosis (up to 64%) in retinal vascular endothelial cells (VEC) compared to the ground control group. Next, proteomics’ analyses revealed that a number of pathways involved in inflammation, cell repair, cell death, and metabolic stress were significantly changed in spaceflight mice compared to the ground controls (Table 3).

Furthermore, significant changes in regulated protein expression were observed in the µ*g* group with the artificial 1*g* group onboard the ISS. The authors concluded that spaceflight induces apoptosis of VEC and changes in retinal protein expression related to immune response, metabolic function, and cellular structure. Notably, the study demonstrates that artificial gravity induced on the ISS may mitigate some of the µ*g*-induced alterations [101].

The subsequent study, also performed by Mao and co-workers [102], aimed to investigate the effects of spaceflight and re-entry to 1*g* on the integrity and structure of the retina and blood–retina barrier (BRB) in the eye. Ten-week-old male mice were launched to the ISS on SpaceX-12 at the KSC in 2017. After a 35-day spaceflight, the animals were returned alive to Earth, and changes in protein expression profiles were assessed. In support of the previous findings from the research group, significant apoptosis in the retina and retinal vascular cells was observed compared to control groups. The study also provides strong evidence that spaceflight impacts BRB integrity [102]. This notion is primarily based on findings showing increased expression of aquaporin-4 (AQP4) and platelet endothelial cell adhesion molecule-1 (PECAM-1) in the spaceflight mice group compared to the control group (Table 3). In addition, a decrease in expression of the BRB-related tight-junction protein, Zonula occludens-1 (ZO-1), was also observed. Given that the BRB is compromised by spaceflight, it is a surprise that the intraocular pressure (IOP) was significantly lower in the post-flight measurement compared to that performed in pre-flight and not the opposite [102]. The authors provide several possible explanations for this discrepancy, including the notion that the 35-day flight may not be sufficient to elicit elevations in IOP or the limitation of utilizing mice to study the effect of spaceflight on human function due to the lack of a cephalad fluid shift [102].

As in the previous study performed by Mao et al. [101], proteomics data revealed that a number of pathways involved in, e.g., inflammation, cell death, and metabolic stress, were significantly altered in the spaceflight compared to the control group (Table 3). However, a limited overlap of significant differentiating proteins and pathway between the 2016 and 2017 flights was observed, possibly reflecting differences in the composition of the eye samples tested.

In the third animal-based study in µ*g*, Overbey and co-workers continued their efforts to investigate whether the space environment triggers oxidative damage on ocular structures and to further characterize the gene expression profiles of the mouse retina exposed to r-µ*g* during a 35-day mission on the ISS [103]. Similar to the previous paper, the ten-week-old male mice used in this study were launched to the ISS on SpaceX-12 at the KSC in 2017 and returned to Earth alive. RNA sequencing of the isolated ocular tissues resulted in detecting 600 DEGs in the retinas of the mice flown onboard ISS (Table 3). Genes related to the phototransduction pathway and visual perception were enriched in the group of DEGs. The authors also sought to determine whether any DEGs from the spaceflight sample were differentially expressed in common retinal diseases. Interestingly, of the 75 genes associated with retinitis pigmentosa, 12 were differentially expressed in murine spaceflight retinas. For diabetic retinopathy, only one gene was differentially expressed, whereas retinal-detachment disease-associated and age-related macular degeneration (AMD) genes were not differentially regulated (Table 3) [103].

A number of transcription factors, including *Cazs1*, *Kdm4a*, *Kdm4b*, and *Kdm6b*, were also found to be differentially expressed, suggesting that spaceflight changes chromatin organization (Table 3). Finally, micro-computed tomography (micro-CT) and immunofluorescence analysis revealed that spaceflight reduces the thickness of multiple layers and increases oxidative stress. These results suggest that retinal functioning may be challenged by exposure to extended periods of µ*g*, resulting in ocular damage [103].

Last but not least, Chen and co-workers investigated the impact of r-µ*g* in 16-week-old female C57BL/6J mice transported to the ISS by SpaceX-4 in 2014 for a 37-day spaceflight mission [104]. The study’s primary aim was to examine whether spaceflight triggers epigenomic and transcriptomic reprogramming in the retina. Second, it was investigated whether reduced gravity alters the epigenetic clock. In this study, the retinal samples were collected in orbit and frozen for later analysis. After careful sectioning of the retina, DNA and RNA species were isolated, and the DNA methylome and transcriptome were analyzed by deep sequencing. In general, spaceflight induced differential methylations of many genes, whereas fewer DEGs were identified (Table 3). The fact that several genes involved in inflammation, oxidative stress, tissue remodelling, mitochondrial function, and angiogenesis were impacted by µ*g* suggests that retinal cell homeostasis was disturbed during spaceflight. The findings also suggest that pathways involved in retinal diseases, such as macular degeneration, were significantly affected [104]. However, on the plus side, the results indicated that reduced gravity decelerates the epigenetic clock in the mouse retina, implying that the mice onboard ISS show a younger biological age compared to the mice housed on Earth. In summary, this study is the first to investigate the effects of spaceflight on both DNA methylome and transcriptome in mouse retinal tissue [104].

Finally, a recent paper investigated the effect of inhibiting three spaceflight-associated miRNAs by small molecule inhibitors (antagomirs (Ag)) in mice before exposure to simulated spaceflight conditions (µ*g* and radiations) [105]. Hindlimb unloading was used to simulate µ*g*, and Ag-treated mice received Ag every three days. A total of 13 gene targets of the 3 miRNAs were identified. However, the eye Ag-treatment only altered the expression of one gene (*Zcchc9*) (Table 3). Inhibition of miRNAs in a preclinical model of simulated spaceflight may provide a basis for analyzing inhibition of biological impairment [105].

#### 3.4.3. Effects of Microgravity on Eye and Brain in Human Subjects

Laurie and co-workers applied the spaceflight equivalent of 6° head-down tilt (HDT) to assess possible underlying mechanisms of ocular alterations [106]. The experiment was conducted on eight male subjects exposed to three 1-h settings (seated, HDT, and HDT with 1% inspired CO_2_ (HDT + CO_2_)). Various methods, including cerebral and ocular ultrasound, optical coherence tomography (OCT) scans of the macular and optic disc, intraocular pressure (IOP), non-invasive intracranial pressure (nICP), and translaminar pressure difference (TLPD = IOP–ICP) were executed. The authors found enlarged IOP, optic nerve sheath diameter, and choroid thickness during the acute headward fluid-shift. Hence, the findings may imply that exposure to acute mild hypercapnia during HDT did not increase physiological factors suggested as being involved in ocular changes during spaceflight. Moreover, the combination of HDT with a CO_2_ level twice as high as on the ISS did not cause ocular structural or functional alterations. Analysis of one-carbon pathway polymorphisms in *MTRR* and *SHMT1* did not reveal any significant changes, probably due to the low number of subjects in the groups (Table 3).

A recent paper by Zwart and co-workers [107] further investigated one-carbon pathway genetics in a cohort study of 11 healthy volunteers (5 women and 6 men). The paper analyzed whether one-carbon metabolic pathway polymorphisms are associated with the development of optic disc oedema during head-down tilt bed rest with CO_2_ exposure. Interestingly, it was shown that more *MTRR* 66G and *SHMT1* 1420C alleles were associated with a greater increase in total retinal thickness during 30 days of 6° HDT bed rest with 0.5% CO_2_ exposure (Table 3). The B-vitamin status was a contributing factor. These interesting results may thus play an important role in understanding the variability in the size of optic disc oedema detected during bed rest studies and spaceflights and thereby advance the development of countermeasures [107].

### 3.5. Effects of Microgravity on Endothelial Cells

Astronauts returning from space often suffer from endothelial dysfunction. This has made the study of endothelial cells (ECs) in µ*g* one of the focal points of gravitational research. As mechanosensitive cells, ECs undergo morphological and functional changes when exposed to µ*g* conditions.

Barravecchia et al. [108] performed high-throughput RNA sequencing of human microvascular ECs to identify the genome-wide effects of µ*g* during the ENDO campaign to the ISS in 2015. They found that 32 gene sets of the Hallmark collection were deregulated due to µ*g* exposure (19 up- and 12 downregulated) activating pathways for metabolism and a pro-proliferative phenotype. Transcriptomics also demonstrated opposing effects of µ*g* and space radiation.

The SPHINX project revealed that *TXNIP* was the most overexpressed transcript in ECs after spaceflight [109]. Cazzaniga et al. [110] demonstrated a *TXNIP* overexpression in human umbilical vein endothelial cells (HUVECs) after 10 days on the RWV, whereas no modulation of *TXNIP* was detected after four days. The authors also found a temporary upregulation of *HSPA1A* (HSP70) in HUVECs after four days on an RWV before it returned to baseline after 10 days [110]. The meta-analysis-assisted detection of gravity-sensitive genes in HUVECs by Liang et al. [111] revealed that the expression level of the prostaglandin transporter gene *SLCO2A1* decreased in response to µ*g*.

Li et al. [112] reported in their autophagy studies with HUVECs in µ*g* that there was no significant difference in *TP53* transcription between the s-µ*g* group and the control group. In contrast, clinorotation decreased the protein level of p53 in HUVECs, suggesting an effect of clinorotation on post-transcriptional modifications of p53.

Another vital area of research in molecular cell biology in recent years has been microRNA. Using the RWV, Pan et al. [113] demonstrated for the first time that s-µ*g* can alter the expression of some microRNAs in HUVECs. They further observed that miR-27b-5p might protect HUVECs from apoptosis on the RWV by targeting the zinc fingers and homeoboxes 1 protein (ZHX1). Xu et al. [114] published that miR-22 was upregulated, and its target genes *SRF* and *LAMC1* were downregulated at mRNA levels in HUVECs exposed to an RCCS. In addition, Kasiviswanathan et al. [115] analyzed the interactome of miRNAs of HUVECs cultured on a clinostat. They reported that miRNAs miR-496, miR-151a, miR-296-3p, miR-148a, miR-365b-5p, miR-3687, miR-454, miR-155-5p, and miR-145-5p effectively influenced the cell proliferation and vascular functions of HUVECs in s-µ*g.*

Zhao et al. [116] studied apoptosis of choroidal vascular ECs (CVECs) in s-µ*g* on an RCCS for the first time. They found that the expression of *BAX*, *CASP3*, and *CYP2D6* increased significantly after 24 and 72 h in µ*g*, while the expression of the anti-apoptotic *BCL2* decreased.

Dittrich et al. [117] used immortalized human vascular ECs (EA.hy926) for long-term studies of tube formation on the RPM. After 35 days, they observed upregulation of *CXCL8* and *FN1* in spheroids formed during random positioning. Furthermore, using the same cell line, Krüger et al. [118] showed that a number of genes (*TIMP1*, *IL6*, *CXCL8*, *CCL2*, *B2M*) were differentially regulated in adherent and spheroid cell populations after 7 and 14 days of s-µ*g* on the RPM. While most genes studied were downregulated in adherent cells, several were upregulated or not regulated in spheroid cells. Li et al. [119] cultured EA.hy926 cells aboard the SJ-10 satellite for 10 days before RNA profiling of supernatant-derived exosomes. *ACTB*, *PGK1*, *HSPA8*, *RPL7A*, and *FTH1* were the five most upregulated protein-coding genes in exosomes from EA.hy926 cells cultured in space compared to those on the ground.

Kong et al. [120] used human peripheral blood-derived endothelial progenitor cells exposed to a newly developed Gravite^®^ device to simulate µ*g*. They measured an increased expression of the angiogenic genes HIF1A and NOS3 after 12- and 24-h exposure to the Gravite^®^ device. Afterwards, the expression levels of HIF1A and NOS3 decreased over time.

In a recent paper, Zhao and co-workers showed the effects of s-µ*g* on apoptosis of CVECs, thereby adding these cells to the growing list of microvascular endothelial cells displaying alterations in gene expression following incubation under µ*g* conditions [116]. VECs exert a fundamental role in tissue homeostasis by orchestrating vessel and blood circulation [121]. Notably, previous studies have suggested that dysfunction of VECs may explain the cardiovascular deconditioning observed in astronauts exposed to µ*g* [116]. This idea has been supported by several experiments showing that exposure to s-µ*g* impacts critical cellular structures, including the cytoskeleton and mitochondria homeostasis, and gene expression affecting cellular functions, such as apoptosis [122,123,124,125,126,127]. To wrap up the conclusions from these findings in a nutshell, VECs may thus sense the reduced gravitational force, thereby inducing cytoskeleton changes, which via activation of so-called secondary messengers, results in various gene responses that eventually may trigger apoptosis [116].

To explore whether CVECs are perceptive in the same way to gravitational alterations, human CVECs were subjected to s-µ*g* employing an RCCS for three days. A prominent decrease in F-actin and the filaments’ sparse or discontinuous appearance was observed after cultivation under s-µ*g* conditions. Notably, flow cytometry showed an increased number of apoptotic CVECs in the s-µ*g* group compared with the 1*g* group [116]. Concomitantly, mRNA and protein levels of caspase3, bax, cytochrome C, p-AKT, p-PI3K, and Bcl2 were altered, suggesting that the Bcl-2 apoptosis pathway and the PI3K/AKT pathway participate in µ*g*-induced damage of CVECs. Furthermore, these alterations were accompanied by ultrastructural changes, including chromatin condensation, mitochondria vacuolization, shrinking of the cell body, and the appearance of apoptotic bodies [116]. Since the dysfunction of CVECs has been implicated in choroidal thickening, which may have an impact on intraocular pressure and visual function, the findings by Zhao and co-workers may provide further knowledge about the underlying mechanisms of µ*g*-induced changes in the eyes of space travelers and, maybe most importantly, advance the development of countermeasures [116]. An overview of all results is given in Table 4.

### 3.6. Microgravity Affects the Immune System in Space

Various immunological alterations have been observed in space crews during and after spaceflight [2].

It has been almost 40 years since Augusto Cogoli showed that spaceflight reduces lymphocyte reactivity to mitogens. Cultures of lymphocytes, purified from blood samples drawn from crew members before and after flight, were exposed to mitogens. Activation was measured by the incorporation of labeled thymidine or uridine into DNA or RNA, respectively. A total of 41 astronauts and 12 cosmonauts were tested. These data were published in [128,129]. This topic was the subject of the authors’ investigations during the flight of Spacelab 1 from 28 November to 8 December 1983 [128].

Spaceflight-associated immune system dysregulation is a severe health problem in humans in space and negatively affects further space exploration to the Moon, Mars, and outer space. Space travelers face different forms of stress influencing their immune system. A paper published in January 2023 showed that RPM exposure altered the Nuclear Factor κB signaling pathway and affects murine dendritic cells (DC) and their function [130]. Splenic DC or Flt-3L-differentiated bone marrow DC (BMDCs) were exposed to the RPM. S-µ*g* reduced BM-conventional DC (cDC) as well as splenic cDC activation/maturation phenotype changes [130]. BMDCs exhibited a decreased production of pro-inflammatory cytokines when exposed to the RPM. The authors demonstrated that non-matured RPM-exposed BMDCs exhibited a more immature phenotype, compared to the control BMDCs. These data correlated with an impaired ability of BMDCs to express pro-inflammatory cytokine transcripts as shown in Figure 3A of [130]. A deregulation of DC function is likely responsible for inducing immune deregulation during spaceflight [130].

A recent study investigated cell-free mitochondrial DNA (cf-mtDNA) in the blood plasma of 14 astronauts [131]. The gene expression analysis of peripheral blood mononuclear cells (PBMC) indicated a significant elevation of inflammation, oxidative stress, and DNA damage markers, a finding supporting the hypothesis that cell-free mitochondrial DNA abundance might be a biomarker of stress or immune response in relation to spaceflight [131].

Deep space exploration needs further studies to predict astronauts’ health risks [132]. Applying deep, error-corrected, targeted DNA sequencing, Brojakowska et al. [132] focused on somatic mutations in clonal hematopoiesis (CH)-driver genes in PBMC isolated from the de-identified blood samples of 14 Space Shuttle astronauts. They found 34 nonsynonymous mutations in 17 CH-driver genes, with the most prevalent mutations in TP53 and DNMT3A. Therefore, future retrospective and prospective investigations of their clinical relevance are necessary [132].

Earlier, the NASA Twins study revealed a significant increase in the proportion of cf-mtDNA inflight and the analysis of post-flight exosomes [133]. These data suggested that cf-mtDNA levels can be a potential biomarker for stress or immune system responses related to spaceflight conditions [133]. In addition, a re-analysis of the landing data of the Twins study revealed signs of muscle regeneration rather than a detrimental inflammatory response [134]. Further analyses of the NASA Twins data also comprised specific analytes associated with fatty acid metabolism [135]. This study showed that cellular lipid metabolism could be responsive and dynamic to spaceflight. In addition, the data revealed mid-flight spikes in the expression of selected genes, indicating transient responses to specific insults during the flight subject’s stay on the ISS [135].

Human Jurkat T cells react and adapt very rapidly to altered gravity conditions. Differentially expressed gene transcript clusters (TCs) in Jurkat T cells in µ*g* provided by a suborbital ballistic rocket flight were compared with TCs expressed as a reaction to 2D clinorotation as well as to 9× *g* centrifuge experiments and rigorous controls for excluding other factors of influence than gravity [136]. During 5 min of either flight-induced µ*g* or clinorotation 11 TCs were significantly altered. It was concluded that less than 1% of all examined TCs displayed the same response in 2D clinorotation and flight-induced µ*g*. Contrary, 38% of differentially regulated TCs identified during the hypergravity phase of the suborbital ballistic rocket flight could be verified with 9× *g* ground centrifugation. Therefore, it is evident that the initial trigger of gene expression response to µ*g* requires less than 1 s reaction time [136].

Furthermore, data from parabolic and TEXUS rocket flight missions revealed that hypoxia inducible factor 1 (HIF-1) and HIF-1-dependent transcripts were differentially expressed in altered gravity conditions [137]. The HIF-1-dependent gene expression was adapted after 5 min r-µ*g*. *PDK1* was detected to be highly responsive to gravitational changes in human U937 myelomonocytic cells and Jurkat T cells. Targeting HIF-1 might be an effective countermeasure to prevent the immune system from weakening during spaceflight [137]. A further study measured gene expression and 3D chromosomal conformational changes in human Jurkat T cells during parabolic flight maneuvers and a suborbital ballistic rocket flight [138]. The authors propose that gravitational forces rapidly influence the cell membrane and that they are mechanically transduced via the cytoskeleton to the nucleus. This leads to 3D chromosomal conformational changes, resulting in region-specific differential gene expression [138]. Using the same flight opportunities, the expression of oxidative stress-related pathways was fast and strong in human myelomonocytic U937 cells but followed by a rapid and severe counter-regulation. Interestingly, oxidative stress-related genes in human Jurkat T cells were not significantly altered [139].

Human Jurkat T lymphocytic cells were studied during a parabolic flight and a Technologische EXperimente Unter Schwerelosigkeit (TEXUS) sounding rocket mission [140]. Applying the Affymetrix GeneChip^®^ Human Transcriptome Array 2.0, the authors found an extensive and rapid change in gene expression associated with regulatory RNAs. They concluded that human cells are equipped with a robust and efficient adaptation potential when exposed to altered gravity conditions [140].

A recent study demonstrated that r-µ*g* (spaceflight) and s-µ*g* significantly decreased macrophage quantity and differentiation and induced metabolic reprogramming with alterations in gene expression profiles [67]. Furthermore, the rat sarcoma (RAS)/extracellular signal-regulated kinase (ERK)/NFκB was detected as a significant µ*g*-regulated pathway. This was also the case for the p53 pathway. These results suggest novel molecular targets to prevent macrophage differentiation deficiency in µ*g* [67].

An in vivo mouse study reported the effects of spaceflight (ISS project, 35 d in orbit) on secondary lymphoid organs at the molecular level [95]. Whole-transcript cDNA sequencing (RNA-Seq) analysis of the spleen revealed that erythrocyte-related genes regulated by *GATA1* were significantly reduced in ISS-flown vs. ground control mice. In addition, the *GATA1* and *Tal1* mRNA expression was downregulated. These reductions were not entirely alleviated by 1*g* exposure on the ISS, advocating that the combined effect of space environments apart from µ*g* could alter gene expression in the spleen. This study showed that the unique ISS environment affects the homeostatic gene expression of the spleen in mice [95].

Female C57BL/6J mice flew with STS-135 in July 2011 as part of the Commercial Biomedical Testing Module-3 (CBTM-3) payload [141]. Despite decreases in splenic leukocyte subsets, elevations in reactive oxygen species (ROS)-related activity could be measured. The functional analysis of gene expression and metabolomic profiles showed that the functional changes are not due to oxidative or psychological stress. No corresponding increase in genes related to ROS metabolism was detectable. An elevation in expression profiles related to fatty acid oxidation with decreases in glycolysis-related profiles. These findings suggest a link between immune function and metabolism in spaceflight [141].

To prepare for future spaceflight experiments and to support findings from r-µ*g*, studies under conditions of s-µ*g* were performed to increase the current knowledge about genomic changes of the immune system in µ*g* in vitro and in vivo.

A recent bioinformatics study investigated how s-µ*g*, created by an RPM, influences circulating and tissue-resident T cells [142]. The 3D cell culture attenuates the effects of RPM exposure on the T-cells’ transcriptome and nuclear alterations compared to 2D cell culture [142]. This study is the first to apply 3D models under the effects of s-μ*g* showing that T cells residing in tissue are less affected by the RPM than circulating T cells in the periphery.

Another study focused on the effects of clinorotation on macrophage phenotypes M0, M1, and M2 [143]. µ*g* results in a decrease in TNF-α expression and an increase in IL-12 and VEGF expression. IL-10 was also significantly increased in M1 and M2, but not M0 macrophages. These data provide new knowledge about the macrophage phenotypic function in µ*g* [143].

Spatz et al. [144] reported the results of PBMCs exposed to an RVW. High-parameter mass cytometry revealed that RWV exposure of PBMCs dampens important innate and adaptive immune cell effector functions. An increase in the suppressive immune cell function was detectable. RWV exposure of PBMC results in a multi-cellular immunosuppressive response that may contribute to the impairment of the immune system and the defense against pathogens [144]. A further study investigated transcriptional and post-transcriptional regulations based on gene and miRNA expression profiles in human peripheral blood lymphocytes exposed to the RWV [145]. Two hundred and thirty dysregulated TF-miRNA (transcription factor and microRNA) feed-forward loops (FFLs) were identified in s-µ*g*. Associations of RWV exposure with dysfunctions of multi-body systems and tumorigenesis were reported [145].

The application of Radio Electric Asymmetric Conveyer (REAC) technology is a new approach to counteract the loss of T-cell activity [146]. RPM exposure and REAC treatment confirmed the T-cell activation recovery and improved the gene expression of *IL2* and *IL2Rα*. In addition, there is evidence that REAC technology could contribute to understanding T-cell growth responsiveness in space [146].

The adrenergic receptor is an essential regulator of the immune system. The impact of µ*g* on the adrenergic system is not yet understood [147]. The authors studied the synergistic effects of isoproterenol, radiation, and RVW exposure on non-stimulated PBMC. The results revealed significant synergistic effects on the expression of the β2-adrenergic receptor gene (*ADRB2*). Radiation alone increased *ADRB2* expression, and cells incubated in µ*g* had more DNA strand breaks than cells incubated in normal gravity. Isoproterenol prevented most of the µ*g*-mediated effects [147].

Paul et al. [148] used an in vivo model for spaceflight simulation (mice, 21d, hindlimb unloading (HLU) combined with continuous low-dose gamma irradiation) to study immune and hematological systems at 7-days post-exposure. Among others, the spleens were analyzed by whole transcriptome shotgun sequencing (RNA-sequencing). Murine Reactome networks indicated that most spleen cells displayed DEGs involved in signal transduction, metabolism, cell cycle, chromatin organization, and DNA repair. DEG analysis of the spleen revealed expression profiles associated with inflammation and dysregulated immune function persist to 1-week post-simulated spaceflight. This work showed differential immune and hematological outcomes 7d post-exposure [148].

Finally, Zhu et al. [149] introduced a ground-based zebrafish disease model of µ*g* using an RCCS. RNA seq analysis revealed that s-µ*g* significantly influenced the retinoic-acid-inducible gene (RIG)-I-like receptor (RLR) and the Toll-like receptor (TLR) signaling pathways. Simulated-µ*g* hampered the TRIM25-mediated K63-linked ubiquitination of RIG-I and attenuated the antiviral innate immune responses. The TRIM25 function–induction positive feedback loop is essential in antiviral immunity, and the reduced TRIM25 expression under s-µ*g* interferes with the feedback loop. This model improves the current knowledge about host antiviral immunity in s-µ*g* [149]. An overview of all results is given in Table 5.

### 3.7. Effects of Microgravity on Cartilage

The only study conducted in r-µ*g* during spaceflight was completed by Fitzgerald et al. [150] (Table 6). The authors exposed mice to 30 days µ*g* on board a BION-M1 capsule and subsequently analyzed articular and sternal cartilage samples for gene expression changes. In articular cartilage, they found a total number of 47 differentially expressed genes (10 upregulated, 37 downregulated) by more than 2-fold, 17 of which were coded for proteins involved in structural cartilage ECM or joint pathology. They measured 30 upregulated and 35 downregulated genes in sternal cartilage compared to ground controls. Interestingly, the differentially expressed genes in both cartilage tissues were regulated in opposite directions. The authors proposed that this may be due to the different biomechanical environments of the two samples [150].

The other studies were conducted under conditions of s-µ*g* on devices such as the RCCS or the RPM. Ma et al. [151] first generated engineered meniscus tissues by seeding meniscus fibrochondrocytes from female and male donors onto a cylindrical type 1 collagen scaffold and incubating them for two weeks in a TGF-b3-rich chondrogenic medium. The resulting meniscus models were then cultivated either as static controls or on an RCCS employing slow-turning lateral vessels for a further three weeks. Subsequently, the samples were subjected to RNA-seq and quantitative reverse transcription polymerase chain reaction (qRT-PCR). The resulting data were further stratified based on the *COL10A1* expression levels. Female donors could be classified into low and high responders, while male donors remained in one group. Remarkably, genes related to osteoarthritis (*BMP8A*, *CD36*, *COL10A1*, *COL9A3*, *FGF1*, *IBSP*, *IHH*, *MMP10*, *PHOSPHO1*, *S100A1*, and *SPP1*) were significantly upregulated only in the female high-responder group, but not in the two others [151].

Further analysis of the same samples showed that the response to mechanical loading and unloading occurred in a sex-dependent manner (Table 6) [152]. A total of 93 regulated genes were unique for men, 163 for women, and 94 genes were regulated in both sexes. Meanwhile, in men and women, the most enriched KEGG pathway was HIF signaling; in women, it was followed by glycolysis/gluconeogenesis, carbon metabolism biosynthesis of amino acids, and steroid biosynthesis. In men, the top five KEGG-enriched pathways were completed by ferroptosis, transcriptional dysregulation in cancer, the VEGF signaling pathway, and glycolysis/gluconeogenesis. The authors suggested that these findings may have implications for possible targets for new drugs against osteoarthritis [152].

On the RPM, gene expression levels of the mechanosensitive ion channel TRPC1 were reduced 5- to 10-fold in gene expression in adherent and suspension cultures of bovine chondrocytes. Furthermore, the authors showed that the *TRPV4* gene expression decreased with progressing dedifferentiation, indicating its vital role in chondrocyte phenotype maintenance [153].

Gene expression analyses on human articular chondrocytes exposed to the RPM for 24 h showed significant upregulations of *IL6*, *RUNX2*, *RUNX3*, *SPP1*, *SOX6*, *SOX9*, and *MMP13.* At the same time, *IL8*, *ACAN*, *PRG4*, *ITGB1*, *TGFB1*, *COL1A1*, *COL2A1*, *COL10A1*, *SOD3*, *SOX5*, *MMP1*, and *MMP2* remained unchanged, indicating that the chondrocytes experienced stress, as evidenced by the expression of markers for osteoarthritis and cartilage damage [154].

A summary of all findings is given in Table 6.

### 3.8. Effects of Microgravity on Muscle Cells

Transcriptomic alterations in muscle tissue were studied in space-flown mice. It was shown that 9 weeks of spaceflight on the ISS induced differential gene expression and differential alternative splicing patterns in the gastrocnemius and quadriceps muscles of 30-week-old female BALB/c mice [155]. The authors found 105 differentially expressed genes exclusive for the gastrocnemius and 55 genes exclusive for the quadriceps. In addition, 15 genes were regulated in both tissues. Differential alternative splicing occurred in 21 genes for both muscles; meanwhile, 159 genes were identified only in the quadriceps and 51 genes only in the gastrocnemius. Notably, transcripts belonging to skeletal muscle proteins were mainly differentially spliced and less differentially expressed, which hints at a key role of alternative splicing in the muscle transcriptomic response to µ*g*. Furthermore, alternative differential splicing was more strongly associated with actual physiological changes in the muscles than differential gene expression [155].

Another study on space-flown adult C57Bl/N6 male mice on board a BION-M1 capsule for 30 days explored the altered gene expression profiles in skeletal muscles [156]. A total of 680 differentially expressed genes were found in the soleus muscle, 72 in the extensor digitorum longus, including 24 of which that were identified in both tissues. Pathway analyses revealed that these genes belonged to relevant biological processes such as contractile machinery, calcium homeostasis, muscle development, cell metabolism, and inflammatory and oxidative stress responses. The authors proposed that these data might help to find new biomarkers and targets for developing and optimizing countermeasures and post-flight rehabilitation [156].

A third study specifically focused on NRF2, regulating the expression of heme oxygenase 1 and inhibiting the NLRP3 inflammasome. For this, male C57BL/6J wild-type and *Nrf2*-knockout (KO) mice were housed for 31 days on board the ISS in r-µ*g*. Afterwards, an RNA-seq analysis was conducted on soleus muscle samples [157]. Overall, it was found that the expression of NRF2 targets under µ*g* was significantly downregulated in KO mice compared to wild-type animals. Furthermore, exposure to µ*g* doubled the differentially expressed genes between KO and wild-type mice (60 under 1*g* vs. 120 under µ*g*). In addition, enrichment analyses revealed glucose metabolic processes and glycolysis/gluconeogenesis to be upregulated and processes such as brown fat differentiation and response to oxidative stress to be downregulated in KO mice under µ*g*. Considering additional histological analyses, which showed that the transition from oxidative to glycolytic muscle fibers was accelerated in KO mice, the authors concluded that NRF2 influences µ*g*-induced myofiber transition [157].

To test artificial gravity as a countermeasure for muscle atrophy in space, three groups of C57BL/6 J male mice were analyzed: one stayed on Earth as the ground control, the second was left in µ*g*, and the third was subjected to an in-flight 1*g* centrifuge during a 35-day stay on the ISS [158]. The authors found that the 1*g* environment prevented decreases in muscle mass and compositional changes in fiber types in the animals’ soleus muscle compared to the µ*g* group. In addition, centrifugation also prevented µ*g*-induced differential gene expression. Lastly, based on in silico transcriptome analyses, this study identified a novel candidate gene linked to muscle atrophy (*Cacng1*), which could be validated in further in vitro tests [158].

A summary of all findings is given in Table 7.

### 3.9. Effects of Microgravity on Bone

Cells from the mouse osteocytic line Ocy454 were exposed to 2, 4, and 6 days of r-µ*g* during the SpaceX Dragon-6 resupply mission and subsequently subjected to a global transcriptome analysis to elucidate the mechanisms of gravisensing and the responses to µ*g* in late osteoblasts and osteocytes. It could be shown that prolonged exposure to µ*g* leads to more substantial changes in gene expression. Enrichment analyses of the regulated genes showed a strong association with osteoporosis, bone resorption, detection of mechanical stimuli, bone development, regulation of osteoclast differentiation, and sensor reception of mechanical stimuli. After 6 days, the top differentially expressed pathways were involved in glucose metabolism, and the chief molecular and cellular functions were carbohydrate metabolism, cell death and survival, and cellular development. Interestingly, control experiments in s-µ*g* on an RCCS could not sufficiently replicate the findings from space. By comparing their data with other datasets from the NASA GeneLab database, the authors could also identify a set of 10 mechanosensitive transcripts regulated over different cell types and which might indicate a common response to µ*g* (Table 8). The study showed that exposure to µ*g* impaired osteocyte differentiation and increased glucose metabolism and oxygen consumption [159].

Another experiment also studying osteogenic differentiation in space analyzed epigenetic changes in human blood-derived stem cells grown in an osteogenic medium for 72 h. It was found that methylation at H3K4me3, H3K27me2/3, H3K79me2/3, and H3K9me2/3 residues were involved in cellular reprogramming and induction of gene expression [160].

In a 14-day co-culturing experiment with osteoclasts, osteoblasts, and endothelial cells on board a SpaceX Dragon spacecraft, it was shown that µ*g* reduced the expression of key osteoblast genes such as *ATF4*, *RUNX2*, and *Osterix* and that these effects could be reversed by supplementation with irisin, which might represent a countermeasure against astronaut bone loss [26].

Besides r-µ*g* in space, s-µ*g* devices were also used to study the effect of µ*g* on bone cells. Li et al. [161] exposed human bone marrow mesenchymal stem cells cultured in an osteogenic medium to altered gravity conditions on an RPM for 2, 7, and 14 days. They found that s-µg inhibited cell proliferation and differentiation towards osteoblasts but drove the cells toward adipogenesis instead. This was reflected by the expression profiles of genes related to the cell cycle (downregulation of *CDKN3*, *MCM5*, *CCNB1*, *CDK1*, and *CDC20*), osteogenic differentiation (downregulation of *RUNX2*, *ALPL*, *BMP2*, and *COL1A1*), and adipogenic differentiation (upregulation of *PPARG*, *CEBPA*, *CEBPB*, and *CFD*). Furthermore, the upregulation of tumorigenic genes was observed at the last time point [161].

To engineer bone tissue under s-µ*g*, Mann et al. [162] exposed human fetal osteoblast cells to an RPM for 7 and 14 days (Table 6). The authors reported differential gene expression of *TGFB1*, *BMP2*, *SOX9*, *ACTB*, *TUBB*, *VIM*, *LAMA1*, *COL1A1*, *SPP1*, and *FN1*. Notably, some of the initially adherently growing cells detached from the substrate and formed multicellular spheroids, which displayed bone morphological properties after 14 days [162].

Braveboy-Wagner and Lelkes [163] used an RPM to simulate different partial gravity conditions comparable to the Moon, Mars, and “full” s-µ*g* (Table 8). They studied the gene expression of the osteogenic markers *ALPL*, *RUN*, and *ON* in 7F2 osteoblasts cultured under different conditions and found that all three altered gravity levels induced a sharp decrease in the gene expression of all three markers. However, in contrast to cell proliferation rates and alkaline phosphatase activities, which were also analyzed in this study, the gene expression changes were not dose-dependent, but all RPM conditions induced identical reductions. This hints towards a certain gravity threshold, which induces a more binary on–off switching of these genes [163].

Cao et al. [164] focused on a relatively new class of non-coding RNAs, the circular RNAs (circRNAs), to determine whether they are involved in the answer of osteoblasts to conditions of s-µ*g*. For this purpose, they cultivated mouse pre-osteoblast MC3T3-E1 cells in an RCCS for 72 h and then performed an RNA-seq transcriptome analysis. As a result, differential expression with a fold change ≥ 2 was observed for 427 circRNAs (232 upregulated, 95 downregulated) and 1912 mRNAs (991 upregulated, 921 downregulated). Furthermore, KEGG analyses revealed an enrichment in the regulation of the actin cytoskeleton, focal adhesion, and RAS signaling pathway in the differentially expressed mRNAs. By taking into account the centrality of the detected circRNAs and mRNAs to osteoblast function, nine core regulatory factors were identified, comprising three circRNAs (*circ_014154*, *circ_010383*, and *circ_012460*) and six mRNAs (*Alpl*, *Bg1ap*, *Col1a1*, *Omd*, *Ogn*, and *Bmp-4*), which were additionally validated by qRT-PCR. Finally, *circ_014154* was the circRNA, which most likely plays a central role in osteogenic differentiation under µ*g* conditions [164].

A summary of all findings is given in Table 8.

### 3.10. The Impact of Microgravity on Cancer Cells

Cancer cells exhibit an elevated survival potential, uncontrolled proliferation, unlimited replicative potential, increased angiogenesis, high invasion potential, and metastasis [165]. Therefore, it is crucial to develop novel strategies to prevent, diagnose, treat, and cure the different types of cancer. Despite improved knowledge of the underlying mechanisms and pathways contributing to disease progression in various cancer types, curing this disease remains challenging.

Humans in space live in an extremely hostile environment with cosmic radiation, microgravity, a hypomagnetic field, and other stress factors [166]. The risk for cancer in space travelers is still unclear, but microgravity plays a role in the carcinogenesis of normal and cancer cells resulting in various alterations on the cellular level. Moreover, deleterious effects of radiation on cells seem to be accentuated under microgravity [166].

A recent study used deep, error-corrected, targeted DNA sequencing to detect somatic mutations in clonal hematopoiesis (CH)-driver genes in peripheral blood mononuclear cells isolated from the blood samples of 14 astronauts attending Space Shuttle missions between 1998 and 2001 [132]. The authors detected 34 nonsynonymous mutations of relatively low variant allele fraction in 17 CH-driver genes [132]. Predominantly mutations in TP53 and DNMT3A were found. Therefore, future retrospective and prospective examinations with a focus on clinical relevance and potential application in monitoring astronauts’ health are necessary. TP53 encodes the tumor suppressor p53 which is induced by various stress stimuli. P53 plays a role in apoptosis, DNA repair, growth arrest, or senescence. P53 dysregulation is a risk factor for developing cancers and is a candidate to be examined after deep exploration missions. Drago-Ferrante et al. [167] recently reviewed the available literature to answer this question, “Extraterrestrial Gynecology: Could spaceflight increase the risk of developing cancer in female astronauts?” The authors found a lack of knowledge on the effects of cosmic radiation and microgravity on gynecologic cancer. As of now, the number of female crewmembers who have attended a long-term mission is too small to answer this question [167].

A research paper published by Reynolds and coauthors [168] studied a group of 301 astronauts and 117 cosmonauts. The authors concluded from their results that if ionizing radiation is impacting the risk of death due to cancer and cardiovascular disease, the effect is not dramatic. They reported that they failed to find evidence sufficient to conclude that historical doses of space radiation pose an excess mortality risk for astronauts and cosmonauts. However, the planned deep space exploration will likely offer higher doses of space radiation than historical doses have, which will lead to a different risk profile for future space travelers [168].

The cancer incidence and mortality in the USA astronaut corps from 1959 to 2017 was published by Reynolds et al. [169]. The cohort consisted of 338 NASA astronauts. The average follow-up time was 28.4 years. In comparison to the general population, US astronauts show a decreased risk of cancer-specific mortality overall [169].

The following subchapters focus on the effects of r- and s-µ*g* on gene expression and genetics in different cancer cell types, such as thyroid, breast, prostate, and lung cancer, as well as the gastrointestinal tract and skin tumors. This concise review clearly shows that µ*g* research has become an important new technology to advance our knowledge in the field of cancer biology and investigates the impact of real and simulated weightlessness on the gene regulation and genetics in cancer. The most important publications are listed in Table 9.

#### 3.10.1. Gene Expression in Thyroid Cancer Cells under the Influence of Gravitational Changes

Unlike other common cancers, thyroid cancer usually occurs at younger ages and affects women two to three times more often than men, and thus becomes the second most common cancer in women under the age of 40 [170]. However, papillary thyroid carcinoma (PTC), which is common and occurs at a younger age, is easily treatable compared to follicular thyroid cancer and medullary thyroid cancer. The PTC is partially inherited and is associated with rearranged during transfection (RET) rearrangements and RET/PTC fusions [171]. In contrast to other thyroid cancer types, anaplastic thyroid cancer, which accounts for about 2% of thyroid carcinomas and occurs mainly in the elderly, has a poor 5-year survival rate of 7% [172]. In the astronaut population study of the US Astronaut Corps between 1959 and 2013 only one astronaut with thyroid cancer was found [169].

On the other hand, a reduction in thyroid activity (hypothyroidism) under space conditions was already observed in the mid-1980s during the “Cosmos-1667” and “Cosmos-1887” missions [173]. This coincidence led to a common interest of space research and cancer research in the molecular mechanisms behind these changes in thyroid function. Accordingly, this chapter summarizes the results of the last five years of molecular medicine research with a focus on gene expression changes in thyroid cancer under µ*g*.

In 2017, Bauer and co-workers presented a differentiated approach to analyzing proteomic data in three publications [174,175,176]. The basis of these three analyses were FTC-133 thyroid cancer-cell spheroids after a two- or five-day exposure on the RPM that were compared to adherent cells of the s-µ*g* experiment and static 1*g* controls. The FTC-133 cell line was initially obtained from a lymph node metastasis of follicular thyroid carcinoma of a 42-year-old male. A total of 5900 proteins were quantified by mass spectrometry [174]. Protein interaction analyses led the authors to hypothesize that during spheroid formation, increased ASAP1 production accompanied by decreased CAV-1 and p130cas levels causes localization of PXN, VCL, and PTK2 to the focal adhesion complex, thereby indirectly leading to the detachment of the cells [174]. The same year, Johann Bauer’s team developed a semantic knowledgebase using the FTC-133 protein dataset and publicly available data to capture the functional properties of proteins in a structured way [175]. This was expanded in 2018 to capture post-translational modifications (PTMs) and to analyze rare experimental results. PTMs, such as phosphorylation, glycosylation, ubiquitination, and acetylation for 69 candidate proteins from the original FTC-133 s-µ*g* experiment, were determined and documented [176].

The type II transmembrane protein aspartate β-hydroxylase (ASPH) is overexpressed in different cancer types and involved in proliferation, invasion, and metastasis [177]. In FTC-133 cells, it was found to be significantly upregulated in the hypergravity phase of the TEXUS-53 rocket flight mission [178]. Simulation of the hypergravity phase of the rocket mission in a centrifuge (18*g* for 1 min) resulted in moderate but significant regulations of *COL1A1*, *VCL*, *CFL1*, *PTK2*, *IL6*, *CXCL8*, and *MMP14* [179]. Nassef and co-workers compared live-cell imaging of the TEXUS-53 mission (FTC-133 thyroid cancer cells) with live-cell imaging of the TEXUS-54 mission (MCF-7 breast cancer cells) and found comparable cytoskeletal alterations such as filopodia and lamellipodia and suggested a common gravitational mechanism in human cancer cells [180].

Wise and co-workers [181] examined supernatants of FTC-133 thyroid cancer cells from the CellBox-1 module of the SpaceX CRS-3 cargo transport to the ISS. The cell culture supernatants comprise extracellular vesicles (EV) containing phospholipid bilayers for extracellular secretion. Transfer of tumor EVs has enhancing effects on the proliferation, migration, and treatment resistance of the recipient tumor cell. Measurements with imaging interferometric reflectance of the single particle from the spaceflight revealed an increase in the CD9/CD81 population compared with 1*g* ground controls. How far this impacts the tumor’s aggression has to be shown in further studies. After remaining on the ISS, FTC-133 thyroid cancer cells developed MCS in all six units [182]. Melnik and co-workers quantified the RNA level in 19 candidate genes via qPCR and the proteins released into the supernatants (n = 25). Comparison of the 5-day µ*g*-grown spheroids with the ground control revealed a significant expression depletion in *VCL*, *PXN*, *ITGB1*, *RELA*, *ERK1*, *ERK2*, *MIK67*, and *SRC*, whereas *ICAM1*, *COL1A1*, and *IL6* were upregulated exclusively in the µ*g*-adherent cells. The protein secretion measured by multianalyte profiling technology was not significantly altered, with the exception of elevated angiopoietin 2 (Ang-2). The supernatants of 10-day space-flown samples showed an increase in the Ang-2 level compared with the corresponding 10-day ground control [182].
cells-12-01043-t009_Table 9Table 9Changes in gene regulation of different cancer cells exposed to microgravity conditions.Cancer TypeCell LineGene RegulationMicrogravityReference*Thyroid**Cancer*FTC-133Downregulation of *VCL*, *PXN*, *ITGB1*, *RELA*, *ERK1*, *ERK2* in RPM-AD and MCSSuppression of *MIK67 and SRC* in MCSUpregulation of *ICAM1*, *COL1A1*, and *IL6* in RPM-AD cellsSpaceflight to the ISS,5 d[182]
FTC-133DEX inhibited dose-dependent spheroid formation;*NFKB2*, *VEGFA*, *CTGF*, *CAV1*, *BCL2* (*L1*), or *SNAI1* were clearly affected by DEX.RPM,4 h, 3 d[183]
FTC-133Exosomal microRNA composition;Array scan of a total of 754 miRNA targets: more than 100 differentially expressed miRNAsSpaceflight to the ISS,12 d[184]
FTC-133WROML-1Nthy-ori 3-1*DEGs MYC*, *NR3C1*, *FKBP5*, *DUSP1*, *MUC1*, *MAPK14*Balance between adhesion, anti-adhesion, and cell–cell connections enable detachment of adherent thyroid cells on the RPMRPM3 d[185]*Breast cancer*CRL2351Upregulation of *VIM*, *RHOA*, *BRCA1*, and *MAPK1* in AD and MCSUpregulation of *ERBB2* in MCS.No significant change in VEGF and *RAB27A.*RPM, 5 d[186]
MCF-7Significant upregulations of the mRNAs of enzymes degrading heme, *ANXA1*, *ANXA2*, *CTGF*, *CAV2*, *ICAM1*, *FAS*, *Casp8*, *BAX*, *p53*, *CYC1* and *PARP1* in MCS cells vs. 1*g* and AD cells.RPM,24 h[187]
MCF-7, MDA-MB-231MCF-7: Downregulation of *FAK1*, *PXN*, *TLN1*, *VCL*, and *CDH1* in AD cells and *PXN*, *TLN*, and *CDH1* in MCS. Vinculin and β-catenin are key mediators of MCS formation.MDA-MB-231: No change in *ACTB*, *TUBB*, *FN1*, *FAK1*, and *PXN* gene expression.Downregulation of *LAMA3*, *ITGB1* mRNAs in AD cells, and *ITGB1*, *TLN1*, and *VCL* mRNAs in MCS.RPM,24 h[188]
MCF-7Early upregulation of *KRT8*, *RDX*, *TIMP1*, *CXCL8* mRNAsDownregulation of *VCL* after the first parabola.TEXUS-54,6 min of r-µ*g*, PFC,31 parabolas[180]
MDA-MB-231Upregulation of *ICAM1*, *CD44* and *ERK1* mRNAs after the first parabola (P1)Delayed upregulation of *NFKB1*, *NFKBIA*, *NFKBIB*, and *FAK1* after the last parabola (P31).*PRKCA*, *RAF1*, and *BAX* mRNAs were not changed.Cleaved caspase-3 was not detectable in MDA-MB-231 cells exposed to PF maneuvers.PFC,31 parabolas[189]
MDA-MB-231DEGs in MCS: *AKT*, *KI67*, *BCL2*, *BAX*, *CD44*, and *MMP9* after 72h; *AKT*, *KI67*, *BCL2*, and *MMP9* after 24 hDEGs in RPM-AD: *BAX*, *CD44*, and *MMP9* after 72 h; *BCL2* and *MMP9* after 24 hRPM24 h, 72 h[190]
MCF-7, MDA-MB-231qPCR-verified genes search in the mammalian phenotype database and the human genome-wide association studies (GWAS) Catalogue*ERK1*, *AKT1*, *MAPK14*, *EGFR*, *CTNNA1*, *CTNNB1*, *ITGB1*, *COL4A5*, *ACTB*, and *TUBB* gene expression of MCSs differentially regulated in both cell types.7 genes (ACTB, CD44, EGFR, ITGB1, PXN, TUBB, and VCL) were successfully analyzed using WCGNARPM14 d[191]*Prostate**cancer*PC-33 d: Downregulation of *VEGF*, *SRC1*, *AKT*, *MTOR*, and *COL1A1* in MCS.No change in *FLT1*, *RAF1*, *MEK1*, *ERK1*, *FAK1*, *RICTOR*, *ACTB*, *TUBB*, and *TLN1* mRNAs.Upregulation of *ERK2* and *TLN1* in AD and *FLK1*, *LAMA3*, *COL4A5*, *FN1*, *VCL*, *CDH1*, and *NGAL* mRNAs in AD and MCS5 d: Downregulations of *VEGF* in AD and MCS, *FN1*, *CDH1*, and *LAMA3* in AD and *SCR1* in MCS.Upregulations of *FLT1*, *AKT*, *ERK1*, *ERK2*, *LCN2*, *COL1A1*, *TUBB*, and *VCL* mRNAs in AD and MCS.Upregulations of *FLK1*, *FN1*, and *COL4A5* in MCS as well as *LAMB2*, *CDH1*, *RAF1*, *MEK1*, *SRC1*, and *MTOR* mRNAs in AD.RPM,3 d, 5 d[192]
PC-324 h: Upregulation of *ACTB*, *MSN*, *COL1A1*, *LAMA3*, *FN1*, *TIMP1*, *FLT1*, *EGFR1*, *IL1A*, *IL6*, *CXCL8*, and *HIF1A* in MCS.Elevation of *LAMA3*, *COL1A1*, *FN1*, *MMP9*, *VEGFA*, *IL6*, and *CXCL8* mRNAs in AD.Downregulation of *TUBB*, *KRT8*, *IL1B*, *IL7*, *PIK3CB*, *AKT1 and MTOR* in AD.RPM,30 min, 2 d, 4 d, and 24 h[193]
PC-3Differentially expressed regulatory lncRNAs and micro RNAs, portfolio of 298 potential biomarkers.NGS: 5 upregulated cytokines (*CCL2*, *CXCL1*, *IL6*, *CXCL2*, *CCL20*), one zinc-finger protein (*TNFAIP3*), and one glycoprotein (*ICAM1*).Regulated *miR-221* and the co-localized lncRNA *MIR222HG* induced by PF maneuvers.PFC,31 parabolas[194]*Lung cancer*A549, H170324 h: A549 and H1703 highly expressed the migration-related genes *MMP2*, *MMP9*, *TIMP1*, and TIMP2 compared to CONT.3D clinostat,24 h, 38 h[195]
CRL-5889*TP53*, *SOX2*, *CDKN2A*, *PTEN*, and *RB1* gene expressions were significantly upregulated in AD.No change in *AKT3*, *PIK3CA*, and *NFE2L2.*RPM,72 h[196]
A549S-µ*g* induced increased expression of *FCGBP*, *BPIFB*, *F5*, *CST1*, and *CFB*Potential biomarkers for lung cancer2D clinostat 24 h, 48 h, 72 h[197]*Colorectal cancer*DLD1 cellsUpregulation of *PTEN* and *FOXO3*Downregulation of *AKT*Increase in apoptosisRCCS-HARV[198]*Gastric**Cancer*EPG85-257 RDB EPG85-257 PReduced expression of genes related to drug resistance and increased DNA/RNA damage marker expressionRCCS,72 h[199]*Liver cancer*HepG2 cellshBTSCsSignificant upregulation of*OCT4*, *SOX17*, *and ALB* in HepG2 cellsRCCS,15 d[200]*Pancreas**Cancer*PaCa-44 Metabolic reprogramming orchestrated by the activation of HIF-1α and PI3K/Akt pathwaysRPM,1 d, 7 d, 9 d[201]*Skin cancer (Melanoma)*BLMDownregulation of the endothelial NOS-sGCMRP4/MRP5 pathway.Suppression of sGC expression and activity correlates inversely to tumor aggressiveness.Downregulation of cancer-related genes *iNOS* and *GC-A/GC-B.*Fast rotating2D clinostat[202]Abbreviations: Adherent (AD); days (d); dexamethasone (DEX); differentially expressed genes (DEGs); high aspect ratio vessels (HARV); hours (h); International Space Station (ISS); parabolic flight campaign (PFC); random positioning machine (RPM); real microgravity (r-µ*g*); rotating cell culture system (RCCS); simulated microgravity (s-µ*g*); Technologische EXperimente Unter Schwerelosigkeit (TEXUS); three-dimensional (3D); two-dimensional (2D).


Dexamethasone (DEX) is an inhibitor of cell proliferation and affects, among others, medullary thyroid cancer cells [203]. It causes a cell cycle arrest at the G1 phase and increased apoptosis. Furthermore, it induces an increase in the *p27* expression and an expression depletion of cyclin-dependent kinases. Using synthetic glucocorticoid DEX, Melnik et al. were able to suppress the spheroid formation in FTC-133 cells, which otherwise usually takes place under the influence of s-µ*g* [183]. Furthermore, a DEX-dependent change in the expression of *NFKB2*, *VEGFA*, *CTGF*, *CAV1*, *BCL2*, and *SNAI1* was detected. DEX thus influences proliferation and migration (*VEGFA*, *CTGF*), cell cycle (*CTGF*), and apoptosis (*BCL2*).

A new study by Melnik et al. [185] demonstrated that this inhibition was selective for two metastatic thyroid cancer-cell lines, FTC-133 and WRO, whereas benign Nthy-ori 3-1 thyrocytes and recurrent ML-1 follicular thyroid cancer cells were not affected by DEX (0, 10, 100, 1000 nM) when exposed for 72 h to the RPM (Table 9). DEX disrupts random positioning-triggered p38 stress signaling in FTC-133 cells [185]. DEX treatment of FTC-133 cells is associated with enhanced adhesiveness. This process is caused by the restored, pronounced formation of a normal number of tight junctions. In addition, RPM-exposed Nthy-ori 3-1 and ML-1 cells exhibit an elevation in the anti-adhesion protein mucin-1, which might be a protection mechanism against mechanical stress [185]. The balance between adhesion, anti-adhesion, and cell–cell connections enables the detachment of adherent human cells exposed to the RPM or not, allowing selective inhibition of thyroid in vitro metastasis by DEX [185].

#### 3.10.2. Breast Cancer Cells in Microgravity with a Focus on Gene Expression

Female astronauts face in-space galactic cosmic radiation (GCR) and have a greater risk for breast cancer development [204]. Simulated GCR-induced expression of *Spp1* coincide with mammary ductal cell proliferation and preneoplastic changes in an Apc^Min/+^ mouse model [204].

Until very recently, there have been insufficient numbers of women in space exposed to long-duration, low-dose rate, and proton and heavy ion radiation to determine their cancer risk [167].

For various cell types of somatic and tumorigenic origin, profound gene expression and genetic changes have been observed [205,206]. The “source“ of µ*g* may be real, such as in free-fall experiments, parabolic flights, and space, or it may be simulated, such as in RWV, RCCS, or RPMs. In breast cancer, genes particularly affected (and most commonly tested) are those related to proliferation, apoptosis, cancer stemness, and metastatic potential (Table 9).

In a recently conducted 72-h RPM experiment with MDA-MB-231 cells, the gene expression of *AKT*, *BAX*, *BCL2*, *CD44*, and *MMP9* was upregulated compared to the 1*g* control [190]. This held particularly true for the cells forming spheroids more than for cells remaining surface-adherent [190]. In the same cell line (MDA-MB-231), various changes in gene expression occurred as early as after 24 h [188]. A downregulation of *LAMA3*, *ITGB1* mRNAs in AD cells and *ITGB1*, *TLN1*, and *VCL* mRNAs in MDA-MB-231 MCS was measured after a 24-h RPM exposure. In contrast, the *ACTB*, *TUBB*, *FN1*, *FAK1*, and *PXN* mRNAs were not significantly altered in MDA-MB-231 cells [188]. In addition, MCF-7 cells exposed to the RPM showed a decrease in *FAK1*, *PXN*, *TLN1*, *VCL*, and *CDH1* in AD cells and *PXN*, *TLN*, and *CDH1* in MCS [188].

MDA-MB-231 cells were also exposed to r-µ*g* during a parabolic flight experiment (31 times μ*g* for 22 s) [189]. *ICAM1*, *CD44*, and *ERK1* mRNA were all upregulated as early as after the first parabola, whereas *NFKB1*, *NFKBIA*, *NFKBIB*, and *FAK1* expression was upregulated towards the end of the campaign. Remarkably, *CD44* and *NFKBIA* expression was also upregulated by hyper-*g* phases. In a further parabolic flight campaign, the same cell line showed upregulations of *KRT8*, *RDX*, *TIMP1*, and *CXCL8* mRNAs and a downregulation of *VCL* expression [180].

In another long-term experiment (2 weeks) on an RPM with MCF-7 cells, the expression of *BCAR1* and *MAPK8* remained unchanged, while a tendency of reduced expression of *CDH1* was observed [207]. Using MCF-7 cells, Wise et al. analyzed cell culture supernatants of cells incubated on an RPM for 5 and 10 days on the change in small EV release compared to the corresponding 1*g* controls. Similar to their previous study, the number of secreted EVs increased in both RPM groups. Comparing the EV release after 5 vs. 10 days of s-μ*g*, the authors did not find a mentionable difference, suggesting that the regulatory changes of the exosome release are happening early on after the onset of the s-μ*g* condition [208].

Investigating the adenocarcinoma cell line (CRL2351), Strube et al. observed an upregulation of *BRCA1*, *RHOA*, *VIM*, *HER2*, and *MAPK*1 in all cells cultured under s-μ*g* (adherent and spheroids), whereas *ERBB2* was upregulated in the spheroids, but not in the adherent cells after a five-day RPM exposure. *RAB27A* was downregulated, while the expression of *VEGF* remained unchanged [186]. In further experiments with the same cell line lasting only 24 h, the *VCAM1* expression was significantly upregulated, while *VIM* was significantly downregulated. Expression of *MAPK1*, *MMP13*, *PTEN*, and *TP53* remained unchanged [209]. As previously described as a phenomenon in µ*g* [210], gene expression levels do not always correlate to detection levels of the respective gene products indicating a counterregulatory effect.

Another study showed that early apoptosis is actively counteracted by activation of the two different survival pathways ERK and Akt occurring in MCF10A and MCF-7 cells (Table 9). However, after 72 h a significant increase in apoptosis was detected only in non-adherent MCF-7 cells, proposing that the loss of adherence and cytoskeletal alterations induced by µ*g* can ultimately overcome the survival strategies of MCF-7 cancer cells [211].

Calvaruso et al. [190] investigated MDA-MB-231 cells under s-μg conditions for 24 and 72 h. MCSs display an increase in both *CD44* (cancer stemness) and *MMP9* (metastasis) expression. qRT-PCR of the following genes and main regulated processes: *AKT* and *KI67* (cancer proliferation) as well as *BAX* and *BCL2* (apoptosis) was performed. AKT and Ki67 were significantly upregulated in MCS after 72 h [190]. *BAX* mRNA was elevated in the RPM-AD populations at 72 h, suggesting that AD cells will undergo apoptosis. In contrast, *BCL2* was low in RPM-AD cells after 72 h, but highly upregulated in MCS after 72 h indicating the different behavior of the two different phenotypes on the RPM (Table 9) [190].

A recent long-term s-µg study demonstrated an association between the real metastatic microtumor environment and breast cancer cells (MCF-7, MDA-MB-231) exposed to the RPM with respect to biological factors related to the extracellular matrix, cytoskeleton, morphology and different cellular signaling pathways [191]. It was shown that the key elements (*ERK1*, *AKT1*, *MAPK14*, *EGFR*, *CTNNA1*, *CTNNB1*, *ITGB1*, *COL4A5*, *ACTB* and *TUBB*) were differentially expressed. The ‘*Weighted Gene Co-Expression Network Analysis‘* (WGCNA) was applied to examine if these DEGs already known to be involved in the 3D formation of breast cancer cells exposed to µg determine MCS formation in a general way. The WGCNA revealed 7 (*ACTB*, *CD44*, *EGFR*, *ITGB1*, *PXN*, *TUBB,* and *VCL*) out of 18 genes. The following enrichment analysis determined that this group of genes comprised the cluster of genes responding to the morphology of breast cancer (Table 9) [191].

#### 3.10.3. Gene Expression in Prostate Cancer Cells under the Influence of Gravitational Changes

Among all cancers, prostate cancer claims the second most lives in the male population of developed countries after lung cancer. Affected men are, on average, 65 years old and have good chances of recovery before metastases occur. However, the 5-year survival rate is reduced from almost 100% to only 30% after the formation of metastases [212]. Therefore, the search for effective and early-responding biomarkers is a highly urgent concern in molecular medicine.

Prostate cancer was the most frequently diagnosed cancer among US astronauts [169]. The incidence was elevated compared to the US general population [169]. This elevation may be due to detection bias generated by early and frequent screening in the astronauts.

The tragic end of the Columbia STS-107 mission on 1 February 2003, also destroyed much of the results of the first, and so far the only, cultivation of prostate cancer cells under real space conditions [213].

In the last three years, three publications described the response of the prostate carcinoma cell line PC-3 to gravitational changes (Table 9) [192,193,194]. PC-3, a prostatic small cell carcinoma model, is characterized by a high metastatic potential and lack of PSA expression [214].

By randomizing the gravity vector in the RPM and thereby generating s-µ*g*, Hybel and co-workers examined the gene expression of 23 candidate genes by qPCR [192]. After three days of s-µ*g* exposure, but especially after five days of s-µ*g* exposure, significant changes in gene expression of VEGF-signaling, extracellular matrix (ECM), and focal adhesion could be detected. This expression profile differs between adherent cells and multicellular spheroids (MCS) (Figure 4). After five days of RPM exposure, moderate but significant expression increases in two investigated genes of the cytoskeleton *ACTB* and *TUBB* could be observed. Since the suppression of the VEGF expression is already established in cancer therapy, the study focused on VEGF-signaling genes. In particular, the increase in FLK1 and LCN2 expression after three days s-µ*g* in adherent cells and MCS is striking. In adherent cells, the *FLK1* expression increase is transient and drops again after five days s-µ*g*. In MCS, the *FLK1* and *LCN2* expression increase coincides with the decreased VEGFA expression. Similar to *FLK1*, for the ECM gene *COL4A5*, the increase in expression in adherent cells levels off after five days of RPM.

Dietrichs and co-workers [193] used a similar setup as in the previous study by Hybel et al. [176]. However, on the one hand, the running periods of the RPM were significantly reduced (30 min to 24 h); on the other hand, the number of genes examined with qPCR was increased to 30, and the protein secretion was measured. Thus, the study focused on observing the early stages of the development of PC-3 spheroids. In the study, spheroid development is associated with a significant increase in the expression of the genes *ACTB*, *MSN*, *COL1A1*, *LAMA3*, *FN1*, *TIMP1*, *FLT1*, *EGFR1*, *IL1A*, *IL6*, *CXCL8*, and *HIF1A* after 24 h of RPM exposure. The gene expression and the protein secretion of interleukins are predominantly regulated during two to four hours of RPM exposure. The authors conclude that the cytokines IL-1α, IL-1β, IL-6, and IL-8 have a significant role in prostate cancer development [193].

Parabolic flight experiments are characterized by embedding a 22-s µ*g* phase in two 20-s hypergravity phases (Table 1). In the study by Schulz and co-workers [194], PC-3 cells were subjected to these brief gravitational changes. The transcriptome-wide analysis of the data obtained by RNA sequencing revealed that, especially after the first parabola, cytokines, and here, in particular, chemokines, were differentially expressed. Besides chemokines of NF-*κ*B signaling, the inflammatory cytokines TNF-α and LIF were highlighted to play a role in interaction with CXCL-8 in cancer development and the chemokines CXCL3 and CXCR2 in their role in cell migration. However, among the 298 differentially expressed genes, lncRNAs (23.9%) and miRNAs (2.5%) were also found. The genomically closely located miRNA miR221 and lncRNA *MIR222HG* are upregulated after the first parabola and play a role in cancer development and progression, respectively. Additionally, the lncRNA *MIR3142HG* is already upregulated after the first parabola. The role of *MIR3142HG* in cancer progression is currently controversial.

#### 3.10.4. Gene Expression Changes of Lung Cancer Cells Exposed to Microgravity Conditions

Lung cancer remains the leading cause of cancer death, with an estimated 1.8 million deaths (18%), but breast cancer has surpassed lung cancer as the most commonly diagnosed cancer, with an estimated 2.3 million new cases (11.7%) [216]. US astronauts have a substantially lower incidence and mortality from lung cancer in comparison to the US general population, and the few cases that have been observed have been less lethal than expected [169].

Multiple effects of both r- and s-μ*g* have also been observed in various lung cancer cell lines. For example, squamous non-small cell lung cancer cells (CRL-5889) exposed to s-μ*g* using an RPM for 72 h showed increased spheroidal apoptosis and upregulation of *TP53*, *CDKN2A*, *PTEN*, *SOX*2, and *RB1*, while the expression of *AKT3*, *PIK3CA*, and *NFE2L2* remained unchanged (Figure 5) [196]. After 24 h on a 3D RPM, the migration-related genes *MMP-2*, *MMP-9*, *TIMP-1*, and *TIMP-2* were all upregulated both in a squamous (H1703) and in an adenocarcinoma (A549) cell line [195].

In addition, the transcription of coding RNA and miRNAs regulating genes associated with cell cycle regulation, apoptosis, and stress response were altered in their expression by s-μ*g* in A549 cells [217].

In another study also performed on A549 and H1703 cells, the expressions of *MMP-2*, *MMP-9*, *TIMP-1*, and *TIMP-2* were downregulated in both cell lines after 36 h of exposure to a 3D clinostat [218]. The effect was also observable but less pronounced after 12 h.

Lung cancer stem cells (derived from the H460 cell line) downregulate their expression of stemness genes *NANOG* and *OCT-4* after 6 h of RPM exposure [219]. The effect was still present but less pronounced after 24 h.

A recent study from December 2022 demonstrated 13 DEGs associated with the prognosis of lung cancer in alveolar basal-epithelial (A549) cells exposed to a 2D clinostat for 24, 48, and 72 h (Table 9) [197]. A gene set enrichment analysis revealed that these DEGs are enriched in humoral immunity pathways. In parallel, a morphology change, a reduced proliferation rate together with an increased epithelial E-cadherin expression and decreased mesenchymal N-cadherin expression were found. An increased expression of *FCGBP*, *BPIFB*, *F5*, *CST1*, and *CFB* and their correlation to epithelial-to-mesenchymal transition (EMT) under clinorotation were detectable as potential tumor suppressor biomarkers [197].

Taken together, these results provide new options to establish new therapeutic strategies for lung cancer patients.

#### 3.10.5. Cancer Cells of the Gastrointestinal Tract Exposed to Microgravity

This subchapter comprises studies with colorectal cancer cells, gastric cancer cells, and pancreas cancer cells cultured under both r-µ*g* and s-µ*g* conditions over the last five years.

Colorectal cancer (CRC) is the third most common malignancy and the second most deadly cancer. This cancer type was responsible for 1.9 million incidence cases and 0.9 million deaths worldwide in 2020 [220]. CRC cells have been one of the first cell types used in µ*g* research [215], mainly for engineering 3D tissues or spheroids. However, in the last five years, no CRC cells have been studied in r-µ*g* research on the ISS, and only one paper investigating the impact of s-µ*g* on CRC cells with a focus on gene expression changes was found following an extensive search of the literature.

An interesting study focused on colon cancer incidence in astronauts, among others. In US astronauts (1959–2017), Reynolds et al. [169] investigated several relevant topics including cancer-specific mortality rates, cancer incidence rates, and cancer case-fatality ratios. In comparison with the US general population, colon cancer revealed a sizeable (but insignificant) decrease in incidence and mortality. A healthy lifestyle and differential screening can explain these trends towards reducing colon cancer among astronauts [169].

A RCCS-HARV was used to investigate the impact of s-µ*g* on the viability and morphology of different CRC cell types [198]. Increased programmed cell death was detected in DLD1, HCT116, and SW620 cells. In addition, gene expression analysis of DLD1 cells revealed an upregulation of tumor suppressors *PTEN* and *FOXO3*, leading to *AKT* downregulation and further induction of apoptosis through the upregulation of CDK inhibitors *CDKN2B* and *CDKN2D* [198]. The detection of enhanced apoptosis in cancer cells grown under µ*g* conditions was also reported earlier for other cancer cell types such as thyroid cancer, lung cancer, or breast cancer cells [196,221,222].

Worldwide, an estimated 1,089,103 patients were diagnosed with gastric cancer in 2020. A total of 768,793 (7.7%) died from gastric cancer in 2020. Stomach or gastric cancer is the fifth most commonly diagnosed cancer worldwide [216].

One recent publication (Table 9) focused on changes in drug resistance of gastric cancer cells (GCC) cultured under s-µ*g* [199]. The human resistant and sensitive GCC (EPG85–257 RDB and EPG85–257 P) were exposed to an RCCS [199]. The authors showed that RCCS exposure, together with doxorubicin treatment, was cytotoxic to the GCC [199]. RCCS-exposed cells exhibited a reduced expression of genes related to drug resistance and increased DNA/RNA damage marker expression. Simulated µ*g* conditions alter the expression of MDR genes in GGG, improve cell survival, and induce cytoskeletal changes to increase the susceptibility of GCC to chemotherapy. An interesting finding was that µ*g* weakened the action of genes associated with drug resistance [199].

In the field of liver cancer, one paper using HepG2 cells in µ*g* was published in 2019. The effects of s-µ*g* using an RCCS on HepG2 and human biliary tree stem/progenitor cells (hBTSCs) were studied [200]. HepG2 cells are derived from a well-differentiated hepatocellular carcinoma. S-µ*g* promoted 3D cultures in both cell types. A significant increase in stemness gene expression was observed in hBTSCs exposed to the RCCS. At the same time, the expression of hepatocyte lineage markers in hBTSCs was impaired by s-µ*g* [200].

HepG2 was cultured in a hormonally defined medium under 1*g* and RCCS conditions. At 1*g*, HepG2 cells exhibited a low or no expression of *OCT4* and *SOX17*, but significant upregulation of both genes was measured under s-µ*g*. In addition, s-µ*g* increased *ALB* gene expression in HepG2 cells. Furthermore, in HepG2 cells, s-µ*g* induced a significantly lower transcription of CYP3A4, a marker of late-stage (i.e., Zone 3) hepatocytes.

RCCS exposure induced the formation of 3D cultures, stimulated pluripotency, and glycolytic metabolism in HepG2 and biliary tree stem/progenitor cells [200].

One paper studying the influence of long-term (1, 7, and 9 days) RPM exposure on pancreas cancer cells (PCC; PaCa-44 cell line) was published in 2022 [201]. Pancreatic ductal adenocarcinoma is one of the most severe tumors worldwide and represents the fourth–fifth cause of death. Therefore, novel strategies are necessary to increase our knowledge of this cancer type. The study comprised proteomic, lipidomic, and transcriptomic analyses. Under s-µ*g*, the cells aggregate in 3D spheroidal structures. Cellular morphology is altered by the modulation of proteins involved in Cdc42 and RhoA signaling. This results in cytoskeletal changes, angiogenesis, and stemness. Furthermore, it is reported that a metabolic reprogramming orchestrated by the activation of HIF-1α and PI3K/Akt pathways, more active glycolysis compared to the adhesion condition, is involved in regulating proliferation, metastasis, and aggressiveness. These data indicate that the PCC exposed to the RPM differentiates toward a more aggressive metastatic stem-cell-like phenotype [201].

#### 3.10.6. Gene Expression Changes of Skin Cancer Cells Exposed to Microgravity

Until today, µ*g* research in the field of skin cancer has been performed on malignant melanoma cells. Skin cancer comprises non-melanoma cancer (basal cell carcinoma, squamous cell carcinoma, Bowen’s disease, and actinic keratoses) and malignant melanoma. Although the latter accounts for only 1% of all skin cancers, it is the most lethal form and is responsible for approximately 80% of all skin cancer deaths [223]. According to GLOBOCAN, 324,635 new cases occurred globally in 2020 (1.7%) [216]. Furthermore, melanoma metastasis is still a significant therapeutic challenge because of its poor response to chemotherapy.

Astronauts face cosmic radiation during their spaceflights, and thus it is vital to determine skin cancer incidence and mortality [169]. For melanomas, significant increases in incidence and mortality have been detected. The elevated incidence of melanoma is comparable to that observed in aircraft pilots. The authors concluded that this elevation might be associated with ultraviolet radiation or changes in lifestyle factors rather than any spaceflight exposure [169].

Ivanova et al. [202] studied the impact of clinorotation using a fast-rotating 2D clinostat on the gene expression of NOS isoforms, sGC, GC-A/GC-B, and multidrug resistance-associated proteins 4/5 (MRP4/MRP5) as selective cGMP exporters in human metastatic melanoma cells (highly metastatic and non-pigmented BLM melanoma cells) (Table 9). The endothelial NOS-sGCMRP4/MRP5 pathway was downregulated in s-µg compared to 1*g*. The suppression of sGC expression and activity correlates inversely to tumor aggressiveness. Furthermore, clinorotation reduced the expression of the cancer-related genes iNOS and GC-A/GC-B. The results suggest that future studies in r-µ*g* can benefit from considering GC-cGMP signaling as a possible factor for melanocyte transformation [202].

As mentioned earlier, s-µ*g* can promote programmed cell death in different cancer cell types [196,221,222]. RPM exposure of BL6-10 cells (highly lung-metastatic B16 melanoma cell line) inhibited cell proliferation/metastasis via the FAK/RhoA-regulated mTORC1 pathway. S-µ*g* increased apoptosis, altered the cytoskeleton, reduced focal adhesions (FAs), and suppressed FAK/RhoA signaling. RPM exposure reduced the expression of the mTORC1-related raptor, pS6K, pEIF4E, pNF-κB, and pNF-κB-regulated Bcl2 [224]. Furthermore, s-µ*g* inhibited the expression of nuclear envelope proteins (NEPs) lamin-A, emerin, sun1, and nesprin-3, which control nuclear positioning, and suppresses nuclear positioning-regulated pERK1/2 signaling. Therefore, the data demonstrate that s-µ*g* induced apoptosis via suppressing the FAK/RhoA-regulated mTORC1/NF-κB and ERK1/2 pathways. This may propose the FAK/RhoA network as a novel target for designing new therapeutics for humans in space on a long-term mission [224]. A second paper from this group demonstrated that s-µ*g* obstructs focal adhesions, leading to inhibition of FAK and RhoA signaling, and blockage of the mTORC1 pathway, which eventually results in activation of the AMPK pathway and reduced melanoma cell proliferation and metastasis [225].

Taken together, our knowledge of skin cancer in µ*g* is based on studies focusing on malignant melanoma cells. However, over the last five years, only a few studies have been published focusing on melanoma cells cultured under microgravity conditions with a focus on altered gene expression. It was demonstrated that melanoma cells exposed to s-µ*g* revealed an increase in apoptosis, alterations of the cytoskeletons, reduced focal adhesions, and the FAK/RhoA signaling pathway.

### 3.11. Recent Findings in Plants under Microgravity Conditions

Plants play a dual role in space biology as a part of bioregenerative life-support systems (BLSS) that enable prolonged habitation in space [226] and at the same time as model organisms to better understand basic plant physiology and improve crop yield on Earth [227]. Plants are currently investigated on different µ*g* platforms. Figure 6 shows a sounding rocket and ISS experiment with *Arabidopsis thaliana*. The lack of ethical concerns and the apparent ease of using plants to conduct observations of complex regulatory and signaling networks in whole organisms make them appealing for research under different gravity conditions. However, plants are also highly susceptible to spaceflight’s many abiotic stresses, which requires careful experimental planning [228]. We discussed gravity-related molecular changes in plants in detail in our last review [229] and therefore want to dedicate the following short chapters to new insights gained under r-µ*g*, the concomitant challenges of designing experiments in space, and promising methods.

#### 3.11.1. Growth and Development

Understanding the effect of µ*g* and other space-related stressors on plant growth and development is a prerequisite for successfully establishing long-term BLSS for off-world colonization or extended space flights, especially regarding the timing of developmental stages and plant fitness across multiple generations.

Experiments on the Chinese space lab Tiangong-2 have shown that the flowering time of *Arabidopsis thaliana* (ecotype Columbia), germinated and grown in µ*g* conditions, was delayed by 20 days compared to control plants on the ground. In Arabidopsis, flowering time is controlled by a circadian clock-dependent mechanism that integrates photoperiod and temperature information to induce the expression of FLOWERING LOCUS T (FT) in leaves, upon which it is transferred to the shoot apical meristem to promote flower formation together with SUPPRESSOR OF OVEREXPRESSION OF CO1 (SOC1), which is locally produced in response to FT accumulation. Using a camera-based observation of transgenic plants expressing GFP under the *FT*-promotor, a significant shift in the peak expression of FT was observed. The subsequent transcriptomic analysis identified a group of differentially expressed genes related to the flowering time that are interactors of FT and SOC1. Therefore, FT and SOC1 could integrate spaceflight-related stimuli into the pathways that regulate flowering [230]. However, in another experiment by the same group, plants uploaded to the space lab after they had already reached the rosette stage showed no such delay [231]. In contrast, growth experiments in the Japanese Kibo module on the ISS describe a mostly normal development of Arabidopsis under microgravity conditions [232]. Developmental studies in the past also support the notion of unhindered completion of the life cycle of Arabidopsis during spaceflight [233,234,235].

A study with the Arabidopsis ecotypes Columbia and Wassilewskija was conducted in the Vegetable Production System aboard the ISS to analyze root skewing behavior and transcriptomic differences between the two ecotypes and two mutants [236]. SPIRAL1 (SPR1) is a regulator of cortical microtubule dynamics and is involved in skewing behavior. The number of differentially regulated transcripts during spaceflight was significantly lower in the *spr1* mutant than the Columbia wild type, indicating a more efficient adaptation response in the *spr1* mutant. A *sku5* mutant line, lacking the skewing-related lipid raft-associated protein SKU5, reacted to spaceflight with a 2–5x increase in the number of differentially regulated transcripts compared to its wild-type Wassilewskija background. Gene ontology enrichment identifies various environmental stress responses, including ABA and ROS signaling, among the most enriched categories [236], which were also found in other ecotypes [237]. While the *spr1* mutant adapts more quickly to the spaceflight environment, the *sku5* mutant expends more resources during the adaptation. SKU5 is involved in plasma membrane reorganization in response to stresses, which requires the activation of alternate pathways upon its loss.

The above findings do not show a clear consensus on the effect of spaceflight on plant development, and the limited information on cross-generational effects warrants further investigation of long-term plant viability during spaceflight. However, the importance of stress adaptation to spaceflight is made clear, and the next steps should involve identifying and characterizing specific stress-response regulators that react to spaceflight.

#### 3.11.2. Cell Wall Reorganization

Cell wall remodelling is a crucial process of plant growth and, according to recent findings, seems to be highly affected by altered gravity conditions.

Reanalysis of transcriptomic datasets from Arabidopsis grown on the ISS using a novel combination of graph-theoretic tools to generate gene regulatory networks produced a candidate list of hub genes that have a regulatory impact on a large group of targets. Among these hub genes are members of the xyloglucan endo-trans glycosylase/hydrolases (XTHs) family, which are directly involved in cell wall remodelling, elongation growth, and even skewing behavior [238].

Arabidopsis seedlings subjected to fractional gravity in the European Modular Cultivation System (EMCS) on the ISS had around 100 differentially regulated transcripts that showed patterns matching the gravity gradient. The most enriched gene ontology categories were transcriptional regulation, heat stress response, and cell wall organization [239]. A similar study with the addition of a blue light stimulus also found an enrichment of cell wall and stress categories across the gravity gradient [240]. Transcriptomic studies on Arabidopsis seedlings in the Biological Research In Canisters (BRIC) hardware aboard the ISS show corresponding results. Peroxidases, likely to play a role in cell wall remodelling, were consistently downregulated in wild-type plants of different ecotypes [237]. Likewise, genes involved in various stress mitigation pathways, such as DNA repair, temperature, light, and oxidative stress, were enriched [237,241]. Thereby, heat shock proteins and factors were among the most prominently upregulated genes, even though the temperature was tightly controlled during the experiment, leading to the assumption that their role in other stress responses caused their induction [237].

Experiments with rice shoots in the Cell Biology Experiment Facility (CBEF) on the ISS could confirm a decrease in glucans, which are an integral part of the cell wall in various crops, in conjunction with the transcriptional upregulation of a glucanase, that breaks down the glucans in the cell wall under microgravity conditions [242]. In addition, regulation of cell wall reorganization was also confirmed in rice calli aboard the Chinese spacecraft Shenzhou-8 [243].

Cell wall stability maintained by pectin methylesterase (PME) is affected by microgravity, as shown in Arabidopsis seedlings under simulated microgravity and real microgravity conditions [244]. PME activity was decreased by clinostat-induced simulated microgravity, and transcriptional analysis identified *AtPMEPCRA* to be downregulated and, thus, a possible regulator of PME activity. Mutant *atpmepcra* seedlings grown on the SJ-10 recoverable satellite for 11 days had stunted leaf growth compared to their wild-type controls in microgravity, indicating a role of *AtPMEPCRA* in microgravity-induced growth effects. Interestingly, the adaptations acquired in microgravity carried over to the F1 generation that was grown on earth and manifested as changes in the DNA methylation pattern of the *AtPMEPCRA* locus. Even though this change was lost in the F2 generation [244], there have been repeated observations of changes in DNA methylation patterns following spaceflight in Arabidopsis and soybean [245,246,247,248].

Combined transcriptomic and proteomic analysis of three-day-old, etiolated Arabidopsis seedlings in the BRIC hardware on the ISS showed a telling regulation of cell wall-related processes. Proteins involved in xylose modification, such as the XTHs mentioned above, were upregulated, while transcripts belonging to the same category were downregulated [249]. Additionally, differential transcription of genes and phosphorylation of proteins involved in auxin transport (PIN, LAX) and response (IAA, ARF, TIR1-family) was observed, representing the primary gravitropic response of plants to modulate auxin distribution in the tissues. Differential phosphorylation of the plasma membrane ATPase AHA2 [249] supports the assumption that AHA2 is involved in cell elongation through acidification of the cell wall in response to an auxin stimulus [250].

#### 3.11.3. Plastid Dysregulation

An uncoupled regulation of transcripts and proteins was also found in the plastids of plant cells. In ground control plants, less than 1 % of differentially regulated transcripts are of plastidic origin. However, when exposed to µ*g*, the portion of differentially regulated transcripts of plastidic origin increases to 25 %. This shift was not apparent at the protein level, and, considering that many of the enriched categories are related to chloroplast metabolism and chlorophyll biosynthesis even though the plants were grown in darkness, indicate a dysregulation of plastid function [249]. Similar observations were made under the same dark conditions and with red and blue light stimuli [251,252]. Mitochondrial dysfunction under microgravity has been documented in humans [253] and drosophila [254] and, together with the recent findings in plant plastids, could suggest a conserved stress response to µ*g* in organelles of endosymbiotic origin.

#### 3.11.4. Post-Transcriptional and Translational Regulation

Multiple indications point to the importance of regulatory mechanisms in the microgravity response of plants. Inverse relationships between transcript-protein abundances under µ*g* suggest a post-transcriptional or translational regulation of the affected genes [249]. Upregulation of RNA decapping protein 5 (DCP5) and RNA silencing protein ARGONAUTE 4 (AGO4) could represent higher RNA turnover and siRNA-mediated gene silencing when exposed to µ*g* [249]. Differential phosphorylation of RNA splicing proteins and transcript isoform abundance analysis shows a direct influence of microgravity on alternative RNA splicing mechanisms [249,255]. In the same dataset, protein degradation pathways are enriched on the transcript level under µ*g* [249].

These findings clearly show the effect of spaceflight across many levels of regulation and demonstrate that a multi-omics approach can yield more reliable data, as is already the standard in other plant fields [256]. Translatomic analyses should address the discrepancy between transcriptomic and proteomic data. A third of all transcripts in Arabidopsis possess at least one upstream open reading frame (uORF), entailing a vast potential for translational regulation [257]. Translational regulation by uORFs is controlled, at least in part, by the target of rapamycin (TOR)-mediated phosphorylation of initiation factors, facilitating or hindering re-initiation at downstream main ORFs [258], which was shown to be triggered by µ*g* conditions [249].

#### 3.11.5. Experimental Setup and Controls

Studies showing effects allegedly caused by µ*g* may be attributed to the peculiar environment of spaceflight, laden with abiotic stressors such as radiation, vibration, temperature differences, and lack of convection which can induce a variety of artefacts unrelated to µ*g*; these are concisely summarized in [259]. Conducting experiments in space is an incredibly challenging endeavor, and past experiments were, rightfully, critiqued for poor control conditions [229]. Fortunately, some recent space experiments could use centrifuges in their setups to have a proper onboard 1*g* control, such as in the EMCS and CBEF. Other approaches include documenting the environmental conditions during the experiment in space and recreating them on the ground during the control run, neglecting impactful stressors such as cosmic radiation and the harsh conditions during sample upload and retrieval. For future planning, it is essential to implement 1*g* onboard controls, ground controls, and ground-based simulated µ*g* controls to ensure the reliability and trustworthiness of the acquired data [260]. A noteworthy strategy is measuring the effect of the hardware and procedures on the organism and considering these effects when assessing data obtained during spaceflight [249].

The most important publication focusing on gene regulation changes in plants exposed to microgravity are shown in Table 10.

## 4. Conclusions

There are new insights into the effects of µ*g* and spaceflight on prokaryotic and eukaryotic organisms, plants, and mammals, including rodents and humans. Space travelers face an increased risk of infection. Bacterial proliferation, biofilm formation, and expression of virulence genes in bacteria are enhanced, while the human immune system, on the other hand, is compromised in space. Studies also revealed significant differences in the immune response of human cells infected with *Salmonella typhimurium* in space or on Earth. The observed dysregulation of the immune system shifts the normal balanced gut microbiome towards a diseased gut biome, possibly due to cellular stress.

The complex interaction of organisms other than bacteria, such as archaea, viruses, and fungi, within the gut, plays a crucial role in the health and well-being of space travelers. However, whether µ*g* affects this intricate interplay needs to be further investigated. Increased virulence and improved antibiotic resistance of some pathogenic bacteria have been identified during µ*g* conditions. Other studies reported the reduced virulence of other bacteria strains. An explanation for discourse may be epigenetic changes introduced by either µ*g* or the space environment.

Furthermore, *Caenorhabditis elegans* studies indicated that certain genes are epigenetically suppressed, thereby supporting previous findings showing the downregulation of genes related to longevity and metabolism. Importantly, and maybe also interesting in relation to the intestinal system of humans, exposure of *Caenorhabditis elegans* to s-µ*g* induces oxidative stress damage and adaptation of insulin signaling pathways in the intestine.

A growing body of data has been collected from experiments performed on the retina from humans and mice exposed to µ*g*. In mice launched to the ISS, induction of apoptosis in the retina, especially in vascular endothelial cells, seems to be a repeating finding. Substantial evidence also points to the fact that the space environment triggers oxidative damage in the retina and reduces the thickness of multiple retina layers. In addition, genes associated with diabetic retinopathy were differentially expressed in the retina of mice flown on the ISS. Taken together, spaceflight seems to triggers epigenetic and transcriptomic reprogramming in the eye, affecting inflammation, oxidative stress, angiogenesis and macular degeneration. Finally, novel findings have investigated the involvement of cytoskeletal remodelling in ARPE-19 cells subjected to µ*g* and pinpointed vimentin as an essential player in this process [99], thereby further reconfirming the pathogenesis underlying SANS

Microvascular EC exposed to µ*g* exhibited substantial deregulated genes resulting in activating pathways for metabolism and pro-proliferative phenotype. *TXNIP*, encoding a major regulator of cellular redox signaling, which protects cells from oxidative stress, was found to be the most upregulated gene in EC following spaceflight. Deregulated miRNAs suggested that HUVECs exposed to s-µ*g* may be protected from apoptosis. However, parallel studies found a decreased expression of anti-apoptotic *BCL2*. µ*g* alters the expression of miRNAs and the interactome of miRNAs suggesting that µ*g* influences proliferation and vascular function of ECs. Finally, a number of genes were differentially regulated in adherent and spheroid populations, suggesting that µ*g* may be the primary cause for ECs’ 3D aggregation.

Chondrocytes exhibited several differentially expressed genes, suggesting that unloading influences phenotype maintenance, HIF signaling, and VEGF signaling. Some of the genes were regulated in opposite directions, probably reflecting sample differences. It seems that many genes were altered in a sex-dependent manner, thereby providing ideas for possible targets for osteoarthritis.

Several mouse-based studies investigating the impact of r-µ*g* on muscle tissue demonstrated (i) that many genes were differentially expressed in various muscle tissues, (ii) that alternative splicing occurs in the muscle transcriptomic response to µ*g*, and (iii) that the affected biological processes may provide targets for the treatment and the development of countermeasures and post-flight rehabilitation.

As in the case of cartilage and muscle, exposure to r-µ*g* changed the expression pattern of several genes in bone. The results showed a strong association with various important processes, including osteoporosis, bone resorption, and bone development. In addition, reduced gravity inhibits cell proliferation and differentiation. The expression of circRNAs was differentially expressed in s-µ*g*, resulting in the altered regulation of the actin cytoskeleton, focal adhesion, and osteogenic differentiation.

Real microgravity and s-µ*g* altered the growth behavior of human cancer cells. Various authors demonstrated a change in the phenotype of the cancer cells exposed to µ*g*. One part of the cells cultured on µ*g*-simulating devices continued growing adherently, and the other detached and formed organoids or multicellular spheroids (Table 9). Studies focusing on the underlying mechanisms for spheroid formation can improve our understanding of in vivo cancer progression, EMT, and metastasis [215]. Moreover, common results comprise alterations in the gene regulation of factors of the cytoskeleton, integrins, extracellular matrix, focal adhesions, cell adhesion, apoptosis, survival, migration, differentiation, and growth.

Finally, the identification of master regulators in the µ*g* response and characterization of genes affecting the development of plants are essential steps to broaden our understanding of plant behavior on Earth and improve their fitness for space applications. Cell wall remodelling and plastid metabolism emerge as highly affected processes and, given their crucial role in plant health and growth, make for excellent targets for further investigations. The newly discovered potential for post-transcriptional and translational regulation during spaceflight is demonstrated by the few but repeated discrepancies between transcriptomic and proteomic data. Importantly, multi-omics approaches are the tool of choice to discover the effects of spaceflight on translation.

We live in the age of space exploration. As a result, we receive new information to increase our current knowledge of the cosmos. Research on the ISS, the new Chinese Space Station, in outer space (Moon and Mars), in extreme environments, and simulation experiments of µ*g* together with new molecular biological methods such as OMICs, can define the space traveler’s health risks and contribute to health protection and to develop adequate countermeasures. These data obtained during spaceflights can support translational medicine on Earth.

## Figures and Tables

**Figure 1 cells-12-01043-f001:**
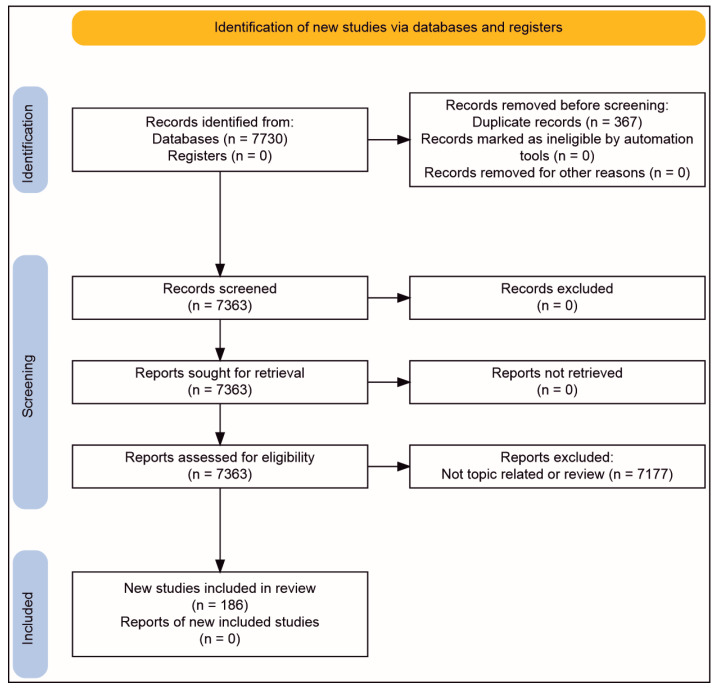
PRISMA flow diagram: literature search results for this concise review (made with https://estech.shinyapps.io/prisma_flowdiagram/), accessed on 9 March 2023.

**Figure 2 cells-12-01043-f002:**
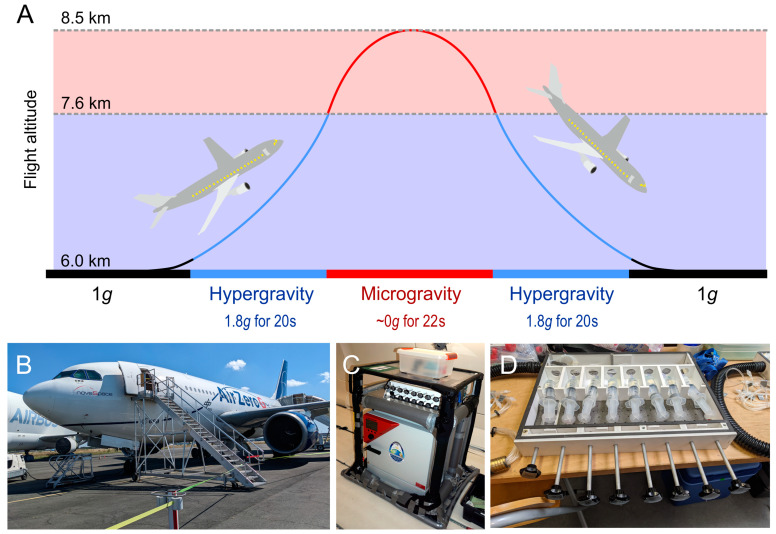
Parabolic flight campaign (PFC). (**A**) Time sequence of a parabola. (**B**) Airbus A310 AirZeroG aircraft from Novespace at the PFC in Bordeaux, France (October 2022). (**C**) The PFC flight rack with an incubator. (**D**) The injection unit of the flight rack during preparation. It is used for fixation of the cells with RNA*later* or other fixatives.

**Figure 3 cells-12-01043-f003:**
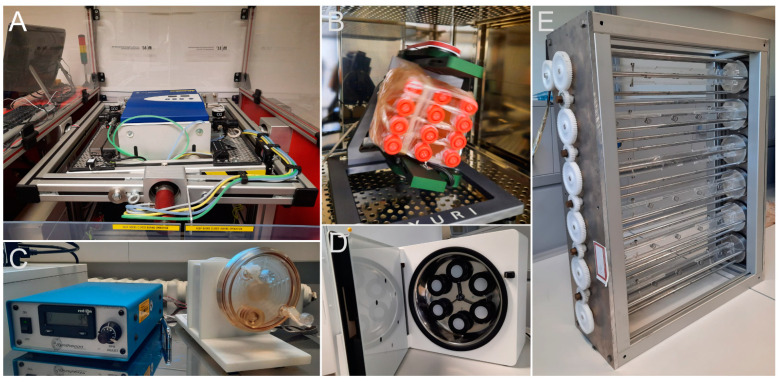
Ground-based s-µ*g* facilities: (**A**) The Random Positioning Incubator (RPI) developed by the ‘*Fachhochschule Nordwestschweiz*’ (FHNW) and the ‘*Eidgenössische Technische Hochschule*’ (ETH) Zurich, Switzerland, (**B**) a desk-top random positioning machine housed in an incubator purchased from Yuri GmbH Meckenbeuren, Germany, (**C**) the NASA-developed Rotating Wall Vessel, (**D**) the ClinoStar CO_2_ incubator with integrated clinostat (CelVivo ApS, Odense, Denmark), and (**E**) a 2D fast-rotating clinostat for adherent cells in slide flasks (developed by the German Space Agency, Cologne, Germany).

**Figure 4 cells-12-01043-f004:**
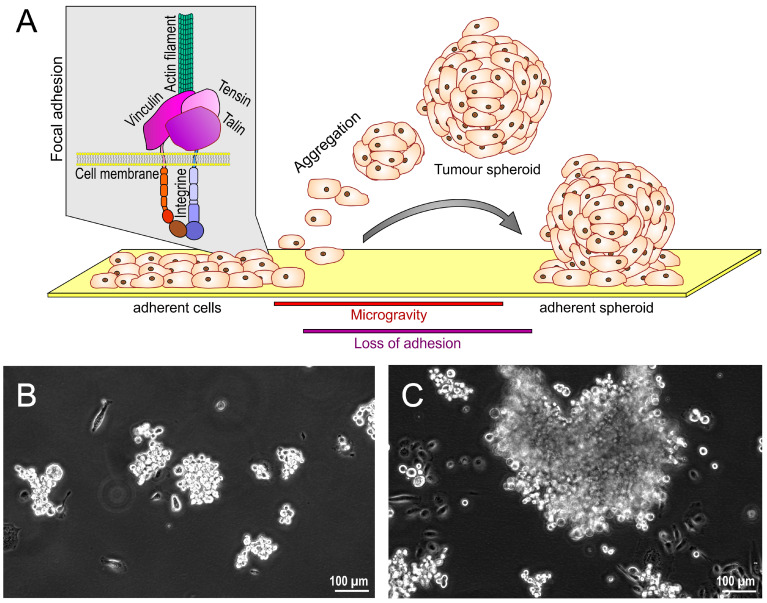
Microgravity-induced in vitro metastasis model. (**A**) µ*g* exposure of adherent tumor cells causes downregulation of focal adhesion molecules. As a consequence, cells detach and form tumor spheroids resembling micrometastases. When gravity is restored, the spheroids reattach to their substrate. (**B**,**C**) Transmitted light microscopy images of spheroid development in PC-3 prostate tumor cells under s-µ*g* (RPI). The graphical representation of cell adherence and spheroid formation is inspired by Grimm et al., 2022, Figure 5 [215].

**Figure 5 cells-12-01043-f005:**
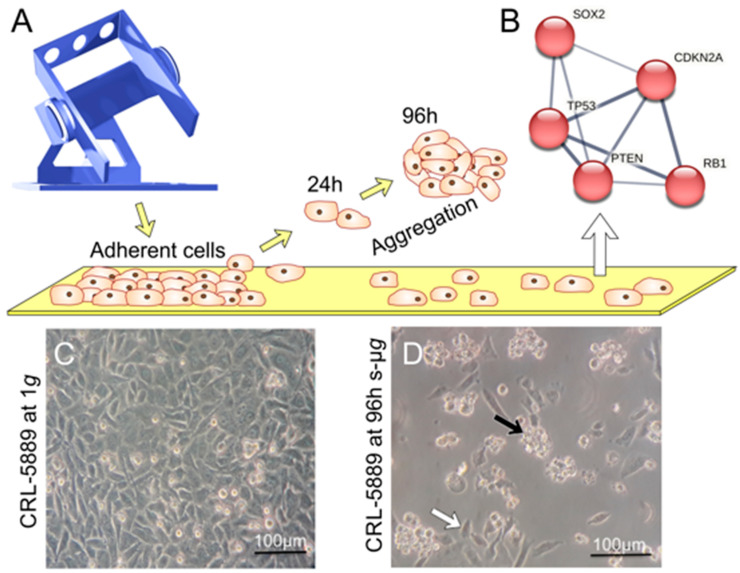
The effects of a four-day RPM exposure on the human squamous non-small-cell lung cancer cell line CRL-5889 according to Dietz and co-workers [196] (**A**–**D**). (**A**) The RPM exposure leads to partial detachment of cells from the bottom of the flask and after 24 h to first aggregations (spheroid formation) that further increase after 96 h. (**D**, black arrow). However, one part of the cells remains attached to the bottom of the bottle (**D**, adherent cells (AD), white arrow). These AD cells exhibit an upregulation of the *PTEN*, *RB1*, *TP53*, *CDKN2A*, and *SOX2* gene expression, whose protein-to-protein interactions were shown by STRING analysis (**B**). In contrast under 1*g* on 2D cell culture plate the cells stay in a state of near confluence (**C**). These results are published in [196].

**Figure 6 cells-12-01043-f006:**
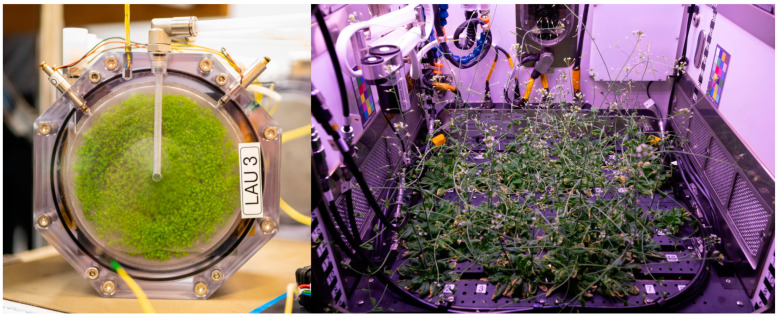
The model organism *Arabidopsis thaliana* is used in various microgravity research applications. (**Left**): 5-day-old seedlings in a fixation chamber moments before integration into a sounding rocket during the TEXUS 57 campaign (hardware designed by Airbus Defence and Space). (**Right**): Flowering Arabidopsis plants inside the Plant Habitat-01 in the European Columbus laboratory module aboard the ISS (image courtesy of NASA).

**Table 1 cells-12-01043-t001:** Key parameters of different microgravity platforms.

Platforms	µ*g* Duration	Residual Acceleration	Preparation Time	Cost Estimates [16]	Reference
Bremen Drop Tower: drop catapult	4.74 s 9.3 s	<10^−6^ × *g* <10^−6^ × *g*	few months	≈5 k€	[17,18]
NASA Zero Gravity Research Facility	5.18 s	<10^−5^ × *g*	few months	≈5 k€	[19,20]
Einstein-Elevator: drop lift/drop	2 s 4 s	<10^−6^ × *g* <10^−6^ × *g*	n/a	n/a	[21]
Parabolic flight	22 s	<10^−2^ × *g*	months	≈125 k€	[22]
Sounding rocket	6 min, 13 min	<10^−4^ × *g*	years	>400 k€	[23]
New Shepard	3 min	<5 × 10^−3^ × *g*	years	n/a	[24]
Satellites, taxi flights	days to weeks	<10^−5^ × *g*	years	n/a	[25,26,27]
ISS, Tiangong	months to years	>10^−6^ × *g*	years	≈1–5 M€	[28]

Abbreviations: Gravity (*g*); not available (n/a); million (M); seconds (s); thousand (k).

**Table 2 cells-12-01043-t002:** Changes in gene regulation in microorganisms and animals exposed to microgravity conditions.

**Organisms and Conditions**	**Observations**	**Reference**
U937 Cells (RCCS) and infection with *Escherichia coli*, mice (HU), gut bacteria, and investigation of intestinal immunity to *Citrobacter rodentium*	Suppressed MAPK pathway under s-µ*g* in U937 cells. Downregulation of 25 transcripts and upregulation of 11 transcripts of the MAPK signaling pathway. In mice: suppression in the production of TNF and IL-6 in colon and spleen. Higher sensitivity against *Citrobacter rodentium* due to suppressed innate immune response.	[86]
Human colonic epithelial cell line, HT-29 (ATCC, HTB-38) infected with *Salmonella typhimurium*	Differences in gene expression between space- and ground-infected cells (among others, TNF signaling, NF-*κ*B signaling, and cellular response to lipopolysaccharides).	[69]
HU mice, investigation of the gut biome	Dysbiosis followed after HU, which could be mitigated with 4-PBA, indicating that unloading leads to cellular stress.	[60]
*Lactobacillus reuteri* in RWV	These important probiotic bacteria respond with increased expression of stress genes.	[58]
Wheat and endophytic bacteria under s-µ*g*	Change in root metabolism and secretion of metabolites. Change in bacteria composition of the rhizosphere.	[61]
*Yersina pestis*, HARV Cultivation	218 differentially regulated genes. Increased biofilm formation and reduced virulence (downregulation of virulence-associated genes).	[70]
*Serratia marcescens* and *Drosophila melanogaster*, HARV cultivation	Increased virulence of *Serratia marcescens* against *Drosophila melanogaster.*	[74]
*Drosophila melanogaster* on the ISS	Flies reared on the ISS showed smaller and weaker hearts, reduced sarcomeric and ECM gene expression, upregulated expression of proteasome subunit genes, and increased number of proteasomes.	[87]
*Listeria monocytogenes* (LSMMG)	Downregulation of virulence genes. Decreased heat and acid resistance, but increased cold resistance (upregulation of cold-stress genes).	[75]
*Bacillus subtilis*, Spaceflight	Increased mutation rate of rpoB.	[76]
*Escherichia coli* long-term HARV cultivation	Genome sequencing revealed an increased mutation rate, which resulted in the acquirement of antibiotic resistance.	[77]
*Acinetobacter pittii* and S*taphylococcus aureus* HARV cultivation	Increased rate of horizontal gene transfer.	[71]
*Vibrio fischeri* HARV cultivation	Impaired membrane stability, increased lipopolysaccharide production.	[81]
*Knufia chersonesos* (colorless and pigmented strain) HARV cultivation	Changes in the secretome, transcriptome, and proteome between 1*g* samples as well among the two strains. No increased stress response in the colorless strain.	[82]
*Cryptococcus neoformans* (melanized and colorless strain) on the ISS	Higher survival rate of melanized strains. Melanin acts as an antioxidant.	[83]
*Eprymna scolopes* HARV co-cultivation with *Vibrio fischeri*	Complex network of extrinsic/intrinsic apoptosis genes revealed; earlier and stronger expression of caspases.	[88]
*Caenorhabditis elegans* on the ISS	Downregulation of metabolism and cytoskeletal genes.	[89]
	Expression levels are regulated epigenetically.	[90]
*Caenorhabditis elegans* on the RCCS	Global upregulation of genes related to oxidative stress. Local overexpression prevented dysfunction but not increased lumen and permeability of the intestines.	[91]
	Expressional adaptations of the insulin signaling pathway in the intestines.	[92]
*Mus musculus* on the ISS	No estrous cycle discontinuation. No difference in the expression of genes related to steroidogenesis or mitochondrial cholesterol uptake.	[93]
	Thymi of ISS mice were smaller, with downregulated genes related to cell cycle control and chromosome organization.	[94]
	Erythrocyte production-related genes in the spleen were downregulated. No influence on gene expression in lymph nodes.	[95]

Abbreviations: 4-phenyl butyric acid (4-PBA); extracellular matrix (ECM); high aspect ratio vessels (HARV); hindlimb unloaded (HU); low-shear modelled microgravity (LSMMG); International Space Station (ISS); mitogen-activated protein kinase (MAPK); nuclear factor-kappa B (NF-*κ*B); rotary cell culture system (RCCS); rotating wall vessel (RWV); tumor necrosis factor (TNF).

**Table 4 cells-12-01043-t004:** Changes in gene regulation of endothelial cells exposed to microgravity conditions.

Cell Type	Gene Regulation	Microgravity	Reference
Human microvascular endothelial cell line HMEC-1	Differentially regulated Hallmark collection gene sets: Up: HALLMARK_ADIPOGENESIS, HALLMARK_ ALLOGRAFT_REJECTION, HALLMARK_ ANDROGEN_RESPONSE, HALLMARK_BILE_ACID_ METABOLISM, HALLMARK_CHOLESTEROL_ HOMEOSTASIS, HALLMARK_COAGULATION, HALLMARK_E2F_TARGETS, HALLMARK_FATTY_ACID_METABOLISM, HALLMARK_G2M_CHECKPOINT, HALLMARK_ MITOTIC_SPINDLE, HALLMARK_MTORC1_SIGNALING, HALLMARK_MYC_TARGETS_V1, HALLMARK_ OXIDATIVE_PHOSPHORYLATION, HALLMARK_ PEROXISOME, HALLMARK_PI3K_AKT_MTOR_ SIGNALING, HALLMARK_PROTEIN_SECRETION, HALLMARK_REACTIVE_OXIGEN_SPECIES_PATHWAY, HALLMARK_SPERMATOGENESIS, HALLMARK_UV_ RESPONSE_DN Down: HALLMARK_ESTROGEN_RESPONSE_EARLY, HALLMARK_ESTROGEN_RESPONSE_LATE, HALLMARK_HEDGEHOG_SIGNALING, HALLMARK_HYPOXIA, HALLMARK_IL2_STAT5_ SIGNALING, HALLMARK_KRAS_SIGNALING_DN, HALLMARK_MYOGENESIS, HALLMARK_NOTCH_ SIGNALING, HALLMARK_P53_PATHWAY, HALLMARK_TNFA_SIGNALING_VIA_NFKB, HALLMARK_UV_RESPONSE_UP, HALLMARK_WNT_BETA_CATENIN_SIGNALING	r-µ*g* on the ISS for 160 h	[108]
HUVECs	585 genes significantly upregulated (top 5: *TXNIP*, *MIR15A*, *ANAPC1*, *TP53INP1*, *ID1*); 438 genes significantly downregulated (top 5: *HSPA1A* and -*B*, *HSP90AA2*, *ATF3*, *CLCA2*, *RYBP*)	r-µ*g* on the ISS for 10 d	[109]
HUVECs	Early upregulation of *HSPA1A* only after 4 days, string upregulation of *TXNIP* after 10 days	s-µ*g* on the RWV for 4 and 10 d	[110]
HUVECs	Downregulation of *SLCO2A1*	s-µ*g* on a clinostat for 7 d	[111]
HUVECs	No significant difference in *TP53* gene expression	s-µ*g* on a clinostat for 24, 48, and 73 h	[112]
HUVECs	Upregulated miRNAs: hsa-miR-628-3p, hsa-miR-3195, hsa-miR-3687, hsa-miR-1257, hsa-miR-3614-5p	s-µ*g* on a clinostat for 48 h	[113]
HUVECs	Upregulation of miR-22 after 72 h, concurrent downregulation of *SRF* and *LAMC1*	s-µ*g* on a HARV for 48, 72 and 96 h	[114]
HUVECs	1870 differentially expressed miRNAs; hsa-mir-496, hsa-mir-151a, hsa-miR-296-3p, hsa-mir-148a, hsa-miR-365b-5p, hsamiR-3687, hsa-mir-454, hsa-miR-155-5p, and hsa-miR-145-5p involved in cell adhesion, angiogenesis, cell cycle, JAK-STAT signaling, MAPK signaling, nitric oxide signaling, VEGF signaling, and wound healing pathways	s-µ*g* on a 3D-clinostat for 2 h	[115]
Human CVECs	Upregulation of *BAX*, *CASP3*, and *CYP2D6* after 1 and 3 days Downregulation of *BCL2* after 1 and 3 days	s-µ*g* on an RCCS for 1 and 3 d	[116]
EA.hy926	Upregulation of *CXCL8* and *FN1* in spheroids	s-µ*g* on an RPM for 35 d	[117]
EA.hy926	Differential regulation of *TIMP1*, *IL6*, *CXCL8*, *CCL2*, *B2M*	r-µ*g* on the ISS for 12 d	[118]
EA.hy926	Top 20 upregulated genes in EA.hy926 exosomes: *ACTB*, *PGK1*, *HSPA8*, *RPL7A*, *FTH1*, *GAPDH*, *HSP90AB1*, *RPL37A*, *TPI1*, *FABP4*, *S100A6*, *TOT1*, *TUBA1B*, *RPS4X*, *RPL5*, *HNRNPK*, *RPS3A*, *RPS2*, *HNRNPA1*	r-µ*g* on the SJ-10 Recoverable Scientific Satellite for 3 and 10 d	[119]
Human peripheral blood-derived endothelial progenitor cells	Increase in *HIF1A* and *NOS3* after 12 and 24 h, then decrease after 48 h	s-µ*g* on a 3D-clinostat for 24 h	[120]

Abbreviations: Days (d); high aspect ratio vessels (HARV); head-down tilt (HDT); hours (h); International Space Station (ISS); random positioning machine (RPM); real microgravity (r-µ*g*); rotating cell culture system (RCCS); simulated microgravity (s-µ*g*); three-dimensional (3D).

**Table 5 cells-12-01043-t005:** Changes in gene regulation of immune system components exposed to microgravity conditions.

Cell Type	Gene Regulation	Microgravity	Reference
Mouse splenic dendritic cells and mouse Flt-3L-differentiated BMDCs	Downregulation of *IL6*, *IL1B*, *IL12B*, and *CXCL10*	s-µ*g* on an RPM for 24 h	[130]
Human blood plasma and peripheral blood mononuclear cells from astronauts	Significant increase in cell-free mitochondrial DNA with high variability (between 2- and 355-fold); *IL6*, *IL8*, *SOD1*, *SOD2*, *GPX1*, *NOX4*, *GADD45*, *CAT1*, *DNA-PK*, and *PARP1* elevated in at least 1 of the postflight time points	r-µ*g* on the ISS for 5–13 d	[131]
Whole blood samples from astronauts	The most commonly mutated gene was *TP53* (7 variants) followed by *DNTM3A* (6 variants) accounting for 38% of mutations detected	r-µ*g* on the ISS (median 12 d)	[132]
Human plasma from two male monozygotic twins, one on Earth, one in space (NASA Twins study)	Significant increase in the proportion of cell-free mitochondrial DNA over time	r-µ*g* on the ISS for 340 d	[133]
Humans blood from two male monozygotic twins, one on Earth, one in space (NASA Twins study)	IL-6, TNFR1, and TNFR2 non-canonical NF-kB pathways were affected	r-µ*g* on the ISS for 340 d	[134]
Human blood from two male monozygotic twins, one on Earth, one in space (NASA Twins study)	No persistent alteration of long-chain fatty acid desaturase and elongase gene expression associated with 1 year in space	r-µ*g* on the ISS for 340 d	[135]
Human Jurkat T cells	5 µg-sensitive transcript clusters: *G3BP1*, *KPNB1*, *NUDT3*, *POMK*, *SFT2D2*	r-µ*g* on a sounding rocket for 5 min and s-µ*g* on a clinostat for 5 min	[136]
Human cell line U937 and human Jurkat T cells	*HIF1A* was differently regulated in early response to altered gravity, quick adaption of HIF-1α-dependent gene expression (only *IL1B* continuously downregulated under µ*g* and SERPINE1, PDK1, and SLC2A3 continuously upregulated under hyper-*g*)	r-µ*g* on a parabolic flight for 20 s and r-µg on a sounding rocket for 5 min	[137]
Human Jurkat T cells	Gene expression upregulation in chromosome 18 and downregulation in chromosome 19 for all alterations of r-µ*g* conditions	r-µ*g* on a parabolic flight for 20 s, r-µ*g* on a sounding rocket for 5 min, and s-µ*g* on a clinostat for 5 min	[138]
Human myelomonocytic U937 cells and human Jurkat T cells	U937 cells: differentially regulated transcripts during parabolic flight and sounding rocket missions: *CYBA*, *PTGS1*, *PXDN*, and *ALOX12*, *GCLM*, *GSR*, *MSRA*, *MT3*, *OXSR1*, *PRDX4*, *PRNP*, *PTGS2*, *SELS*, *SOD1*, respectively. Jurkat T cells: no response of oxidative stress-related transcripts to altered gravity	r-µ*g* on a parabolic flight for 20 s, r-µ*g* on a sounding rocket for 5 min	[139]
Human Jurkat T cells	5 transcripts found to be gravity-regulated in both independent experiments: *ATP6V1A*, *ATP6V1D*, *IGHD3-3*, *IGHD3-10*, *LINC00837*	r-µ*g* on a parabolic flight for 20 s, r-µ*g* on a sounding rocket for 5 min	[140]
Mouse hematopoietic progenitor cells	Downregulation of genes of the RAS/ERK/NFκB signaling pathways	r-µ*g* on the Tianzhou-1 cargo ship for 12 days and r-µ*g* on the SJ-10 recoverable satellite for 12 d	[67]
Male C57BL/6 J mice	Downregulation of *GATA1* and *Tal1*, downregulation of GATA and *Tal1*-mediated transcripts	r-µ*g* on the ISS for 35 d	[95]
Female C57BL/6J mice	Selected genes involved in glycogen metabolism: *Foxo1*, *Gbe1*, *Gys2*, *Ppp1cb*, *Ppp1ca*, *Gsk3b*, *Pck1* upregulated; *Pygl* downregulated	r-µ*g* on Space Shuttle mission STS-135 for 13 d	[141]
Human Jurkat T cells	Resting T cells: F*LT1*, *OTULIN*, *GPBP1L1*, *TP53BP1*, *STK38*, *PDS5A*, *PIK3R4*, *ABCC5*, *BRWD3*, *HSPH1*, *BRD8*, *NAPB*, *SLC5A3* upregulated; *CXCR3*, *RBM3*, *C19orf60*, *APOBEC2C*, *SNHG11*, *SNHG17*, *ASPSCR1*, *SFXN2*, *DCAF16* downregulated Activated T cells: *KDM58* upregulated; *RBM3*, *HES1*, *CXCR3*, *TUBA4A*, *HCST*, *ARID5A*, *LIMD2*, *YRDC*, *LY9*, *APOBEC3C*, *LANCL2*, *RNASEH1*, *STT3B*, *PHF7* downregulated	s-µ*g* on an RPM for 24 h	[142]
Murine macrophage RAW 264.7 cells	M0 phenotype: *Cd86*, *Actb* upregulated, *Arg1* downregulated M1 phenotype: *Cd86*, *Mrc1*, *Actb* upregulated M2 phenotype: *Cd86*, *Mrc1*, *Arg1*, *Actb* upregulated	s-µ*g* on an RWV for 3 d	[143]
Human peripheral blood mononuclear cells	s-mg decreased effects of 1.5 h-long ConA/anti-CD28 stimulation on *IL2RA*, *TNFA*, *CD69*, and *CCL4*	s-µ*g* on a HARV for 18 h	[144]
Human peripheral blood lymphocytes	230 dysregulated transcription factor and microRNA feed-forward loops were found in µ*g* including immune, cardiovascular, endocrine, nervous, and skeletal system subnetworks	s-µ*g* on an RWV for 24 h	[145]
Human peripheral blood mononuclear cells	The Radio Electric Asymmetric Conveyer technology could increase *IL2* and *IL2R* gene expression under s-µg	s-µ*g* on an RPM for 2, 4 and 12 h	[146]
Human peripheral blood mononuclear cells	*BAX*, *CASP3*, *PCNA*, *LIG4*, and *MDM2* were positively correlated with all cytokines, expressions of *AKT1*, *TP53*, *PARP1*, *OGG1*, and *APXE*1 were negatively correlated with these cytokines	s-µ*g* on an RWV for 24 h (+10μM (-)-isoproterenol hydrochloride and/or 0.8 or 2 Gy radiation)	[147]
Female C57BL/6J mice	Top 5 upregulated: *Slc22a4*, *Wt1os*, *1700063H04Rik*, *Eral1*, *Zfp341* Top 4 downregulated: *Tmem161b*, *Prrc1*, *Kbtbd8*, *Gm33989*, *Vps39*	s-µ*g* by hindlimb unloading + 0.04 Gy irradiation	[148]
*Danio rerio* embryos	Both RLR and TLR signaling pathways were enriched with upregulated genes upon poly(I:C) stimulation and enriched with downregulated genes under s-µ*g* conditions (*trim25*, *nlrx1*, *traf6*, and *traf2a* and *traf6*, *tlr7*, respectively)	s-µ*g* on an RCCS for 24 h	[149]

Abbreviations: Bone marrow dendritic cells (BMDCs); days (d); high aspect ratio vessels (HARV); head-down tilt (HDT); hours (h); International Space Station (ISS); random positioning machine (RPM); real microgravity (r-µ*g*); rotating cell culture system (RCCS); Rotating wall vessel (RWV); simulated microgravity (s-µ*g*); three-dimensional (3D).

**Table 6 cells-12-01043-t006:** Changes in gene regulation of cartilage exposed to microgravity conditions.

Tissue/Cell Type	Gene Regulation	Microgravity	Reference
Mouse articular cartilage	Upregulated: *Igkv6-20*, *Eif3m*, *Stfa2*, *Ms4a3*, *Gstm2*, *Gm10417*, *Top2a*, *Rbm3*, *Ifitm6*, *Igkv4-91* Downregulated: *Omd*, *Olfr1437*, *Gm6432*, *Fmod*, *Gm10673*, *Ogn*, *C1s*, *Olfr118*, *Olfr1454*, *Serpina1b*, *Olfr764*, *Gsn, Ccdc80*, *Slc35e3*, *Olfr948*, *Clu*, *Dcn*, *Olfr347*, *Olfr1014*, *Gm21428*, *Ect2l*, *Dpt*, *Prg4*, *Prelp*, *Olfr338*, *Angptl7*, *Col10a1*, *Retnla*, *Myoc*, *Thbs4*, *Pcolce2*, *Cyp2e1*, *Cxcl13*, *Clec3a*, *Comp*, *Ecrg4*, *Cytl1*	r-µ*g* spaceflight BION, 30 d	[150]
Mouse sternal cartilage	Upregulated*: Klhl38*, *Acot2*, *Fbxo32*, *Nr1d1*, *Trim63*, *Htra4*, *Lox*, *Prg4, Slc43a1*, *Lmod2, Aldoc*, *Slc39a8*, *Etv5*, *n-R5s88, Chac1*, *Tango2*, *Pdk4*, *Chi3l1*, *Impdh2*, *Zfp600*, *Sesn1, Myf6*, *Cfhr2*, *Tsen15, Omd*, *Inmt*, *Sgcg*, *Gm5886*, *Tacc2*, *Ankrd1* Downregulated*: Taf1d*, *Hba-a2*, *Mki67*, *Slc4a1*, *Atp6v0d2*, *Sfrp2*, *Car1, Nr4a1*, *Mpo*, *Svs3b*, *Hist1h3f*, *Retnlg*, *Hist1h3a*, *Top2a, Hist1h3d*, *Hist1h3i*, *Hist1h2ab*, *Alpl*, *Hist2h3c2*, *Hbb-bs, Hbb-bt, Col1a1*, *Acp5*, *Hp*, *Car2*, *Igkv4-55*, *Gypa*, *Ltf*, *Igkv1-117*, *Ibsp*, *Mmp13*, *Mmp9*, *S100a9*, *S100a8*, *Ngp*		
Meniscus constructs from healthy human meniscus fibrochondrocytes (male and female donors)	Upregulation: *BMP8A*, *CD36*, *COL10A1*, *COL9A3*, *FGF1*, *IBSP*, *IHH*, *MMP10*, *PHOSPHO1*, *S100A1*, *SPP1* Only in a subpopulation of high-responding (related to *COL10A1* expression) cells from female donors	s-µ*g* on an RCCS for 3 weeks	[151]
Tissue-engineered menisci from human female and male meniscus fibrochondrocytes	Top 10 Upregulated: *IGFBP1*, *OLFML2A*, *NET O 1*, *ADAMTS14. PCSK9*, *VSTM2L*, *BMPER*, *NTM*, *HMOX1*, *CAPG* Top 10 Downregulated: *LEP*, *R3HDML*, *APLN*, *STC1*, *NGF*, *TGFA*, *VEGFA*, *DSCAML1*, *ADAMTSL2*, *PDE4C*	s-µ*g* on an RCCS for 3 weeks	[152]
Primary bovine chondrocytes	Downregulation: *TRPC1* (5- to 10-fold)	s-µ*g* on the RPM for 6 and 8 d	[153]
Human articular chondrocytes	Upregulation of *IL6*, *RUNX2*, *RUNX3*, *SPP1*, *SOX6*, *SOX9*, and *MMP13*	s-µ*g* on the RPM for 24 h	[154]

Abbreviations: Days (d); hours (h); random positioning machine (RPM); real microgravity (r-µ*g*); rotating cell culture system (RCCS); simulated microgravity (s-µ*g*).

**Table 7 cells-12-01043-t007:** Changes in gene regulation of muscle exposed to microgravity conditions.

Muscle Type	Gene Regulation	Microgravity	Reference
Mouse gastrocnemius and quadriceps muscles	DEGs in both muscles: *Atp5g2*, *Peg3*, *Mid1*, *Agpat2*, *Cyfip2*, *Pgk1*, *Rpl37a*, *Nav2*, *Rpl24*, *Hspb7*, *Fabp3*, *Agbl1*, *Bdh1*, *Ccnb1ip1*, *H2afz*	r-µ*g* on the ISS for 9 weeks	[155]
Mouse soleus and extensor digitorum longus muscles	DEGs in both muscles: *Orm1*, *Myo5a*, *Hp*, *Adipoq, Lcn2*, *Snhg5*, *Acp5, Snhg1*, *Cdkal1, Cfd*, *Cdkn1a*, *Myf6*, *Eif4ebp1, Synj2*, *Sesn1*, *Pnmt*, *Snhg1, Npr3*, *Sorbs1*, *Aspn*, *Ung*, *Mafb*, *Tfrc*, *Dbp*	r-µ*g* on a BION-MQ-capsule for 30 d	[156]
Mouse soleus muscle	A total of 1130 differentially expressed genes were found between wild-type and *Nrf2* knock-out mice; top 10 enriched pathways: mitochondrial dysfunction, oxidative phosphorylation, sirtuin signaling pathway, remodelling of epithelial adherens junctions, ILK signaling, epithelial adherens junction signaling, integrin signaling, germ cell–sertoli cell junction signaling, actin cytoskeleton signaling, EIF2 signaling	r-µ*g* on the ISS for 31 d	[157]
Mouse soleus muscle	Identification of voltage-dependent calcium channel gamma-1 subunit (*Cacng1*) as a novel candidate gene for muscle atrophy	r-µ*g* on the ISS for 35 d	[158]

Abbreviations: Days (d); differentially expressed genes (DEGs); International Space Station (ISS); real microgravity (r-µ*g*).

**Table 8 cells-12-01043-t008:** Changes in gene regulation of bone cells exposed to microgravity conditions.

Cell Type	Gene Regulation	Microgravity	Reference
Mouse Ocy454	Top 5: 2 d upregulated: *Cxcl14*, *Mmp13*, *C2cd4a*, *Cdc25c*, *Gdf7* 2 d downregulated: F*abp3*, *Ptprz1*, *Sparcl1*, *Slc12a2*, *Ogn* 4 d upregulated: *Scgb1b12*, *Acan*, *Taar5*, *Pappa2*, *Cfap20dc* 4 d downregulated: *Slc12a2*, *Tfrc*, *Ajuba*, *Idi1*, *Dmp1* 6 d upregulated: *Car9*, *Egln3*, *Serping1*, *Tmem45a*, *Selenbp1* 6 d downregulated: *Cth*, *Ctps*, *Pinx1*, *Ddx21*, *Ostn*	r-µ*g* on the ISS for 2, 4, and 6 d	[159]
Human blood-derived stem cells	Methylation at histone H3 sites H3K4me3, H3K27me2/3, H3K79me2/3, and H3K9me2/3 mediates cellular reprogramming that drives gene expression	r-µg on the ISS for 72 h	[160]
Murine osteoblasts and osteoclasts	Downregulation: *Atf4*, *RunX2*, *Osterix*	r-µ*g* on the ISS for 14 d	[26]
Human bone marrow mesenchymal stem cells	Downregulation: *CDKN3*, *MCM5*, *CCNB1*, *CDK1*, *RUNX2*, *ALPL*, *BMP2*, and *COL1A1* Upregulation: *RUNX2*, *ALPL*, *BMP2*, and *COL1A1*	s-µ*g* on an RPM for 2, 7, and 14 d	[161]
Human fetal osteoblast cells	Differential gene expression of *TGFB1*, *BMP2*, *SOX9*, *ACTB*, *TUBB*, *VIM*, *LAMA1*, *COL1A1*, *SPP1*, and *FN1*	s-µ*g* on an RPM for 7 and 14 d	[162]
7F2 murine pre-osteoblasts	Downregulation: *Alpl*, *Run*, and *on* after 6 days, no difference between *g*-levels	s-µ*g* (“full”, 0.16 *g* (Moon), and 0.38 *g* (Mars)) on an RPM for 2, 4, and 6 d	[163]
MC3T3-E1 mouse pre-osteoblasts	Identification of 3 core circRNAs *(circ_014154*, *circ_010383*, and *circ_012460*) and six core mRNAs (*Alpl*, *Bg1ap*, *Col1a1*, *Omd*, *Ogn*, and *Bmp-4*) involved in osteogenic differentiation under µ*g*	s-µ*g* on an RCCS for 72 h	[164]

Abbreviations: Days (d); hours (h); random positioning machine (RPM); real microgravity (r-µ*g*); rotating cell culture system (RCCS); simulated microgravity (s-µ*g*).

**Table 10 cells-12-01043-t010:** Gene expression changes in various plant organisms in response to spaceflight. Use of in-flight controls (FC) or ground-based controls (GC) is indicated.

Organism	Gene Regulation	Tissue	Microgravity	Reference
*Arabidopsis thaliana* (Col-0)	Upregulated: DNA replication, DNA repair, far red/red light response, photosynthesis, secondary metabolite biosynthesis Downregulated: Abscisic acid, response to stress, amino acid catabolism, sucrose starvation, absence of light, ribosome biogenesis, translation Notes: Unfolded protein response seems to play a minor role in spaceflight adaptation.	Whole seedlings, 14 days old	BRIC hardware on the ISS, 14 days of µ*g*. (GC)	[241]
*Arabidopsis thaliana* (Col-0 and Ws-0)	Differential alternative splicing in response to spaceflight in 48 genes in Ws-0 and 27 genes in Col-0.	Roots, 4 and 8 days old	VEGGIE system on the ISS, up to 8 days of µ*g*. (GC)	[255]
*Arabidopsis thaliana* (Col-0 and Ws-0)	Col-0 vs. *spr1*: Less DEGs in *spr1* than in Col-0 WT during spaceflight. Enriched categories: Response to salicylic acid, carbon fixation, RuBisCo genes, apoplast-localized genes Ws-0 vs. *sku5*: 2–5 times more DEGs in *sku5* than in Ws-0 WT during spaceflight Enriched categories: Abscisic acid signaling, response to stress, seed development, post-embryonic development	Roots, 4 and 8 days old	VEGGIE system on the ISS, up to 8 days of µ*g*. (GC)	[236]
*Arabidopsis thaliana* (Col-0, Ler-0, Ws-2, Cvi-0)	Upregulation across all ecotypes: *HSP101*, *HSP70*, *HSP23.5*, *HSP81.1*, AT2G32120.2 Downregulation across all ecotypes: AT5G44417.1, *RCI3*, AT1G64370.1, *AtPrx22* Notes: Enrichment of 21 upregulated heat shock protein/factor transcripts and 14 downregulated peroxidase transcripts across some but not all ecotypes show characteristics of oxidative stress.	Whole seedlings, 8 days old	BRIC hardware on the ISS, 8 days of µ*g*. (GC)	[237]
*Arabidopsis thaliana* (Ler-0)	Enriched GO categories per g-level: µ*g*—response to light, photosynthesis 0.1*g*—response to stress, chemicals 0.3*g*—cell wall, membrane, plastid Decrease in DEGs with higher g-levels Changes in expression along the g-gradient: Mitochondria, plastid, cell wall, and cell membrane-related processes. F-box/RMI-like/FDB-like domain family genes (AT5G44980, AT5G56370, AT5G56380, AT5G42460) Notes: Highest number of DEGs in 0.1 *g* condition with mostly stress-related pathways. Authors suggest competition of weak phototropism with weak gravitropism leading to stress.	Whole seedlings, 6 days old	EMCS hardware on the ISS, 6 days of µ*g*, 0.1*g*, 0.3*g*, 0.5*g*, 0.8*g*, and 1*g*. Blue light stimulus for the last 48 h. (FC)	[240]
*Arabidopsis thaliana* (Col-0)	Enriched categories: Cell wall organization (transcripts downregulated, proteins upregulated) Redox homeostasis and ROS signaling (increased transcript regulation, no protein regulation) Protein degradation (only transcripts) Auxin-related genes (*PIN*, *LAX*, *IAA*, *ARF*, *TIR1*-family) Post-transcriptional regulation (upregulation of *DCP5* and *AGO4*) Plastid translation, import to stroma, targeting to chloroplast, plastid organization, chlorophyll biosynthetic process (only transcripts) Notes: Large discrepancy between transcript changes and protein changes in plastids, suggesting plastid dysregulation under µ*g*.	Whole seedlings, 3 days old	BRIC hardware on the ISS, 3 days of µ*g*. (GC)	[249]
*Arabidopsis thaliana* (Col-0 and Ws-0)	Network analysis reveals 5 strongly regulated hub genes of the cell wall-modifying *XTH*-family.	Whole seedlings and root tips, 11–12 days old	ABRS hardware on the ISS, up to 12 days of µ*g*. (GC)	[238]
*Arabidopsis thaliana* (Col-0)	Identification of 101 genes with dosage-dependent response to g-level. Enriched categories: Transcription factors (20 genes) Chaperones (9 genes) Defense response (12 genes) Cell wall associated (16 genes) Root development (8 genes) Auxin-related (3 genes)	Roots, 5 days old	EMCS hardware on the ISS, 5 days of µ*g*, 0.53*g*, 0.65*g*, and 0.88*g*. (FC)	[239]
*Arabidopsis thaliana* (Ler-0)	Upregulated categories: Ribosome synthesis Oxidative phosphorylation Downregulated categories: Photosynthesis/antenna proteins Porphyrin and chlorophyll metabolism Protein processing in ER Starch and sucrose metabolism Carotenoid biosynthesis	Whole seedlings, 6 days old	EMCS hardware on the ISS, 6 days of µ*g* and 1*g*. Blue light stimulus for the last 48 h. (FC)	[251]
*Arabidopsis thaliana* (Ler-0)	Enriched categories and pathways: Transcription factors (WRKY, ERF, NAC, MYB) Plastidic transcription (enriched only in µ*g*) Auxin (*GH3* and SAUR genes, activated in µ*g*, inhibited at 0.3*g* + darkness) Cytokinins (activated in µ*g*, inhibited at 0.3*g*) ABA (activated in all conditions except µ*g* + darkness) Brassinosteroid (inhibited by upregulation of BRI1 suppressor 1) Jasmonic acid (upregulation of repressors) Notes: Hormone pathways are more affected by g-level than light conditions. At 0.3*g*, red light can reverse auxin pathway inhibition. Stress-related pathways are more activated at 0.3*g* than in µ*g*.	Whole seedlings, 6 days old	EMCS hardware on the ISS, 6 days of µ*g* and 0.3*g*. Red light or darkness for the last 48 h. (GC)	[252]
*Arabidopsis thaliana* (Col-0)	Upregulated categories: Metabolic process, response to stress, temperature and wounding, protein stabilization Downregulated categories: Response to stimulus, reproductive developmental process, regulation of the metabolic process, circadian rhythm, gibberellins, mRNA processes Notes: Flowering genes (53) were mostly downregulated under µg, affecting the flowering hub genes *FT* and *SOC1*. Delay of flowering time in µ*g* by 20 days.	Rosette leaves, 48 days old	Plant culture box hardware on the Chinese spacelab TG-2, 48 days of µ*g*. (GC)	[230]
*Arabidopsis thaliana* (Col-0)	Altered gene methylation in DNA methylation-associated genes, cell-wall modification genes, hormone signaling-related genes and transposable elements. Induction of transposable elements by unstable methylations in response to spaceflight.	Whole seedlings, 8.5 days old	Cultivation units on the recoverable satellite SJ-10, 60 h of µg after 6 days of pre-culturing on the ground. (GC)	[247]
*Arabidopsis thaliana* (Col-0)	Altered gene methylation and expression in the first offspring generation after spaceflight in the abscisic acid-activated pathway, protein phosphorylation, and nitrate signaling pathway. Partial retention of phenotypic differences and differentially methylated regions in the following generation. Notes: Could not confirm methylation changes in parent generation from previous study.	Whole seedlings, 11 days old, multiple generations	Cultivation units on the recoverable satellite SJ-10, 11 days of µg after 6 days of pre-culturing on the ground. (GC)	[246]
*Arabidopsis thaliana* (Col-0)	Altered gene methylation and expression of pectin methylesterase regulator gene *AtPMEPCRA* in parent and offspring generation after spaceflight.	Whole seedlings, 11 days old, multiple generations	Cultivation units on the recoverable satellite SJ-10, 11 days of µg after 6 days of pre-culturing on the ground. (GC)	[244]
*Arabidopsis thaliana* (Ws-0)	Portion of differentially expressed genes, that are also differentially methylated: Leaves: 143 of 743 Roots: 21 of 75 Increased methylation in CHG and CHH contexts in leaves during spaceflight. Notes: Nearly half of all differentially expressed genes in leaves were reactive oxygen signaling-related.	Leaves and roots, 11 days old	VEGGIE system on the ISS, 11 days of µ*g*. (GC)	[245]
*Oryza sativa*	Differential regulation of 8 cell-wall-related transcripts and 3 aquaporins.	Calli, 17.5 days old	SIMBOX hardware on the Shenzhou 8, 324 h of µ*g* and 1*g*. (FC + GC)	[243]
*Oryza sativa* (Koshihikari)	Upregulation of endo-1,3:1,3-β-glucanase (*OsEGL1*) under µg, leading to a reduced 1,3:1,3-β-glucan content in the cell wall.	Shoots, 4–5 days old	CBEF hardware on the ISS, 99 h, 127 h and 136 h of µ*g* and 1*g*. (FC)	[242]
*Glycine max* (T75 and Z9)	Increased activity of transposable elements and genomic plasticity after space mutation. Enriched gene categories affected by genomic restructuring: Stress response and stimulus—278 genes Cell wall-related—27 genes Auxin metabolism and transport—26 genes General signal transduction—77 genes	Leaves, 15 days old, 6 years of breeding after space mutation	Space mutation for 15 days of µ*g* aboard recoverable satellite Shijian 8. (GC)	[248]

Abbreviations: Biological research in canisters (BRIC); Cape Verde Islands (Cvi); Columbia (Col); differentially expressed genes (DEGs); European modular cultivation system (EMCS); flight control (FC); gene ontology (GO); ground control (GC); Landsberg erecta (Ler); Taiwan 75 (T75); science in microgravity box (SIMBOX); vegetable production system (VEGGIE); Wassilewskija (Ws); Zhexian 9 (Z9).

## Data Availability

Not applicable.

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
