# Peer review of "Current Knowledge about the Impact of Microgravity on Gene Regulation"

_cells, 2023, doi:10.3390/cells12071043_

Round 1
Reviewer 1 Report
This ms is a comprehensive review on the most recent findings in terms of the effects of microgravity on gene expression. The authors take care in providing the reader with the adequate background story and technical details and are very exhaustive in presenting the relevant literature and experimental model, ranging from single cells to humans.
I think this review will be useful for the community of researcher in the filed of space life science, a key field for the future of humanity in space. I have no improvement to suggest and think the ms can be published in its present form.
Author Response
Answers to the Reviewers
We thank the editor and reviewers for their insightful comments regarding our manuscript; we are grateful for the possibility to respond to the comments and address them in a revised manuscript. The comments from the reviewers are addressed sequentially below. A revised manuscript with revision marked in red is provided along with a clean manuscript as pdf file.
Reviewer 1
Comments and Suggestions for Authors
This ms is a comprehensive review on the most recent findings in terms of the effects of microgravity on gene expression. The authors take care in providing the reader with the adequate background story and technical details and are very exhaustive in presenting the relevant literature and experimental model, ranging from single cells to humans.
I think this review will be useful for the community of researcher in the filed of space life science, a key field for the future of humanity in space. I have no improvement to suggest and think the ms can be published in its present form.
Answer: We like to thank you very much for your kind review.
Reviewer 2 Report
This is a highly relevant review, especially considering the current race to space. Microgravity is one of the parameters to which organisms are exposed during space exploration. It is important to gather all details and information on how it affects organisms to better plan missions.
However, there are several things that you should consider. Please find below some comments and suggestions.
General comments:
There is a high focus on cancer, which would understandable for this topic, but since you’ve itemized this topic if feels that some information was left out and you end up only having one later section on cancer (3.10). Also, there is little to no information on the effects of microgravity on fungi or protists or g and seven bacteria, though you dedicate an entire chapter to these. For all other section I was expecting something similar to table 3. It would be interesting if you could do similar tables for at least some of the other sections.
For many of the sections, you can complement them with other references. E.g., there is much published on microorganisms, pathogens, and infections in space or affected by microgravity, and you’re not including these in your manuscript.
There are too many general sentences with little detail in many sections, some unclear text, and some repetition of information which should be avoided.
Make sure that all figures have a good resolution.
Also, consider revising your conclusion to make it shorter and more assertive.
Detailed comments:
Lines 34-36: this reads more like a concluding remark, consider making this a last paragraph.
Line 48: Remove “Some weeks ago,”. Since you specify the date there is no need to start the sentence with this.
Lines 50-54: What about other programs? Why the focus on ASTEMIS for the introductory paragraph?
Lines 95-96: Note that here you mention “gene expression of protists, rodents, humans, human cells, and plants”, and at the conclusion, you mention “bacteria, protists, fungi, rodents, humans, human cells, and plants”. Be consistent and avoid unnecessary repetition of information.
Lines 133-135: why did you use a different search method for the “bacteria, protists, fungi, and animals excluding mice”?
Line 150: Consider adding some lines to facilitate reading the table.
Line 172: Add the acronym “(CAS)” as you did before for other institutions in lines 51 and 52.
Lines 197-203: How about biological experiments done in cubesats?
Line 208: Write all acronyms in full when mentioning them for the first time.
Line 217: Consider adding a table similar to table 1, showing the different types of simulators, the gravity ranges, and any other relevant parameters.
Lines 261-268: Consider moving this into the previous section. And consider dividing that section into r-µg and s-µg structures and equipment.
Line 271: Why are you referring microbiomes here? Consider moving this into the following section.
Lines 279-282: This is too general information. Consider removing or replacing with detailed and referenced data.
Lines 290-310: You’re focusing on human immune systems, but this title is on bacteria, protists, and fungi. Revise this section and its title. Note that most statements should be referenced.
Line 311: This reads as a statement, not a title. Consider re-writing.
Line 314-315: You already explained this, no need to include this quotation.
Line 321: Write acronyms in full at their first mention.
Line 336: Note that scientific names must be italicized.
Line 350: How was that change you mention? Be more specific.
Line 360: If you’re mentioning the effects of µg on bacteria, why didn’t you put this in the section with the appropriate title?
Line 364: This image, even though interesting, is not adding anything to the manuscript, I suggest removing it. If you decide to keep it, revise the legend and avoid repeating the information in the main text or explaining the figure content.
Line 365: Add a reference.
Line 368-369: This does not read like a fact, more like an opinion. Avoid these types of unreferenced statements. Give more details on the mutation rates if these are known.
Line 374-375: What type of s- µg? What was the level of increase?
Lines 359-407: These two sections seem to be about the same topic, consider turning them into only one or edit and move the text to make them into clear and separate topics.
Line 407: Your title mentions microorganisms, but your table lists organisms as a title in a column where you include cell types. Please revise and make these coherent. Consider adding lines to facilitate the reading of the table.
Line 409: This section title is supposed to be on gene expression in animals, but you start the sentence by mentioning a bacterium. Please revise.
Line 417: C. elegans is a nematode, not an animal. You should revise the title of this section as well as the title.
Line 432: Drosophila is not an animal. You should revise the title of this section as well as the title.
Line 442-444: Please add a reference at the end of this sentence.
Line 459: Table 2 mentions many different experiments. Be specific.
Line 460: What same animals are these? Same as what?
Line 461: To which group are you referring?
Line 462: Revise.
Line 470: Is this section about the human eye and brain? It is not clear since you previously mentioned mice. What exactly are the differences in sections 3.4.1. and 3.4.2.? This is not clear since both titles seem to be about the same topic but then section 3.4.1. mentions endothelial cells, what is the relation of these cells with the eye or brain? Please revise.
Line 489: Please remove the word ”elegantly”, this is not appropriate.
Line 538: Which study are you mentioning?
Line 666: No need to detail where this work was presented. This information should appear on the reference.
Lines 675-726: This section has several parts of the text highlighted, please revise format.
Lines 735-874: This section title implies that the text here is all about humans. However, this is not the case. Note, for example, that your reference 128 is about mice macrophages, and from line 797 onward you’re not discussing humans anymore. In this section, you end by adding a paragraph about zebrafish. Please revise this section.
Line 737: How was this shown?
Lines 1608-1612: Consider adding this paragraph to the previous one.
Author Response
Answers to the Reviewers
We thank the editor and reviewers for their insightful comments regarding our manuscript; we are grateful for the possibility to respond to the comments and address them in a revised manuscript. The comments from the reviewers are addressed sequentially below. A revised manuscript with revision marked in red is provided along with a clean manuscript as pdf file.
Reviewer 2
Comments and Suggestions for Authors
This is a highly relevant review, especially considering the current race to space. Microgravity is one of the parameters to which organisms are exposed during space exploration. It is important to gather all details and information on how it affects organisms to better plan missions.
Answer: Thank you very much for this comment.
However, there are several things that you should consider. Please find below some comments and suggestions.
General comments:
- There is a high focus on cancer, which would understandable for this topic, but since you’ve itemized this topic if feels that some information was left out and you end up only having one later section on cancer (3.10). Also, there is little to no information on the effects of microgravity on fungi or protists or g and seven bacteria, though you dedicate an entire chapter to these. For all other section I was expecting something similar to table 3. It would be interesting if you could do similar tables for at least some of the other sections.
Answer: The focus of this review is as follows: We summarize the current knowledge of the impact of microgravity on gene regulation. We focus on different types of bacteria, protists, fungi, as well as animals and cells exposed to s- and r-µg-conditions with regard to the eye, brain, cartilage, muscle, bone, endothelium, immune system, various cancer types in human studies as well as findings in plants.
Concerning the cancer chapter 3.10 no information on the impact of microgravity on gene regulation was left out. All PubMed-listed publications published from 2017 until today were collected and discussed. New references from December 2022 and 2023 were added. The search was performed according the PRISMA recommendations.
We added more information to fungi, protists and bacteria as requested.
5 additional tables (similar to Table 3) have been included in the revised manuscript.
- For many of the sections, you can complement them with other references. E.g., there is much published on microorganisms, pathogens, and infections in space or affected by microgravity, and you’re not including these in your manuscript.
Answer: The review addressed gene expression, genetics, genome changes of bacteria, protists, fungi, animals, cells, humans and plants exposed to microgravity conditions. Papers published from 2017-2023 are included in this paper. But not all recent references can be considered for all topics. However, according to your advice, more recent publications were incorporated.
- There are too many general sentences with little detail in many sections, some unclear text, and some repetition of information which should be avoided.
Answer: We agree. The review was checked for repetitions and unclearness. We hope that all the corresponding paragraphs and sentences are now removed or modified.
- Make sure that all figures have a good resolution.
Answer: Thank you. Yes, the figures have a good resolution.
- Also, consider revising your conclusion to make it shorter and more assertive.
Answer: Thank you. We agree that the conclusion was too long. We have shortened it significantly and it is more focused now.
Detailed comments:
- Lines 34-36: this reads more like a concluding remark, consider making this a last paragraph.
Answer: We agree and have written this sentence in the end of the abstract.
Line 48: Remove “Some weeks ago,”. Since you specify the date there is no need to start the sentence with this.
Answer: We have changed this.
- Lines 50-54: What about other programs? Why the focus on ASTEMIS for the introductory paragraph?
Answer: ARTEMIS is the name of NASA's, CSA’s, JAXA’s and ESA’s program to return astronauts to the lunar surface. Space exploration and the planned Moon village starts very soon. Microgravity and other hazards will have impact on the gene expression of various genes. We mentioned the ARTEMIS 1 mission because with this mission the Artemis program series start. The crewed Artemis-2 will launch in 2024 and will give us the opportunity to get more information about the cancer risks of women.
- Lines 95-96: Note that here you mention “gene expression of protists, rodents, humans, human cells, and plants”, and at the conclusion, you mention “bacteria, protists, fungi, rodents, humans, human cells, and plants”. Be consistent and avoid unnecessary repetition of information.
Answer: We agree. The involved sections have been revised accordingly.
- Lines 133-135: why did you use a different search method for the “bacteria, protists, fungi, and animals excluding mice”?
Answer: Because the chapter was far less specific as compared to the others and a similar specific search pattern would not have rendered a sufficient number of results. Therefore, we decided to cast a wide net and then eliminate by excluding irrelevant hits. The difference in the search syntax is due to the different format requirements for search queries. Regarding the mice, we found that many of the studies about muscles, nerves and bones use mice as models, not human tissue. Available studies with mice are discussed in 3.7, 3.8 and 3.9. Therefore, we excluded mice except for studies that were not covered by other sections.
- Line 150: Consider adding some lines to facilitate reading the table.
Answer: Thank you for drawing our attention to this, the last line was missing. The layout of the table depends on the wishes of the publishers.
- Line 172: Add the acronym “(CAS)” as you did before for other institutions in lines 51 and 52.
Answer: We do not agree. In line 51 we refer to the Canadian Space Agency. In line 172 a facility is constructed at the Chinese Academy of Sciences. We have not abbreviated the latter since it is only used in line 172.
- Lines 197-203: How about biological experiments done in cubesats?
Answer: We included cubesats experiments. Please see lines 210-218, new ref. 34-36
- Line 208: Write all acronyms in full when mentioning them for the first time.
Answer: We have written acronyms in full when mentioning them for the first time in the text.
- Line 217: Consider adding a table similar to table 1, showing the different types of simulators, the gravity ranges, and any other relevant parameters.
Answer: That is an interesting suggestion worth considering. Using table 1 as a guideline, none of the parameter makes much sense: the simulated µg duration is pretty much unlimited and depends more on the effort made for maintaining the biological samples alive. Residual accelerations vary markedly, even within each experimental setup (e.g., at the center of rotation of an RPM vs. closer to the periphery), and the quality of the simulated µg is lively debated (as indicated by the references). Preparation time would be comparably short (days to weeks), given the actual simulation hardware is available and ready to go. As it is more common for researchers to invest in their own clinostat or RPM, cost estimates are difficult and would go down with every new experiment using already available hardware. Taking all these points into account, we do not see what parameters we could put together to present a meaningful table.
- Lines 261-268: Consider moving this into the previous section. And consider dividing that section into r-µg and s-µg structures and equipment.
Answer: We removed the paragraph.
- Line 271: Why are you referring microbiomes here? Consider moving this into the following section.
Answer: We moved the section.
- Lines 279-282: This is too general information. Consider removing or replacing with detailed and referenced data.
Answer: We have added more recent references (original articles) in order to support the statements. In particular:
Bai, P., Zhang, B., Zhao, X., Li, D., Yu, Y., Zhang, X., ... & Liu, C. (2019). Decreased metabolism and increased tolerance to extreme environments in Staphylococcus warneri during long‐term spaceflight. MicrobiologyOpen, 8(12), e917. https://doi.org/10.1002/mbo3.917
Zhang, B., Bai, P., Zhao, X., Yu, Y., Zhang, X., Li, D., & Liu, C. (2019). Increased growth rate and amikacin re-sistance of Salmonella enteritidis after one‐month spaceflight on China’s Shenzhou‐11 space-craft. Microbiologyopen, 8(9), e00833. https://doi.org/10.1002/mbo3.833
Bigley AB et al. 2019 NK cell function is impaired during long-duration spaceflight. J. Appl. Physiol. 126, 842–853. (doi:10.1152/japplphysiol. 00761.2018)
Shi L, Tian H, Wang P, Li L, Zhang Z, Zhang J, Zhao Y. 2021 Spaceflight and simulated microgravity suppresses macrophage development via altered RAS/ERK/NFκB and metabolic pathways. Cell. Mol. Immunol. 18, 1489–1502. (doi:10.1038/s41423- 019-0346-6)
Krieger, S. S., Zwart, S. R., Mehta, S., Wu, H., Simpson, R. J., Smith, S. M., & Crucian, B. (2021). Alterations in saliva and plasma cytokine concentrations during long-duration spaceflight. Frontiers in Immunology, 12, 725748.https://doi.org/10.3389/fimmu.2021.725748
As well as
Crucian, B. E., Choukèr, A., Simpson, R. J., Mehta, S., Marshall, G., Smith, S. M., ... & Sams, C. (2018). Immune system dysregulation during spaceflight: potential countermeasures for deep space exploration missions. Frontiers in immunology, 9, 1437.https://doi.org/10.3389/fimmu.2018.01437, as requested from reviewer 3
- Lines 290-310: You’re focusing on human immune systems, but this title is on bacteria, protists, and fungi. Revise this section and its title. Note that most statements should be referenced.
Answer: We moved the section to microbiome-host interaction.
- Line 311: This reads as a statement, not a title. Consider re-writing.
Answer: We have corrected this point.
- Line 314-315: You already explained this, no need to include this quotation.
Answer: The quotation was deleted accordingly.
- Line 321: Write acronyms in full at their first mention.
Answer: As mentioned above, we written acronyms in full when mentioning them for the first time.
- Line 336: Note that scientific names must be italicized.
Answer: We are aware, sorry for the lapse.
Line 350: How was that change you mention? Be more specific.
Answer: We have added more information about particular observed correlations between root metabolites and bacteria groups.
- Line 360: If you’re mentioning the effects of µg on bacteria, why didn’t you put this in the section with the appropriate title?
Answer: Because the entire section 3.2 is about bacteria, subsection 3.2.1 about microbiome-host interactions, subsection 3.2.2 about bacteria and virulence, and 3.2.3 about direct effects on bacteria. Therefore, we are in the right section.
- Line 364: This image, even though interesting, is not adding anything to the manuscript, I suggest removing it. If you decide to keep it, revise the legend and avoid repeating the information in the main text or explaining the figure content.
Answer: Figure 4 was removed according to your suggestion.
- Line 365: Add a reference.
Answer: A reference was added [63].
- Line 368-369: This does not read like a fact, more like an opinion. Avoid these types of unreferenced statements. Give more details on the mutation rates if these are known.
Answer: This statement was removed.
- Line 374-375: What type of s-µg? What was the level of increase?
Answer: We have corrected this point.
- Lines 359-407: These two sections seem to be about the same topic, consider turning them into only one or edit and move the text to make them into clear and separate topics.
Answer: 3.2.2 is specific for the effects of microgravity on virulence, 3.2.3 is not.
- Line 407: Your title mentions microorganisms, but your table lists organisms as a title in a column where you include cell types. Please revise and make these coherent. Consider adding lines to facilitate the reading of the table.
Answer: Yes, thank you. This is correct, because the experiments include cell-microorganism interaction. Therefore, the column mentions the cells which were infected with a microorganism.
- Line 409: This section title is supposed to be on gene expression in animals, but you start the sentence by mentioning a bacterium. Please revise.
Answer: The study investigated host-microorganism interaction. We wrote about V. fischeri before, but since in this study an animal participated, we decided to put it here.
- Line 417: elegans is a nematode, not an animal. You should revise the title of this section as well as the title.
Answer: Yes, it is a nematode, and as such an animal. The authors we cited agree to this in another publication: “C. elegans is the first animal in which RNA interference (RNAi) by double stranded RNA (dsRNA) was observed.” Higashitani, A., et al., C. elegans RNAi space experiment (CERISE) in Japanese Experiment Module KIBO, Uchu Seibutsu Kagaku 23/4 (2009), pp. 183–187.
- Line 432: Drosophila is not an animal. You should revise the title of this section as well as the title.
Answer: Respectfully, yes, it is an animal.
- Line 442-444: Please add a reference at the end of this sentence.
Answer: Thank you. A reference is Included in the revised manuscript [83].
- Line 459: Table 2 mentions many different experiments. Be specific.
Answer: Yes, we agree. We are sorry, that this seems to be confusing. The authors mention in their paper that the experiment was later repeated (not individually published). We are going to remove the reference from table 2.
- Line 460: What same animals are these? Same as what?
Answer: Yes, we agree, that is a bit biased. References 94 and 95 are from the same working group. They worked with the same animals: Ref. 94 studied the thymi, while in ref. 95 the results with the spleens and lymph nodes are published. We merged the two paragraphs to make it clearer.
- Line 461: To which group are you referring?
Answer: Please see the previous answer.
- Line 462: Revise.
Answer: We think that you mean GO terms. GO means Gene Ontology. We have given this abbreviation.
- Line 470: Is this section about the human eye and brain? It is not clear since you previously mentioned mice. What exactly are the differences in sections 3.4.1. and 3.4.2.? This is not clear since both titles seem to be about the same topic but then section 3.4.1. mentions endothelial cells, what is the relation of these cells with the eye or brain? Please revise.
Answer: This section is about the eye and brain either conducted on cells derived from these organs or on animals as well as human. Section 3.4.1 is about cells derived from brain and the eye. Section 3.4.2 describes studies on animals with emphasis on eye and brain. To improve the section the text has been revised as suggested. The section about endothelial cells has been move to section 3.5.
- Line 489: Please remove the word ”elegantly”, this is not appropriate.
Answer: As suggested, the word “elegantly” has been removed.
- Line 538: Which study are you mentioning?
Answer: The relevant reference has been inserted [99].
- Line 666: No need to detail where this work was presented. This information should appear on the reference.
Answer: The relevant text has been deleted.
- Lines 675-726: This section has several parts of the text highlighted, please revise format.
Answer: The relevant text has been formatted.
- Lines 735-874: This section title implies that the text here is all about humans. However, this is not the case. Note, for example, that your reference 128 is about mice macrophages, and from line 797 onward you’re not discussing humans anymore. In this section, you end by adding a paragraph about zebrafish. Please revise this section.
Answer: Thank you. Yes, we agree and have changed the title.
- Line 737: How was this shown?
Answer: Cultures of lymphocytes, that had been purified from blood samples drawn from crew members before and after flight, were exposed to mitogens. Activation was measured by incorporation of labeled thymidine or uridine into DNA or RNA respectively. A total of 41 astronauts and 12 cosmonauts were tested. This data was published in [128, 129].
- Lines 1608-1612: Consider adding this paragraph to the previous one.
Answer: This part of the conclusion was revised and added to the previous paragraph.
Reviewer 3 Report
The topic of this review is quite relevant and I suggest some modifications to increase the level of interest. My comments follow the text from its beginning to its end. I apologize in advance for not necessarily grouping them together.
The decision diagram is interesting but requires a better description because the passage from 7343 articles to 166 finally relevant suggests a real problem on the design of the sorting. The description of the reading of the articles, the choice of final inclusion and the work of each of the authors is missing (double reading of each publication, which authors wrote which part, there are notable differences in the expression according to the parts).
It is of course absolutely necessary to indicating the selection criteria of each article and what is its central message, as is the case for the table concerning cancer cells for all the 166 articles.
It is quite conceivable to limit the period of study of the literature on the subject to a defined and limited period, but for the reader to understand the new contributions, it is essential to make a brief summary of what was previously known.
The objective of such a review should be to allow the comparison of studies to find their specificities and common points. For this, on each tissue and organ cell type, a more relevant presentation of the results could be done. It is possible to review the MS to have a better formatting in terms of presentation.
Table 1 is very informative in this type of review, however the following descriptions of the different simulated microgravity systems could be improved by emphasizing what differentiates them from each other.
Figure 2A has been proposed very often, a better critical description of existing systems for exposing cell cultures to gravity change would be more informative for specialist and non-specialist readers of space biology. A critical description of clinostats would be welcome. The paragraphs of this part should be better differentiated.
L278-289 references are not indicated.
Figure 4 does not illustrate the point of the articles mentioned.
The paragraphs concerning the microbiota should mention the work of the group of JP Frippiat whose work has highlighted their alterations. This work was published before 2017 but is the basis of the research that the authors wish to highlight.
The data on C Elegans and mus musculus should not be put in table 2 devoted to micro-organisms but should be the subject of more complete comments in the ad-hoc paragraphs. Work on C Elegans could be better discussed.
Paragraph 3.4 it would be appropriate to complete this paragraph with a table indicating the nature and context of the experiments (animal, age of animals, cell in culture, duration of exposure to µG, type of exposure, etc.).
A reminder of the princeps experiments on endothelial cells would make it possible to better situate the interest of the genome expression analyzes which are included in this MS.
The reminder of the works of Cogoli is important since they are founding just like the works of Frippiat which are in line with them and which should also be recalled.
Copy the text with the correct nomenclature for writing people and proteins (Cacng1 for gene and CACNG1 for protein); reread the abbreviations and their order of appearance in the text (some are probably useless to avoid confusing them with genes.
For bone, a table with cellular models, gravitational modifications and genes whose expression is modified would be of great help in the critical reading of the results presented. Like table 3 of the MS.
In table 3 we have forgotten indication of Angpt2 which is mentioned in the text l1099.
L1027-1032 sentence which has no interest for the journal.
Review the MS to have a consistent formatting in terms of presentation.
Missing a paragraph on the risk of cancer in astronauts, on the interest of looking at the behavior of cancer cells in µG (division adhesion processes etc.
Warning between ground models of µG without radiation and real µG which includes a significant additional ionizing radiation
Diagram figure 5 shows similar diagrams from research publications (using the flumias system) mentioning it or indicating how this diagram is more original.
3.10.4 change the title it is definitely not about the behavior of lung cancer in µG.
same for 3.10.6. for this paragraph, the data on astronauts should be more detailed. Why not take the conclusions and perspectives of the publications concerning the cancers of astronauts to make a link with the data obtained later and described here ?
What is the use of the presentation of the risk of each cancer? The risks in astronaut population are not indicated for each type of cancer which is also very disturbing. Finally, the interest of µgravity to reduce the risk of cancer or if µgravity per se increases the risk of cancer should be developped.
L1463-L1482 there are error in chapter numbering.
L1536 SANS is not well written
Author Response
Answers to the Reviewers
We thank the editor and reviewers for their insightful comments regarding our manuscript; we are grateful for the possibility to respond to the comments and address them in a revised manuscript. The comments from the reviewers are addressed sequentially below. A revised manuscript with revision marked in red is provided along with a clean manuscript as pdf file.
Reviewer 3
Comments and Suggestions for Authors
The topic of this review is quite relevant and I suggest some modifications to increase the level of interest. My comments follow the text from its beginning to its end. I apologize in advance for not necessarily grouping them together.
Answer: Thank you for your kind words.
- The decision diagram is interesting but requires a better description because the passage from 7343 articles to 166 finally relevant suggests a real problem on the design of the sorting. The description of the reading of the articles, the choice of final inclusion and the work of each of the authors is missing (double reading of each publication, which authors wrote which part, there are notable differences in the expression according to the parts).
Answer: We have applied the PRISMA guidelines for assembling the flow diagram and have closely followed the officially suggested format. In order to be able to find all relevant literature, the search terms were deliberately chosen to be not too specific. Therefore, many hits proved to be unrelated to the review topic (as indicated in step 3 of the screening phase in Figure 1). It is not surprising, that search terms such as “(lung cancer) AND (microgravity)” or “(brain) AND (weightlessness)” etc. produce many results that could not be used for this manuscript, however, we do not see that as a flaw of our strategy, but rather as an attempt to be as exhaustive as possible.
The author contributions are listed at the end of the manuscript according to the publisher’s specifications.
- It is of course absolutely necessary to indicating the selection criteria of each article and what is its central message, as is the case for the table concerning cancer cells for all the 166 articles.
Answer: The selection criteria are given in the Methods section. We have added similar tables to other chapters with the requested information.
- It is quite conceivable to limit the period of study of the literature on the subject to a defined and limited period, but for the reader to understand the new contributions, it is essential to make a brief summary of what was previously known.
Answer: We feel that our introduction gives this short summary. While we agree with the reviewer that more information is always interesting, the purpose of an introduction is to briefly put the work into the context of the field and explain its purpose. By going into more detail as requested, this section would have become almost a mini-review itself, something we wanted to avoid.
- The objective of such a review should be to allow the comparison of studies to find their specificities and common points. For this, on each tissue and organ cell type, a more relevant presentation of the results could be done. It is possible to review the MS to have a better formatting in terms of presentation.
Answer: We have reviewed the available literature on each tissue or organ cell type as it appeared in PubMed/Scopus. Only the latest articles from 2017 until today were discussed with the exception of some important older articles of for example Dr. Augusto Cogoli, who started to investigate the immune system in the early 80s. The format of the review is given by the IJMS template.
- Table 1 is very informative in this type of review, however the following descriptions of the different simulated microgravity systems could be improved by emphasizing what differentiates them from each other.
Answer: The table is meant to show emphatically the differences of some key parameters, such as duration of µg, µg-quality, and costs (including time for preparation). However, we included some more differences regarding availability of µg platforms in the text.
- Figure 2A has been proposed very often, a better critical description of existing systems for exposing cell cultures to gravity change would be more informative for specialist and non-specialist readers of space biology. A critical description of clinostats would be welcome. The paragraphs of this part should be better differentiated.
The topic of this review is microgravity and not gravity change. The focus of this review is as follows: We summarize the current knowledge of the impact of microgravity on gene regulation. We focus on different types of bacteria, protists, fungi, as well as animals and cells exposed to s- and r-µg-conditions with regard to the eye, brain, cartilage, muscle, bone, endothelium, immune system, various cancer types in human studies as well as findings in plants.
We cited two references which deal specifically with the pitfalls of clinostats, further discussion does not really fit into this review. The paragraphs of this section describe the various µg platforms in ascending order of µg duration, which translates also into ascending order of cost and availability. Admittedly, the paragraph on drop towers is long due to the fact that in recent years new formats were introduced with smaller towers and the Einstein Elevator. The paragraphs about simulated µg are in increasing order of experimental volume from cells to humans.
- L278-289 references are not indicated.
Answer: We have added more recent references (original articles) in order to support the statements. In particular:
Bai, P., Zhang, B., Zhao, X., Li, D., Yu, Y., Zhang, X., ... & Liu, C. (2019). Decreased metabolism and increased tolerance to extreme environments in Staphylococcus warneri during long‐term spaceflight. MicrobiologyOpen, 8(12), e917. https://doi.org/10.1002/mbo3.917
Zhang, B., Bai, P., Zhao, X., Yu, Y., Zhang, X., Li, D., & Liu, C. (2019). Increased growth rate and amikacin re-sistance of Salmonella enteritidis after one‐month spaceflight on China’s Shenzhou‐11 space-craft. Microbiologyopen, 8(9), e00833. https://doi.org/10.1002/mbo3.833
Bigley AB et al. 2019 NK cell function is impaired during long-duration spaceflight. J. Appl. Physiol. 126, 842–853. (doi:10.1152/japplphysiol. 00761.2018)
Shi L, Tian H, Wang P, Li L, Zhang Z, Zhang J, Zhao Y. 2021 Spaceflight and simulated microgravity suppresses macrophage development via altered RAS/ERK/NFκB and metabolic pathways. Cell. Mol. Immunol. 18, 1489–1502. (doi:10.1038/s41423- 019-0346-6)
Krieger, S. S., Zwart, S. R., Mehta, S., Wu, H., Simpson, R. J., Smith, S. M., & Crucian, B. (2021). Alterations in saliva and plasma cytokine concentrations during long-duration spaceflight. Frontiers in Immunology, 12, 725748.https://doi.org/10.3389/fimmu.2021.725748
As well as Crucian, B. E., Choukèr, A., Simpson, R. J., Mehta, S., Marshall, G., Smith, S. M., ... & Sams, C. (2018). Immune system dysregulation during spaceflight: potential countermeasures for deep space exploration missions. Frontiers in immunology, 9, 1437.https://doi.org/10.3389/fimmu.2018.01437, as requested
- Figure 4 does not illustrate the point of the articles mentioned.
Answer: According to your suggestion, figure 4 is omitted in the revised manuscript.
- The paragraphs concerning the microbiota should mention the work of the group of JP Frippiat whose work has highlighted their alterations. This work was published before 2017 but is the basis of the research that the authors wish to highlight.
Answer: We have added references. Please see ref. [84, 97]. However, for the period of interest there was not much published by JP Frippiat.
- The data on C Elegans and mus musculus should not be put in table 2 devoted to micro-organisms but should be the subject of more complete comments in the ad-hoc paragraphs. Work on C Elegans could be better discussed.
Answer: The table is meant to show all organisms used in sections 3.2 and 3.3. We modified the title of the table.
- Paragraph 3.4 it would be appropriate to complete this paragraph with a table indicating the nature and context of the experiments (animal, age of animals, cell in culture, duration of exposure to µG, type of exposure, etc.).
Answer: As suggested, a table highlighting the findings in Eye and brain, has been included (Table 3).
- A reminder of the princeps experiments on endothelial cells would make it possible to better situate the interest of the genome expression analyzes which are included in this MS.
Answer: We included a short sentence in the beginning of the involved section highlighting one of the principal experiments on endothelial cells in regard to gene expression analysis. Please see ref. [109].
- The reminder of the works of Cogoli is important since they are founding just like the works of Frippiat which are in line with them and which should also be recalled.
Answer: We also have included a publication from Prof. Jean-Pol Frippiat relevant for this topic in the chapter 3.6 immune system. Please see ref. 130.
- Copy the text with the correct nomenclature for writing people and proteins (Cacng1 for gene and CACNG1 for protein); reread the abbreviations and their order of appearance in the text (some are probably useless to avoid confusing them with genes.
Answer: The correct nomenclature for genes and proteins has been applied.
- For bone, a table with cellular models, gravitational modifications and genes whose expression is modified would be of great help in the critical reading of the results presented. Like table 3 of the MS.
Answer: We have included new tables 3–6, and 8 in the revised manuscript.
- In table 3 we have forgotten indication of Angpt2 which is mentioned in the text l1099.
Answer: We do not have forgotten the protein angiopoetin-2. Table 7 (former Table 3) shows gene changes in cancer cells exposed to microgravity conditions.
- L1027-1032 sentence which has no interest for the journal.
Answer: This part has been removed.
- Review the MS to have a consistent formatting in terms of presentation.
Answer: The MS was extensively reviewed by all authors. The Cells publication template was used.
- Missing a paragraph on the risk of cancer in astronauts, on the interest of looking at the behavior of cancer cells in µG (division adhesion processes etc.
Answer: A paragraph on the risk of cancer in astronauts is given. Please see chapter 3.10 lines 1084-1116.
- Warning between ground models of µG without radiation and real µG which includes a significant additional ionizing radiation
Answer: A short sentence have been included on L276 to address this important point. One way to solve this issue is to use an onboard 1g centrifuge. But it has to be taken in account that our focus is microgravity on gene regulation in different organisms, cells and plants and not cosmic radiation or other stressors.
- Diagram figure 5 shows similar diagrams from research publications (using the flumias system) mentioning it or indicating how this diagram is more original.
Answer: The graphical representation of cell adherence and spheroid formation is inspired by our own paper Grimm et al. 2022, Figure 5 [PMID: 35328492]. Our team has performed the original FLUMIAS experiments on a parabolic flight mission and on the TEXUS-52 mission. The results are published in Corydon et al. 2016 [PMID: 26818711]. Moreover, Prof. Grimm is the PI of the German National Experiment ‘CANCEROIDS on FLUMIAS on the ISS’.
- 10.4 change the title it is definitely not about the behavior of lung cancer in µG.
Answer: The title of 3.10.4 was changed.
- same for 3.10.6. for this paragraph, the data on astronauts should be more detailed. Why not take the conclusions and perspectives of the publications concerning the cancers of astronauts to make a link with the data obtained later and described here ?
Answer: The title of 3.10.6 was changed. We have added data on astronauts as published by Reynolds et al. [174,175]. The results of the study [175] are added to the separate subchapters of 3.10.
- What is the use of the presentation of the risk of each cancer? The risks in astronaut population are not indicated for each type of cancer which is also very disturbing. Finally, the interest of µgravity to reduce the risk of cancer or if µgravity per se increases the risk of cancer should be developped.
Answer: GLOBOCAN 2020 is an online database which publishes the global cancer statistics and estimates of incidence and mortality in 185 countries for 36 types of cancer, and for all cancer sites combined. This is of interest for the readers of the review. Many of them are medical doctors and find this information helpful. The risks in the astronaut population are still unclear. But also, humans in space face the risks of cancer on Earth. We have added the information obtained by the interesting study of Reynolds et al. Please see [175].
- L1463-L1482 there are error in chapter numbering.
Answer: The error was corrected.
- L1536 SANS is not well written
Answer: The SANS part is improved in the revised manuscript (L1706-1712).
Round 2
Reviewer 2 Report
Dear authors,
Thank you for answering all my previous comments. The current version of the manuscript reads better. Many issues have now been sorted, but some others still need to be addressed:
Lines 43-44: This final conclusive sentence needs some context. It seems to be a different content from the previous sentence.
Line 421: Regarding tables and figures, note that these should stand alone, meaning that the reader should be able to read and understand their content without having to read the main text. Therefore, all acronyms should have their full meaning written in the legend. Also, when writing scientific species names, the second name is never written in capital letters and the first time you mention a species in the table, it should be written in full. Keep in mind that tables are treated separately from the text. Please revise all species names, and make sure that you only italicize the species names. As for the layout of the tables, as they are they are difficult to read. I understand that the publishers will have the final word on this, but if you check the bacteria Bacillus subtilis from your table, it is not clear if the respective reference is [74] or [75], or both.
Also, for the column with the title “Organisms” I suggest including only the species names. Information like “(colourless and pigmented strain)” does not seem relevant to this table. If you want to keep all this information in this column, consider changing its title.
Line 431: For species name, you do not need to write it as an acronym. Please delete “(C.)”.
Line 474: Regarding “the group” you’re mentioning, it is not clear that you are referring to the authors of the study, I suggest replacing it with something along the lines of “… Horie et al. …”.
Line 672: Regarding the information on the table from references 102 and 103, can't you put these data together?
Lines 1524 and 1526: are these terms supposed to be written in all capital fonts?
Line 1670: Note that you do not explain in the table what are the Col and Ws. Also, can't you collect information from several rows and different references into only one for the same "organism"? Or, can you organise the data from this table in a more readable manner? Can you present the common data from the different studies?
Regarding your conclusion, I still think it is too extensive. I suggest trying to be more assertive and shortening it.
Author Response
We thank the editor and reviewers for their insightful comments regarding our manuscript; we are grateful for the possibility to respond to the comments and address them in a revised manuscript. The comments from the reviewers are addressed sequentially below. A revised manuscript with revision marked using track-changes is provided along with a clean manuscript as pdf file.
Reviewer 2
Dear authors,
Thank you for answering all my previous comments. The current version of the manuscript reads better. Many issues have now been sorted, but some others still need to be addressed:
Lines 43-44: This final conclusive sentence needs some context. It seems to be a different content from the previous sentence.
Answer: We have changed the sentence.
Line 421: Regarding tables and figures, note that these should stand alone, meaning that the reader should be able to read and understand their content without having to read the main text. Therefore, all acronyms should have their full meaning written in the legend. Also, when writing scientific species names, the second name is never written in capital letters and the first time you mention a species in the table, it should be written in full. Keep in mind that tables are treated separately from the text. Please revise all species names, and make sure that you only italicize the species names. As for the layout of the tables, as they are they are difficult to read. I understand that the publishers will have the final word on this, but if you check the bacteria Bacillus subtilis from your table, it is not clear if the respective reference is [74] or [75], or both.
Answer: We have listed the abbreviations in the table legends.
We agree, the table is a bit hard to read without horizontal lines, but this will ultimately depend on the publisher’s format. There was, however, an error in the lines of references 74 and 75, which we corrected.
Also, for the column with the title “Organisms” I suggest including only the species names. Information like “(colourless and pigmented strain)” does not seem relevant to this table. If you want to keep all this information in this column, consider changing its title.
Answer: This has been corrected.
Line 431: For species name, you do not need to write it as an acronym. Please delete “(C.)”.
Answer: This has been corrected.
Line 474: Regarding “the group” you’re mentioning, it is not clear that you are referring to the authors of the study, I suggest replacing it with something along the lines of “… Horie et al. …”.
Answer: This has been corrected.
Line 672: Regarding the information on the table from references 102 and 103, can't you put these data together?
Answer: These are two different papers, one focusing on the proteome and the other one on the transcriptome. We do not wish to combine them.
Lines 1524 and 1526: are these terms supposed to be written in all capital fonts?
Answer: Yes, these Arabidopsis proteins are supposed to be written in all capital. However, we took this opportunity to make another sentence containing capital names more readable.
Line 1670: Note that you do not explain in the table what are the Col and Ws. Also, can't you collect information from several rows and different references into only one for the same "organism"? Or, can you organise the data from this table in a more readable manner? Can you present the common data from the different studies?
Answer: Good point, an explanation of the abbreviations was added to the table legend. Summarising the table further would mix up the information about which study used which experimental conditions and would make finding the appropriate reference for a dataset harder for the reader. We propose leaving the table as is, to conserve the informational value of each row. The common data points are explained in the text.
Regarding your conclusion, I still think it is too extensive. I suggest trying to be more assertive and shortening it.
Answer: We agree and have shortened the conclusion by approximately 25% as requested.
Reviewer 3 Report
I understand that your choice was to follow the PRISMA guidelines, but you can conclude on critical point of view from your analysis of the 188 publications. Indeed, this suggests an analysis of the 188 articles finally selected without prior critical reading. It should then be shown whether these articles contain all the elements necessary for them to be remarkable in terms of reproducibility of the data. Thus this critical reading would ensure an additional level of reliability on the reported data. The tables are there effectively to report the data of the various articles and the text must carry a critical analysis and not a simple repetition of the contents of the tables. Moreover, many sentences can be reworked to make the text more impactful (what is the point of saying that the works are recent when they do not date from before 2017). There could be a comment like 3.11.5 for plants for cell or animal experiments.
In my point of view, it very unfortunate that the work reported in the MS is not better compared to older work (confirmatory or not, how they open up new avenues of research, etc.) this considerably reduces the scope and therefore the impact of this MS. All the more unfortunate that this review is proposed by a large number of collaborators. But this the choice of the authors.
Major comments:
The choice of organs and tissues would require a paragraph in itself because it seems to me that some tissues have been deliberately excluded (not in the list of Material and methods).
Table 1, all of the costs should be indicated for all the experiments even if they are to be found elsewhere than in reference 16. The durations of µG for the ISS and Tiangong are not unlimited since these stations have finite lifetimes and that "unlimited" experiences have not yet taken place to my knowledge. Levels of µg from taxi flights are known and those from satellites as well (see BION M1 for example).
There is no table corresponding to paragraph 3.2 to access the criteria for using the bibliography.
Table 2 has a title that is not sufficiently informative is it a table including the modifications induced by µG between animal cells and pathogens (bacteria, fungi and viruses)?
The title of paragraph 3.3 is too ambiguous and ultimately is perhaps only an introduction to the following paragraphs.
There is no table for endothelial cells or the immune systemWhy not make tables for each organ or tissue you have chosen to identify as the one created for cartilage.
In your answers you indicate that it is a review on the µg and that as such you have not always taken into account the experiments carried out on clinostat, but there are some to mention. I think this should be clarified, by excluding or indicating the limitations of these devices, and indicating them in a table on the means of simulating µg. I persist in the remark concerning angiopoietin 2 which is mentioned in line 1182 but which is not found elsewhere in the text or in the tables. Can you then add a bibliographic reference to this and check if this is the angpt2 or ang2 or angiopoietin you want to talk about. I tested windows search on this abbreviation and word in all text.
The paragraphs on cancers in astronauts are ambiguous, this would require an explanation when you report the studies suggesting that the risk of cancer is increased decreased or not affected ref 201 173 174 175. What makes these publications and conclusions different.
Author Response
We thank the editor and reviewers for their insightful comments regarding our manuscript; we are grateful for the possibility to respond to the comments and address them in a revised manuscript. The comments from the reviewers are addressed sequentially below. A revised manuscript with revision marked using track-changes is provided along with a clean manuscript as pdf file.
Reviewer 3
I understand that your choice was to follow the PRISMA guidelines, but you can conclude on critical point of view from your analysis of the 188 publications. Indeed, this suggests an analysis of the 188 articles finally selected without prior critical reading. It should then be shown whether these articles contain all the elements necessary for them to be remarkable in terms of reproducibility of the data. Thus this critical reading would ensure an additional level of reliability on the reported data. The tables are there effectively to report the data of the various articles and the text must carry a critical analysis and not a simple repetition of the contents of the tables. Moreover, many sentences can be reworked to make the text more impactful (what is the point of saying that the works are recent when they do not date from before 2017). There could be a comment like 3.11.5 for plants for cell or animal experiments.
Answer: We are surprised to see what kind of assumptions the reviewer makes on our editorial processes merely based on the PRISMA flow chart. As stated earlier, the high number of initial hits stemmed from using redundant/overlapping search term, which generated many duplicates. How eliminating such duplicates can be a sign of a lack of critical reading is beyond our understanding. In the same context, the statement “I understand that your choice was to follow the PRISMA guidelines, but you can conclude on critical point of view from your analysis of the 188 publications.” is, frankly spoken, incomprehensible and we have difficulties to follow this line of reasoning without proper explanations and arguments. Rest assured, that each article has been read by at least two authors.
Furthermore, we have yet to encounter a review, where for every cited article a detailed justification is given why it was included in the text, as demanded by this reviewer. We feel, that by presenting the findings of the respective papers, the reasons for their inclusion become self-evident. Overall, we are a somewhat bewildered by the personal tone of the criticism.
We have reworked some text passages to be more precise, however concerning the statement “what is the point of saying that the works are recent when they do not date from before 2017?”: The authors and the reviewer might have a different idea about the timespan that the word “recent” implies, but we felt that covering papers from the last 5 years fits well into the generally recognized interpretation if this term.
The tables have been added at the request of reviewer 2.
In my point of view, it very unfortunate that the work reported in the MS is not better compared to older work (confirmatory or not, how they open up new avenues of research, etc.) this considerably reduces the scope and therefore the impact of this MS. All the more unfortunate that this review is proposed by a large number of collaborators. But this the choice of the authors.
Answer: The objective of this review is to focus on the recent knowledge about changes in gene regulation of among others organisms, tissues, plants and cells exposed to real and simulated microgravity conditions. Recent means the last five years.
We disagree, a large number of authors who are experts in the different topics is absolutely necessary. Several authors have years of experience and a large expertise in space research, gravitational biology, plant and cancer research as well as genetics, bioinformatics, space medicine and translational regenerative medicine.
Major comments:
The choice of organs and tissues would require a paragraph in itself because it seems to me that some tissues have been deliberately excluded (not in the list of Material and methods).
Answer: The focus of this review is gene regulation in microgravity. The majority of the organs and tissues that have been investigated are reviewed in this MS. The authors are experts in space medicine and space biology and know the available literature with respect to human cells and tissues exposed to microgravity and focus on gene expression/regulation.
Table 1, all of the costs should be indicated for all the experiments even if they are to be found elsewhere than in reference 16. The durations of µG for the ISS and Tiangong are not unlimited since these stations have finite lifetimes and that "unlimited" experiences have not yet taken place to my knowledge. Levels of µg from taxi flights are known and those from satellites as well (see BION M1 for example).
Answer: We agree, the information about the costs would be interesting. Budgetary information of this kind is usually not included in scientific publications. We simply did not find any reliable cost information for the platforms marked n/a.
It goes without saying that an experiment can only be performed on a Space Station as long as the Space Station exists. However, we adjusted the information. We also found information on g-levels in Foton satellites and included this reference.
There is no table corresponding to paragraph 3.2 to access the criteria for using the bibliography.
Answer: We have included a reference to Table 2 in the end of paragraph 3.2.
Table 2 has a title that is not sufficiently informative is it a table including the modifications induced by µG between animal cells and pathogens (bacteria, fungi and viruses)?
Answer: We adjusted the title according to the other tables.
The title of paragraph 3.3 is too ambiguous and ultimately is perhaps only an introduction to the following paragraphs.
Answer: The paragraphs before described bacteria (including microbiome and virulence) and fungi, 3.3 describes animals.
There is no table for endothelial cells or the immune system. Why not make tables for each organ or tissue you have chosen to identify as the one created for cartilage.
Answer: The two new tables have been included in the revision 2.
In your answers you indicate that it is a review on the µg and that as such you have not always taken into account the experiments carried out on clinostat, but there are some to mention. I think this should be clarified, by excluding or indicating the limitations of these devices, and indicating them in a table on the means of simulating µg. I persist in the remark concerning angiopoietin 2 which is mentioned in line 1182 but which is not found elsewhere in the text or in the tables. Can you then add a bibliographic reference to this and check if this is the angpt2 or ang2 or angiopoietin you want to talk about. I tested windows search on this abbreviation and word in all text.
Answer: This is a misunderstanding. Clinostat experiments are included in this paper when published during the time period 2017-today. The devices are explained in chapter 3.1.
Angiopoietin-2 is a result of our own study (Thyroid cancer cells on the ISS; Cellbox-2 experiment). The protein secretion was measured by multianalyte profiling technology. Angiopoietin-2 was elevated in space-flown samples. Correct: Angiopoetin-2 (Ang-2) is the protein; this was changed in the text. The reference is given in the text.
The paragraphs on cancers in astronauts are ambiguous, this would require an explanation when you report the studies suggesting that the risk of cancer is increased decreased or not affected ref 201 173 174 175. What makes these publications and conclusions different.
Answer: You suggested to include papers studying the risk of cancer in astronauts. These are the available publications and the results. This is all what is known at the moment. The risk of cancer in space is not the focus of this MS. We added the information because of your suggestion in revision 1 and to inform the readers.